# Astrocytic Ca²⁺ signaling is reduced during sleep and is involved in the regulation of slow wave sleep

Laura Bojarskaite [1,8], Daniel M. Bjørnstad[1,8], Klas H. Pettersen [1], Céline Cunen [2], Gudmund Horn Hermansen[2], Knut Sindre Åbjørsbråten[1], Anna R. Chambers[3], Rolf Sprengel [4], Koen Vervaeke[3], Wannan Tang[1,5], Rune Enger [1,6,7,9 ✉] & Erlend A. Nagelhus [1,7,9]

Astrocytic Ca²⁺ signaling has been intensively studied in health and disease but has not been quantified during natural sleep. Here, we employ an activity-based algorithm to assess astrocytic Ca²⁺ signals in the neocortex of awake and naturally sleeping mice while monitoring neuronal Ca²⁺ activity, brain rhythms and behavior. We show that astrocytic Ca²⁺ signals exhibit distinct features across the sleep-wake cycle and are reduced during sleep compared to wakefulness. Moreover, an increase in astrocytic Ca²⁺ signaling precedes transitions from slow wave sleep to wakefulness, with a peak upon awakening exceeding the levels during whisking and locomotion. Finally, genetic ablation of an important astrocytic Ca²⁺ signaling pathway impairs slow wave sleep and results in an increased number of microarousals, abnormal brain rhythms, and an increased frequency of slow wave sleep state transitions and sleep spindles. Our findings demonstrate an essential role for astrocytic Ca²⁺ signaling in regulating slow wave sleep.

[1] Letten Centre and GliaLab, Division of Physiology, Department of Molecular Medicine, Institute of Basic Medical Sciences, University of Oslo, 0317 Oslo, Norway. [2] Statistics and Data Science group, Department of Mathematics, Faculty of Mathematics and Natural Sciences, University of Oslo, 0316 Oslo, Norway. [3] Lab for Neural Computation, Division of Physiology, Department of Molecular Medicine, Institute of Basic Medical Sciences, University of Oslo, 0317 Oslo, Norway. [4] Research Group of the Max Planck Institute for Medical Research, Institute for Anatomy and Cell Biology, Heidelberg University, 69120 Heidelberg, Germany. [5] Department of Clinical and Molecular Medicine, Norwegian University of Science and Technology, Trondheim, Norway. [6] Division of Anatomy, Department of Molecular Medicine, Institute of Basic Medical Sciences, University of Oslo, 0317 Oslo, Norway. [7] Department of Neurology, Oslo University Hospital, Rikshospitalet, 0027 Oslo, Norway. [8] These authors contributed equally: Laura Bojarskaite, Daniel M. Bjørnstad. [9] These authors jointly supervised this work: Rune Enger, Erlend A. Nagelhus. ✉email: rune.enger@medisin.uio.no

We spend approximately one third of our lives sleeping, yet the purpose of sleep remains one of the greatest unsolved mysteries in biology. Recent studies have shown that not only neurons, but also glial cells are essential for sleep[1]. Still, the nightlife of the main glial cell in the brain—the astrocyte—is poorly characterized. There is evidence that astrocytes regulate sleep drive[2], promote sleep-dependent brain waste clearance[3], and facilitate cortical oscillations that are important for learning and memory[4,5], but the signaling mechanisms that astrocytes employ to mediate these sleep-dependent functions remain elusive.

Astrocytic $Ca^{2+}$ signals have been extensively studied in vitro and in vivo in anesthetized and, more recently, in awake animals and are considered to orchestrate neuronal circuit activity by regulating extracellular ion concentration and promoting the release of signaling substances[6–8]. In addition, astrocytes not only sense local synaptic activity[9,10], but also respond with $Ca^{2+}$ signals to neuromodulators[11–13] that are involved in sleep-wake state regulation[14,15]. Furthermore, astrocytic $Ca^{2+}$ signals have been characterized under urethane anesthesia, a state that has some resemblance to sleep[4,5]. However, even though urethane anesthesia has similarities to sleep, anesthesia will always be fundamentally different from natural sleep as the patterns of physiological state progression are not identical, and there is a lack of awakenings and microarousals[16].

Recent advances in optical imaging and genetically encoded activity sensors have enabled high-resolution imaging of astrocytic $Ca^{2+}$ signals in unanesthetized mice, revealing an exceedingly rich repertoire of astrocytic $Ca^{2+}$ signals[17–19]. Conventional tools for analysis of astrocytic $Ca^{2+}$ signals are based on static regions-of-interest (ROIs) placed over morphologically distinct compartments. However, the highly complex and dynamic spatiotemporal nature of astrocytic $Ca^{2+}$ signals that has become apparent using ultrasensitive genetically encoded $Ca^{2+}$ sensors, is not captured by static ROI analyses[17,18].

Here, we characterize astrocytic $Ca^{2+}$ signaling in mice during natural head-fixed sleep using an automated activity-based analysis tool. We employ dual-color two-photon $Ca^{2+}$ imaging to capture the activity of neocortical astrocytes and neurons simultaneously, combined with electrocorticography (ECoG), electromyography (EMG), and behavioral monitoring. We show that $Ca^{2+}$ signaling in astrocytes is reduced during sleep compared to wakefulness and exhibits distinct characteristics across sleep states. Strikingly, an increase in astrocytic $Ca^{2+}$ signaling precedes transition from slow wave sleep (SWS)—but not rapid eye movement (REM) sleep—to wakefulness. Finally, we demonstrate that the inositol triphosphate ($IP_3$)-mediated astrocytic $Ca^{2+}$ signaling regulates SWS by maintaining uninterrupted SWS, and affecting SWS state dynamics and sleep spindles. Taken together, our data indicate a role for astrocytic $Ca^{2+}$ signaling in regulating SWS.

## Results

**Two-photon imaging of awake and naturally sleeping mice.** To explore the characteristics of $Ca^{2+}$ signaling in astrocytes and neurons across sleep-wake states, we employed dual-color two-photon $Ca^{2+}$ imaging of neurons and astrocytes in layer II/III of the mouse barrel cortex by viral delivery of the green $Ca^{2+}$ indicator GCaMP6f to astrocytes and the red $Ca^{2+}$ indicator jRGECO1a to neurons (Fig. 1a). We chose barrel cortex because it has been intensively studied and has a well-characterized circuitry, and because the sensory input is easily monitored by whisker tracking[20]. The *glial fibrillary acidic protein* (*GFAP*) and the human *synapsin1* (*SYN*) promoters were used to target astrocytes and neurons, respectively. Imaging was performed at a

frame rate of 30 Hz, in accordance with recent reports underscoring the importance of high image acquisition rates for capturing fast populations of astrocytic $Ca^{2+}$ events[17]. Concomitantly, we recorded mouse behavior with an infrared (IR)-sensitive camera, ECoG and EMG for classification of sleep-wake states (Fig. 1a). The transduced *SYN*-jRGECO1a and *GFAP*-GCaMP6f specifically labeled neurons and astrocytes without inducing astrogliosis or microglial activation (Supplementary Fig. 1).

We identified three different behavioral states of wakefulness by analyzing mouse movements on the IR video footage (Fig. 1b): locomotion, spontaneous whisking, and quiet wakefulness. Since locomotion in mice is tightly associated with natural whisking[21], our locomotion behavioral state comprises both movement and whisking. Using standard criteria on ECoG and EMG[22,23] signals, we identified three sleep states (Fig. 1c, d): non-rapid eye movement (NREM) sleep, intermediate state (IS) sleep, and REM sleep. NREM sleep and IS sleep are sub-states of SWS, where IS sleep is a transitional state from NREM to REM sleep, found at the end of a NREM episode and characterized by an increase in sleep spindle frequency, increase in sigma (10–16 Hz) and theta (5–9 Hz) ECoG power and a concomitant decrease in delta (0.5–4 Hz) ECoG power[23] (Fig. 1c–e). To verify that the head-fixed situation did not perturb sleep we compared sleep characteristics between freely moving and head-fixed mice. We found nearly identical sleep characteristics between the two conditions, except less time spent in REM sleep and higher ECoG power in delta and theta range in the head-fixed condition (Supplementary Fig. 2). The increase in delta and theta power could at least partially be explained by delayed sleep onset and consequently higher sleep pressure in head-fixed mice (Supplementary Fig. 2a), which has been shown to increase both delta and theta ECoG power in NREM and REM sleep[24].

To capture the high spatiotemporal complexity of astrocytic $Ca^{2+}$ signaling, we developed an automated activity-based analysis algorithm (Supplementary Fig. 3). The algorithm utilizes three-dimensional filtering and noise-based thresholding on individual pixels over time to detect fluorescence events. Connecting adjacent active pixels in space and time results in regions-of-activity (ROAs) that can subsequently be combined with conventional, manually drawn ROIs or analyzed separately. The specificity of the algorithm was tested by applying the ROA method on time series from control mice expressing a $Ca^{2+}$ insensitive fluorescent indicator (enhanced green fluorescent protein, eGFP) in cortical astrocytes (Supplementary Fig. 4). Furthermore, we compared the characteristics of $Ca^{2+}$ event detection with ROI and ROA analyses (Supplementary Fig. 5 and Supplementary Movies 1 and 2). $Ca^{2+}$ signals in astrocytic somata and processes were much more complex than what could be captured by static ROIs. Notably, small, low-amplitude events in microdomains remained undetected with ROI analysis (Supplementary Fig. 5c, d), resulting in up to 90% fewer detected events (Supplementary Fig. 5e, f).

In all, 13 h of wakefulness and over 15 h of natural sleep (7 h NREM, 5 h IS, 3 h REM) (Fig. 1f) in 6 wild type (WT) mice were analyzed. Representative wakefulness and sleep trials are shown in Supplementary Figs. 6 and 7, and Supplementary Movies 3 and 4.

**$Ca^{2+}$ signaling is reduced during sleep and is state specific.** We used the ROA analysis to explore astrocytic $Ca^{2+}$ signaling during wakefulness and natural sleep (Fig. 2). Astrocytic $Ca^{2+}$ signals across the sleep-wake cycle displayed a broad repertoire of size, duration and volume (Supplementary Fig. 8). The spatial extent of ROAs ranged from ~0.9 $\mu m^2$ (lower detection limit) to the full field-of-view (FOV) (Supplementary Fig. 8a), whereas the

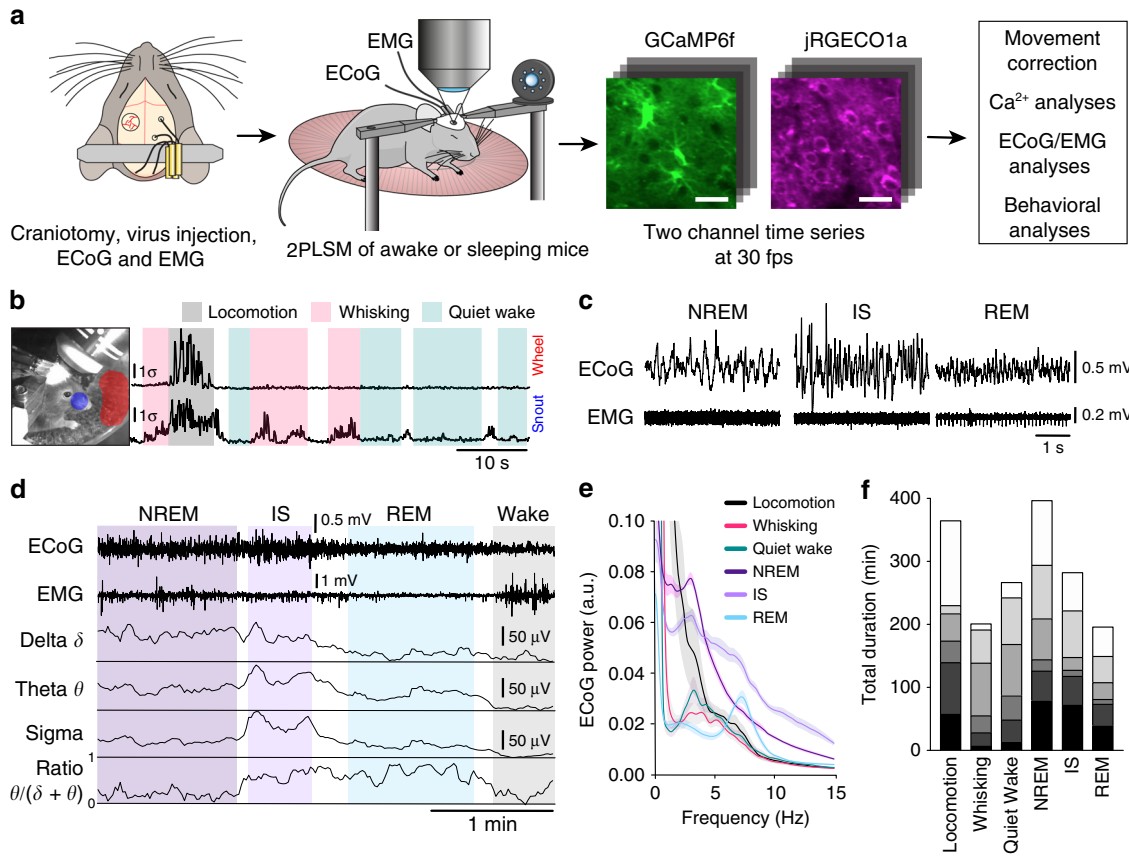

**Fig. 1 Two-photon imaging of Ca$^{2+}$ signals in awake and naturally sleeping mice. a** Experimental setup. A craniotomy exposing the barrel cortex was made, and a mixture of rAAV-*GFAP*-GCaMP6f and rAAV-*SYN*-jRGECO1a was injected for visualization of astrocytic and neuronal Ca$^{2+}$ signals, respectively. Scale bar 30 μm. In all 6 mice per genotype were prepared in this fashion. ECoG and EMG electrodes were implanted, and in combination with IR surveillance video allowed classification of the sleep-wake state. **b** Wakefulness was separated into locomotion, whisking, and quiet wakefulness based on movement of the mouse's snout and of the running wheel detected in the surveillance video. **c** Representative ECoG and EMG recordings of NREM, IS, and REM sleep states. **d** Sleep is categorized into NREM, IS, and REM sleep states based on the delta ($\delta$), theta ($\theta$), and sigma ($\sigma$) frequency band power, EMG and theta to delta frequency band ratio. **e** ECoG power density spectrum of sleep-wake states, normalized to the average total power in the 0.5–30 Hz frequency range during NREM sleep, $n = 6$ mice, mean ± SEM. **f** Total duration of each wake-sleep state from every mouse. Individual mice are indicated as separate blocks in bars, $n = 6$ mice. See Supplementary Figs. 1–7.

duration of the events ranged from 0.05 s to 100 s (Supplementary Fig. 8b). The majority of astrocytic Ca$^{2+}$ events were small and short-lasting across all sleep-wake states (~80% events <10 μm$^2$ and <1 s) (Supplementary Fig. 8d, e, f). On average, Ca$^{2+}$ events were of largest area and volume during active wakefulness (locomotion and whisking) (Supplementary Fig. 8g, h), and of longest duration during sleep (Supplementary Fig. 8i).

Since wakefulness encompasses a spectrum of sub-states that serve distinct perceptual and behavioral functions[25], we investigated astrocytic Ca$^{2+}$ signaling across the different states of wakefulness. Voluntary locomotion and spontaneous whisking were associated with increased percentage of active voxels ($x$-$y$-$t$) (11-fold during locomotion, 2.5-fold during whisking) and increased ROA frequency (3-fold during locomotion, 1.5-fold during whisking) compared to quiet wakefulness (Fig. 2c, d, middle), similar to previous reports[18,19]. By contrast, astrocytic Ca$^{2+}$ activity was reduced during sleep compared to overall wakefulness, with a 94% reduction in the mean percent of active voxels ($x$-$y$-$t$) and a 77% reduction in the frequency of ROAs (Fig. 2c, d, left). The reduction in astrocytic Ca$^{2+}$ activity was consistent for all sleep states, not only compared to active waking states, but also compared to quiet wakefulness (Fig. 2c, d, right). Although astrocytic Ca$^{2+}$ activity was substantially reduced during sleep compared to wakefulness, further analyses revealed

that the remaining activity during sleep nonetheless varied between states. The percentage of active voxels ($x$-$y$-$t$) and ROA frequency was lower during IS sleep than during NREM or REM sleep (Fig. 2c, d, right).

Importantly, our dataset yielded same trends when analyzed by another recently published astrocytic Ca$^{2+}$ event analysis tool[26] (AQuA), but not when analyzed by conventional ROI analysis (Supplementary Figs. 9 and 10). To conclude, we demonstrate that astrocytic Ca$^{2+}$ signaling is reduced during sleep compared to wakefulness, and is sleep-state specific.

**Ca$^{2+}$ signals in sleep are most frequent in processes.** Since Ca$^{2+}$ transients in astrocytic somata and processes may have different underpinnings and functional roles[18,27], we investigated the subcellular compartmentalization of Ca$^{2+}$ signals during natural sleep. As previously shown for wakefulness and anesthesia[18], activity maps indicated that astrocytic Ca$^{2+}$ signals were most frequent in astrocytic processes in neuropil also in natural sleep (Fig. 3a). We then quantified the subcellular distribution of the astrocytic Ca$^{2+}$ signals by running the ROA algorithm within manually drawn ROIs over astrocytic somata and neuropil (Fig. 3b). This analysis also confirmed that astrocytic Ca$^{2+}$ signals were of higher frequency within neuropil ROIs than within

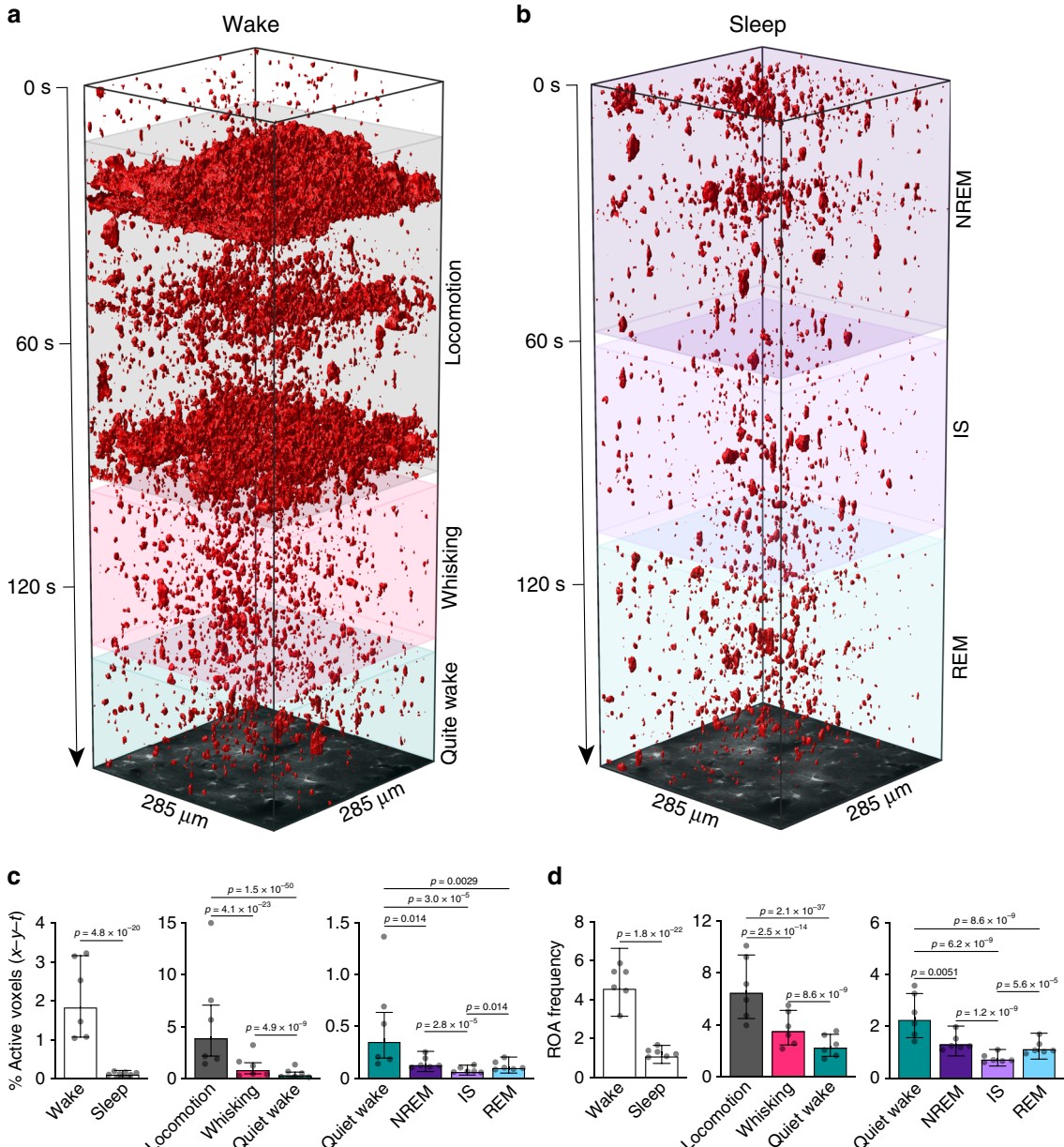

**Fig. 2 Astrocytic Ca$^{2+}$ signaling is reduced during sleep and is sleep state specific. a** and **b** Representative *x-y-t* rendering of ROAs during wakefulness (**a**) and sleep (**b**). **c** Percentage of active voxels (*x-y-t*) during overall wakefulness and overall sleep (left), during locomotion, whisking and quiet wakefulness (middle), and during NREM, IS, and REM sleep as compared to quiet wakefulness (right). **d** Same as **c** but for ROA frequency expressed as number of ROAs per 100 µm$^2$ per minute. Data represented as estimates ± SE and *p*-values (two-sided test, no adjustment for multiple comparisons) derived from linear mixed effects models statistics, $n = 6$ mice, 283 trials. For details on statistical analyses, see "Methods." See also Supplementary Figs. 8–10.

astrocytic somata ROIs across all sleep-wake states (Fig. 3b). Still, both astrocytic somata and processes displayed similar magnitude and direction of change in Ca$^{2+}$ signaling across sleep-wake states (Fig. 3b and Supplementary Fig. 11). To conclude, astrocytic Ca$^{2+}$ signals are more frequent in processes than somata not only during wakefulness, but also during sleep.

**Spatial stability of astrocytic Ca$^{2+}$ signals.** If astrocytic Ca$^{2+}$ signals are specifically integrated in sleep-wake dependent circuitry, one would expect to find some stability of active regions specific to sleep-wake states. We found that generally the overlap of active areas between the two episodes of the same state was relatively low (ca. 5%) except during episodes of locomotion

(ca. 25%), where typically most of the FOV was active (Fig. 4a). To evaluate whether some of the astrocytic Ca$^{2+}$ signals occurred at sleep or wakefulness specific locations, we first created individual heatmaps representing the level of Ca$^{2+}$ activity of every episode of all of the sleep-wake states within a FOV. Then, we analyzed the distances between heatmaps, here defined as 1 minus the Jaccard similarity coefficient (see "Methods"), by performing a permutational multivariate analysis of variance[28]. We first checked whether there was state-specific overlap within sub-states of wakefulness (locomotion, whisking, quiet wake) and within sub-states of sleep (NREM, IS, REM) (Fig. 4b). Here, we found that 25% of FOVs (19 of 76), including only wakefulness sub-states, exhibited state-specific activation—i.e., within these FOVs there was a smaller overlap between episodes of different

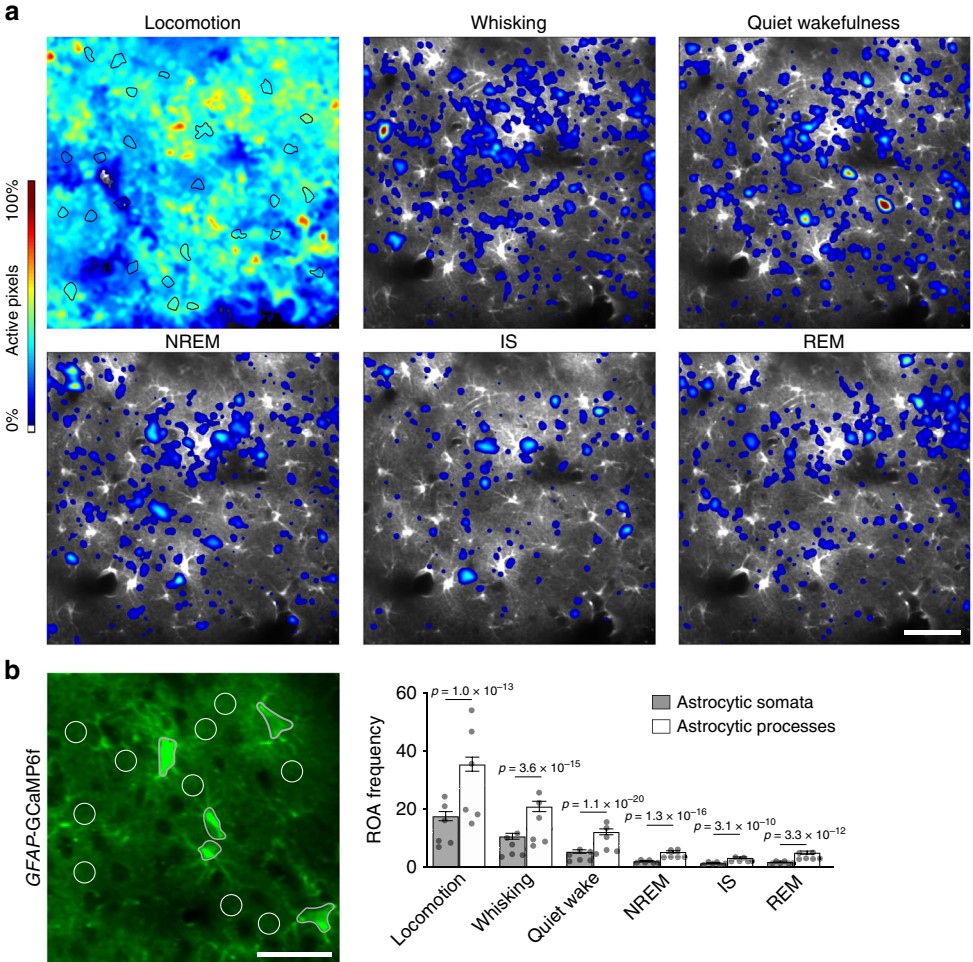

**Fig. 3 Astrocytic Ca$^{2+}$ signals during sleep are most frequent in processes. a** Representative ROA frequency heatmaps showing the localization of Ca$^{2+}$ signals during single episodes of locomotion, whisking, quiet wakefulness, NREM sleep, IS sleep, and REM sleep states. Astrocytic somata outlined in black in the locomotion image. Scale bar 50 μm. **b** Representative image of *GFAP*-GCaMP6f fluorescence in astrocytes with ROIs over astrocytic somata (black) and neuropil, containing astrocytic processes (white circles of 10 μm in diameter) (left). Scale bar 30 μm. ROA frequency, expressed as number of ROAs per ROI per minute, in astrocyte somata and processes from ROIs shown in left across sleep-wake states (right). Data represented as estimates ± SE and *p*-values (two-sided test, no adjustment for multiple comparisons) derived from linear mixed effects models statistics, $n = 6$ mice, 247 trials. For details on statistical analyses, see "Methods." See also Supplementary Fig. 11.

states (locomotion-whisking, locomotion-quiet wake, whisking-quiet wake), compared to episodes of the same state (locomotion-locomotion, whisking-whisking, quiet wake-quiet wake) (*p*-values under 0.05 as indicated by the dashed line, Fig. 4b, left). In these FOVs, the degree of overlap explained by sub-states of wakefulness was still relatively low, as indicated by the $R^2$ of ~0.3 (Fig. 4b, right). $R^2$ reflects the total overlap within episodes of the same state relative to the total overlap between episodes of same and different states. A high $R^2$ value indicates that episodes within the same state are very similar, while episodes from different states are very different. No state-specific activation was found between the sleep states (Fig. 4b, left).

Next, we assessed whether there could be activity patterns specific to either sleep or wakefulness. For FOVs with both sleep and wakefulness, 50% of FOVs (43 of 86) showed a significant level of state-specific activation (Fig. 4c, left). $R^2$ of FOVs with both sleep and wakefulness states (Fig. 4c, right) was generally higher than $R^2$ of FOVs with only wakefulness or only sleep states (Fig. 4b, right), suggesting that active areas in sleep are somewhat different from areas that are active during wakefulness.

Taken together, these data show a low degree of overlap of astrocytic Ca$^{2+}$ activity across sleep-wake states, but indicate a

moderate degree of sleep and wakefulness specific spatial activation patterns.

**Astrocytic Ca$^{2+}$ signals increase prominently upon awakening.** We observed that astrocytic Ca$^{2+}$ activity was not evenly temporally distributed within a given brain state, but rather was clustered at transitions from one state to another (Supplementary Figs. 6 and 7, red squares). To explore the relationship between state transitions and astrocytic Ca$^{2+}$ activity, we plotted ROA frequency aligned to the beginning of states, i.e., from quiet wakefulness to locomotion or whisking, and from either NREM, IS, or REM sleep to wakefulness (Fig. 5a). The start of locomotion and whisking was detected by movement of the whisker or wheel in the surveillance video, whereas the start of wakefulness during sleep-to-wake transitions was manually determined as the first sign of ECoG desynchronization[29] (Fig. 5b). Astrocytic Ca$^{2+}$ events were strongly clustered around specific brain state transitions (Fig. 5a, c, d). As expected, transitioning from wakefulness to sleep was associated with a decrease in Ca$^{2+}$ signals (Supplementary Fig. 12e). However, all other state transitions started with small Ca$^{2+}$ events that eventually merged to form larger and longer events (Supplementary Fig. 13). During transitions from

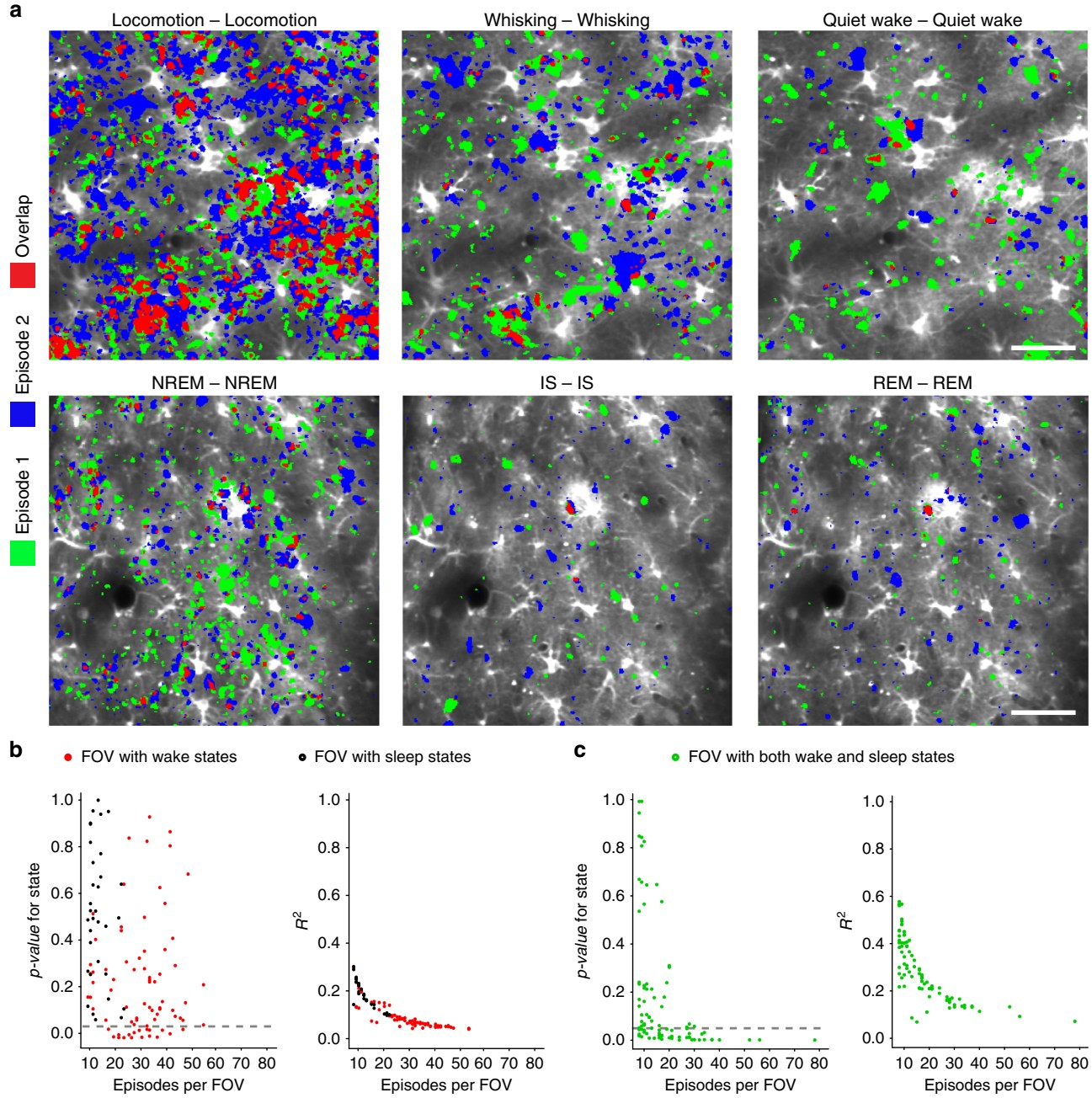

**Fig. 4 Spatial distribution of astrocytic Ca²⁺ signals across sleep-wake states. a** Representative astrocytic Ca²⁺ activity maps outlining areas active during episode 1 (green) and episode 2 (blue) of each of the sleep-wake states, and the overlap between the active areas (red). Scale bar 50 μm. In all 165 FOVs were compared. **b** Every dot represents a single FOV. Red dots are FOVs with only wakefulness sub-states whereas black dots are FOVs with only sleep sub-states. (Left) The probability, under the null model of no state-specificity in activation, of the observed degree of overlap between episodes of different states relative to the degree of overlap between episodes of the same state (locomotion-locomotion, whisking-whisking, quiet wake-quiet wake, NREM-NREM, IS-IS, REM-REM). Only FOVs below dashed line ($p = 0.05$) have a significant effect of state (representing some degree of state-specific overlap in activity). (Right) The degree of overlap within episodes of the same state relative to the total degree of overlap between episodes of same and different states (a high $R^2$ value indicates that episodes within the same state are very similar, while episodes from different states are very different). **c** Same as **b** but for FOVs with both sleep and wakefulness sub-states. $n = 6$ mice, 165 unique FOVs, 3793 episodes. Statistical analysis by use of permutational multivariate analysis of variance. For details on statistical analyses, see "Methods".

quiet wakefulness to locomotion or whisking, increases in astrocytic Ca²⁺ signaling followed the transition in the ECoG signal, muscle activity, and mouse movement (Fig. 5a, c, e). The onset of astrocytic Ca²⁺ did not differ between transitions to locomotion or whisking, but the peak Ca²⁺ response was larger for locomotion (Fig. 5f, g).

We then investigated the relationship between astrocytic Ca²⁺ activity and the transition from sleep to wakefulness (Fig. 5a, b, d). Waking up from REM sleep was associated with larger peak Ca²⁺ signals than other transitions (Fig. 5g), but was temporally more similar to state transitions of wakefulness, as astrocytic responses were delayed compared to the transition in ECoG

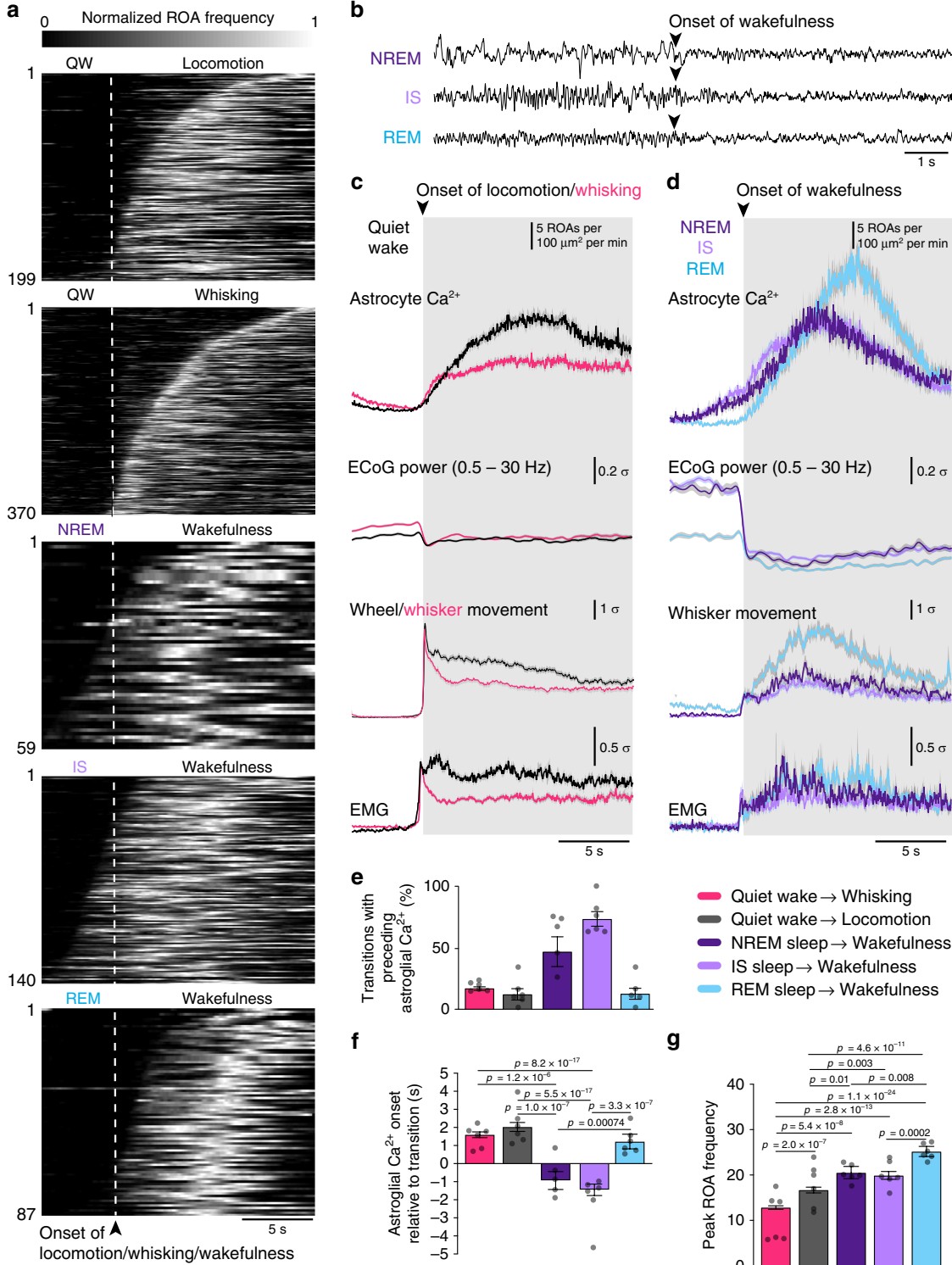

**Fig. 5 Astrocytic Ca$^{2+}$ signals increase prominently upon awakening. a** Temporal raster plots of normalized ROA frequency during the transitions (top to bottom): quiet wakefulness to locomotion ($n = 199$ transitions); quiet wakefulness to whisking ($n = 370$ transitions); NREM sleep to wakefulness ($n = 59$ transitions); IS sleep to wakefulness ($n = 140$ transitions); REM sleep to wakefulness ($n = 87$ transitions). **b** Examples of transitions from NREM, IS, and REM sleep to wakefulness in the ECoG traces. **c** Mean time-course of (top to bottom) astrocytic Ca$^{2+}$ signals and z-scores of ECoG power, wheel or whisker motion and EMG activation during transitions from quiet wakefulness to locomotion or whisking. Data represented as mean ± SEM, $n = 6$ mice. **d** Same as **c** but for transitions from NREM, IS, or REM sleep to wakefulness. **e** Percentage of transitions with preceding astrocytic Ca$^{2+}$ signals. Data represented as mean ± SEM, $n = 6$ mice, number of samples for each transition described in **a**. **f** Astrocytic Ca$^{2+}$ signal onset relative to state transition. Data represented as estimates ± SE and p-values (two-sided test, no adjustment for multiple comparisons) derived from a linear regression model, $n = 6$ mice, number of samples for each transition described in **a**. **g** Peak frequency of ROAs during state transitions. Data represented as estimates ± SE and p-values (two-sided test, no adjustment for multiple comparisons) derived from linear regression models, $n = 6$ mice, number of samples for each transition described in **a**. For details on statistical analyses, see "Methods". See also Supplementary Figs. 12 and 13.

signal, muscle activity, and mouse movement (Fig. 5a, d, f). However, to our surprise, astrocytic $Ca^{2+}$ signals typically preceded the awakenings from SWS (NREM and IS sleep states). We observed a prominent increase in astrocytic $Ca^{2+}$ signal frequency 1–2 s before the shift in ECoG, EMG, and mouse movement in 60% of NREM sleep to wakefulness and 72% of IS sleep to wakefulness transitions (Fig. 5d–f). As astrocytic $Ca^{2+}$ signaling did not precede the transition from NREM or IS sleep to wakefulness in all cases, we investigated if ECoG power could determine the temporal profile of astrocytic $Ca^{2+}$ signal onset (Supplementary Fig. 12). We found that high delta ECoG power was associated with earlier astrocytic $Ca^{2+}$ onset in NREM sleep (Supplementary Fig. 12a).

Our observation of enhanced astrocytic $Ca^{2+}$ signaling upon awakenings—particularly preceding transitions from SWS to wakefulness—suggests that astrocytic $Ca^{2+}$ signaling may play a causal role in modulating sleep-to-wake transitions.

### Correlation of astrocytic and neuronal $Ca^{2+}$ activity.

Having identified sleep-wake state-specific spatial and temporal features of astrocytic $Ca^{2+}$ activity, we then sought to investigate sleep-wake state-specific relationships between astrocytes and local neurons. To capture neuronal $Ca^{2+}$ activity, we measured jRGECO1a fluorescence in hand-drawn ROIs over neuronal somata and neuropil (Fig. 6a). The frequency of $Ca^{2+}$ signals in neuronal somata, like in astrocytes, was lower during sleep compared to wakefulness, although the reduction was modest (Fig. 6b). In order to examine the temporal relationship between astrocytic and neuronal $Ca^{2+}$ signals across sleep-wake states, we calculated the onset of astrocytic $Ca^{2+}$ signals in neuropil ROIs relative to $Ca^{2+}$ events in soma ROIs of neurons (Fig. 6c). During locomotion and spontaneous whisking, and to a lesser extent during quiet wakefulness, a population of astrocytic $Ca^{2+}$ signals displayed a modest degree of temporal alignment with nearby neuronal somatic $Ca^{2+}$ events (Fig. 6c), indicating some level of astrocyte-neuron synchrony during wakefulness, similar to previous reports[19]. By contrast, during all sleep states astrocytic $Ca^{2+}$ signals displayed a broad distribution of onset time differences relative to neighboring neurons (Fig. 6c). Next, we analyzed whether astrocytic signals were synchronized to neuronal activity in the neuropil. We quantified the correlation between astrocytic and neuronal $Ca^{2+}$ signals within the same neuropil ROIs (Fig. 6d). Similar to the temporal relationship with the neuronal somata, astrocytic $Ca^{2+}$ signals displayed a modest correlation with neuronal signals in neuropil during wakefulness, but were significantly decorrelated during sleep (Fig. 6e).

### Astrocytic IP₃-mediated $Ca^{2+}$ signaling regulates SWS.

Our observations of astrocytes during sleep revealed a spatiotemporal specificity of $Ca^{2+}$ activity, particularly during SWS, that could indicate causal roles for astrocytic signaling in sleep regulation. To identify these roles of astrocytic $Ca^{2+}$ signals in sleep, we employed the $Itpr2^{-/-}$ mouse model, in which astrocytic $Ca^{2+}$ signaling is strongly attenuated, but not abolished (as shown in experiments on awake and anesthetized mice)[18].

In agreement with previous reports, we found that $Itpr2^{-/-}$ mice exhibited reduced astrocytic $Ca^{2+}$ signaling in all states of wakefulness as measured by ROA frequency and active voxels (x-y-t) (Supplementary Fig. 14a, b)[18]. However, astrocytic $Ca^{2+}$ activity measured by both the percentage of active voxels (x-y-t) and ROA frequency did not significantly differ between WT and $Itpr2^{-/-}$ mice during sleep (Fig. 7b, c). Even so, $Itpr2^{-/-}$ mice exhibited $Ca^{2+}$ signals with disrupted spatiotemporal features— namely, longer duration and smaller spatial extent (Fig. 7a, d, e).

This finding was observed in all states of wakefulness, but during sleep was restricted to SWS (NREM and IS states).

Next, we investigated whether IP₃-dependent astrocytic $Ca^{2+}$ signaling had any effect on sleep dynamics. We compared sleep architecture and spectral ECoG properties between the two genotypes and found that $Itpr2^{-/-}$ mice exhibited more frequent NREM and IS bouts that were of shorter duration than in the WT mice (Fig. 8a, b). More fragmented SWS sleep could be a consequence of more frequent microarousals (short wakefulness intrusions characterized by a reduction of low-frequency ECoG power) and awakenings, which interrupt the sleep states, or more frequent NREM-to-IS and IS-to-NREM transitions. The number of awakenings did not differ between the genotypes in any of the sleep states (Fig. 8d). However, we found that $Itpr2^{-/-}$ mice have ~20 more microarousals per hour than WT mice (Fig. 8f). Such microarousals were associated with abrupt increases in astrocytic $Ca^{2+}$ signaling in WT mice, whereas no such response was observed in $Itpr2^{-/-}$ mice (Fig. 8g). Surprisingly, $Itpr2^{-/-}$ mice were completely devoid of the prominent astrocytic $Ca^{2+}$ increases seen upon awakenings in WT mice (Fig. 8e).

NREM-to-IS and IS-to-NREM transitions were more frequent in $Itpr2^{-/-}$ mice compared to WT mice (Fig. 8h), indicating abnormal SWS state dynamics in the knockouts. Interestingly, in WT mice, NREM-to-IS transitions were preceded by a decrease in $Ca^{2+}$ signaling, whereas IS-to-NREM transitions were followed by an increase in astrocytic $Ca^{2+}$ signaling. This was not the case in $Itpr2^{-/-}$ mice (Fig. 8i), and it is tempting to hypothesize that IP₃-mediated astrocytic $Ca^{2+}$ signaling is important to sustain uninterrupted SWS by regulating NREM and IS state transitioning and possibly preventing microarousals.

Finally, we assessed the spectral ECoG properties of NREM, IS, and REM sleep between the genotypes. We detected a decrease in delta power during NREM sleep, an increase in theta during REM sleep and an increase in sigma power during IS sleep in $Itpr2^{-/-}$ mice (Fig. 9a). ECoG activity in the sigma frequency range is indicative of sleep spindles—bursts of neuronal activity linked to memory consolidation[30]. As $Itpr2^{-/-}$ mice displayed higher sigma power in IS, we next evaluated the occurrence of sleep spindles in WT and $Itpr2^{-/-}$ mice (Fig. 9b). The frequency of sleep spindles in IS sleep was indeed considerably higher in $Itpr2^{-/-}$ mice than in WT mice (Fig. 9c). Intriguingly, sleep spindles were followed by an IP₃-dependent increase in astrocytic $Ca^{2+}$ signals (Fig. 9d). These data indicate a role for astrocytic IP₃-mediated $Ca^{2+}$ signaling pathway in regulating the architecture and brain rhythms of SWS.

## Discussion

Astrocytes are emerging as crucial components of neural circuits that actively take part in signal processing in the brain. In anesthetized and awake state $Ca^{2+}$ signaling has been shown to be the central signaling mechanism in astrocytes, and our study here reports on astrocyte $Ca^{2+}$ activity in natural sleep. We have demonstrated that astrocytic $Ca^{2+}$ signaling changes prominently across the sleep-wake cycle, being reduced during sleep while abruptly becoming elevated upon awakening, often before behavioral and neurophysiological signs of the sleep-to-wake transition. Genetic ablation of IP₃R2, an important astrocytic $Ca^{2+}$ signaling pathway, led to abnormal sleep architecture, state dynamics and brain rhythms of SWS. Taken together, our data show that astrocytes are essential for normal slow-wave sleep through mechanisms involving intracellular $Ca^{2+}$ signals. The concept that a non-neuronal cell type is indispensable for appropriate SWS will guide future studies aimed at deciphering sleep regulatory mechanisms and identifying novel treatment strategies for sleep disorders.

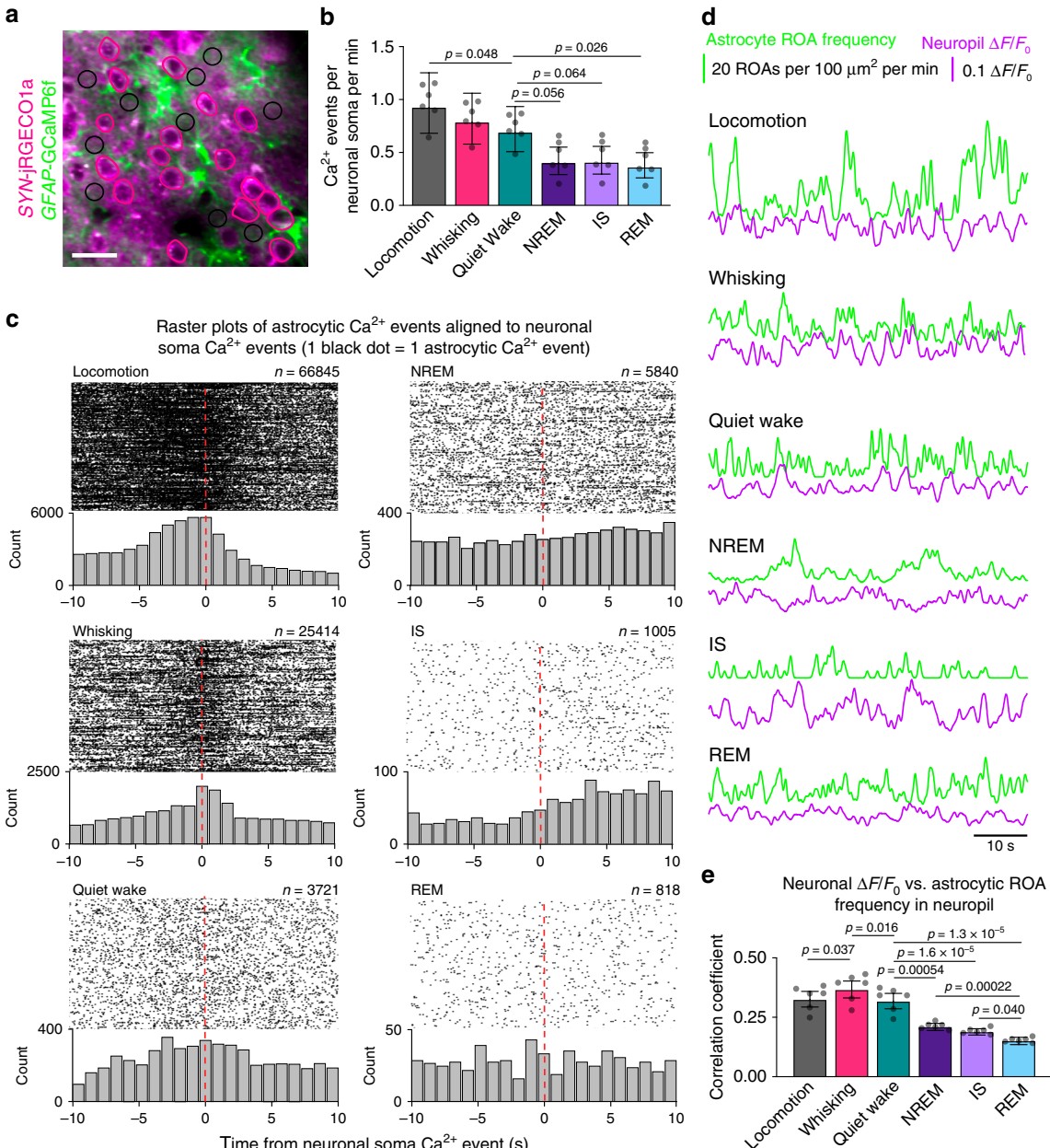

**Fig. 6 Correlation of astrocytic and neuronal Ca²⁺ signals. a** Representative image of *SYN*-jRGECO1a fluorescence in neurons and *GFAP*-GCaMP6f fluorescence in astrocytes, and ROIs over neuron somata (pink) and neuropil (black circles of 10 μm in diameter). ROIs were manually segmented in 92 time series from 6 mice. Scale bar 20 μm. **b** Frequency of Ca²⁺ signals in neuron somata across sleep-wake states . Data represented as estimates ± SE, and p-values (two-sided test, no adjustment for multiple comparisons) derived from linear mixed effects models statistics, $n = 6$ mice, 92 trials. **c** Raster plots and histograms (1 s bins) of the onset time difference for astrocytic Ca²⁺ events in neuropil ROIs (black circular ROIs in **a**) relative to a Ca²⁺ event in neuron somata ROIs (pink ROIs over neuron somata in **a**). Each black dot represents the temporal location of one astrocytic Ca²⁺ event relative to a Ca²⁺ event in a neuronal soma. **d** Example traces of continuous astrocytic Ca²⁺ event frequency in neuropil ROIs (number of ROAs per 100 μm² per minute), and neuronal Ca²⁺ signal $\Delta F/F_0$ in neuropil ROIs during locomotion, whisking, quiet wakefulness, NREM sleep, IS sleep, and REM sleep. **e** Correlation coefficient between continuous traces of astrocytic Ca²⁺ event frequency in neuropil, measured as number of ROAs per 100 μm² per minute, and neuronal Ca²⁺ signal $\Delta F/F_0$ in neuropil across sleep-wake states. Data represented as estimates ± SE and p-values (two-sided test, no adjustment for multiple comparisons) derived from linear mixed effects models statistics, $n = 6$ mice, 81 trials. For details on statistical analyses, see "Methods".

The use of ultrasensitive genetically encoded Ca²⁺ indicators and two-photon microscopy has revealed that astrocytes display a wide range of complex Ca²⁺ signals. These signals predominantly occur in fine astrocytic processes in neuropil, which contact thousands of neuronal synapses[9,17,18,31]. Deciphering the spatiotemporal dynamics of astrocytic Ca²⁺ signals in vivo can reveal how astrocytes coordinate and/or react to highly localized changes in neuronal and metabolic activity[17,32]. However,

conventional analytic methods based on static ROIs are insufficient in capturing the true complexity of astrocytic Ca²⁺ signaling. Here, we developed and employed an activity-based algorithm that enabled us to probe the complexity of astrocytic Ca²⁺ signals during sleep and sleep-wake transitions. Our ROA analysis was able to capture sleep and wake state-dependent modulations in astrocytes that were undetected using static ROIs. Further, analysis of our data with the ROA tool resulted in almost

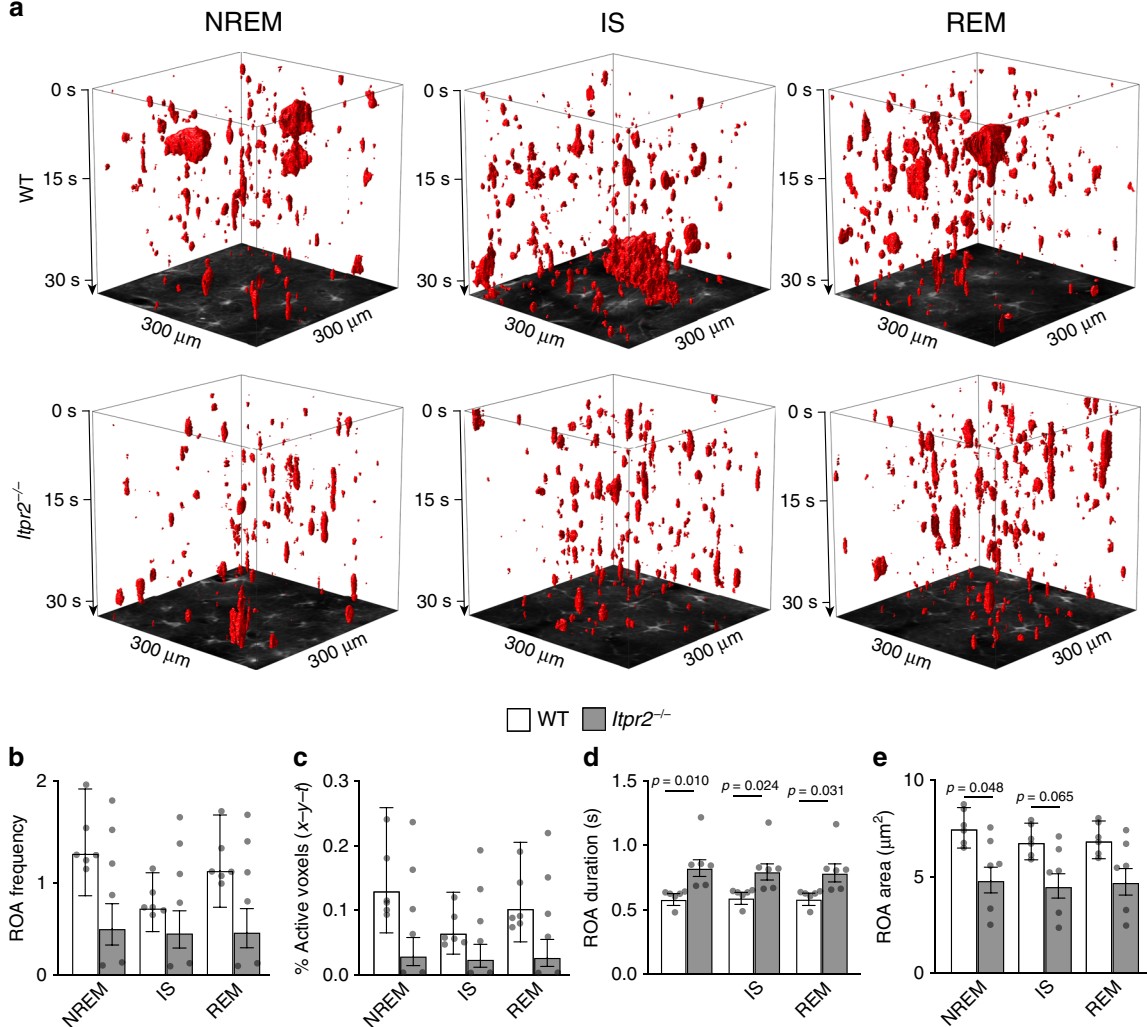

**Fig. 7 Astrocytic Ca²⁺ signaling during sleep is dependent on the IP₃ pathway. a** Representative *x-y-t* rendering of ROAs during NREM, IS, and REM sleep in WT and *Itpr2⁻/⁻* mice. **b–e** Mean ROA frequency expressed as number of ROAs per 100 μm² per minute (**b**), the percentage of active voxels (*x-y-t*) (**c**), ROA duration (s) (**d**), and ROA area (μm²) (**e**) during NREM, IS, and REM sleep in WT and *Itpr2⁻/⁻* mice. Data represented as estimates ± SE and *p*-values (two-sided test, no adjustment for multiple comparisons) derived from linear mixed effects models statistics, *n* = 6 mice, 196 trails for WT, *n* = 6 mice, 100 trials for *Itpr2⁻/⁻*. For details on statistical analyses, see "Methods." See also Supplementary Fig. 14.

identical astrocytic dynamics as detected with the recently published AQuA tool[26], but with the added benefit of fewer input parameters and shorter processing times. This data processing efficiency is increasingly important when coupled with high imaging frame rates, which result in large datasets but are necessary for capturing rapid (subsecond) changes in astrocytic Ca²⁺ signaling[17,19]. It is important to mention that our ROA method lacks spatial signal deconvolution to identify local Ca²⁺ signals that merge into larger signaling events. Even so, one could argue that, as astrocytes are connected in a syncytium, it is not necessarily conceptually most fruitful to think of small localized events merging into larger events as separate events. It is also important to acknowledge that single-plane imaging will lose information about the three-dimensional nature of astrocytic Ca²⁺ signals[17].

Until now, astrocytic Ca²⁺ signals in vivo have only been studied in anesthetized and more recently, in awake animals, with prominent differences in signaling patterns reported for the two brain states[17,33,34]. Even though urethane anesthesia has similar behavioral characteristics to sleep, such as loss of consciousness, distinct physiological and behavioral differences clearly

distinguish the two states, such as lack of awakenings and microarousals, and physiological cycling through the different sub-states of sleep. Here, we quantified astrocytic Ca²⁺ signaling during natural sleep. Rodent sleep is typically only divided into two sleep phases, namely NREM and REM sleep. At the end of a NREM sleep episode there is a short transitory phase to REM sleep termed the IS state[23]. This transitory phase is important for proper establishment of sleep states and its dysregulation is implicated in sleep pathologies such as narcolepsy[35]. Interestingly, one of the main signatures of IS sleep state is an increase in sleep spindles, which are corticothalamic neuronal bursts critical for learning and memory[36]. Still, IS sleep is one of the less studied aspects of sleep. In our study, we included this transitory phase in addition to NREM and REM sleep. Although sleep was associated with reductions in astrocytic Ca²⁺ signaling overall, astrocytes were significantly more silent in the IS sleep state in particular, compared to NREM and REM states. Further, IS-associated astrocytic silencing began during the transition from NREM to IS, before the neurophysiological (ECoG) signatures of IS sleep were observed (Fig. 8i). In *Itpr2⁻/⁻* mice, in which astrocytic Ca²⁺ signaling is strongly attenuated, SWS was fragmented due to

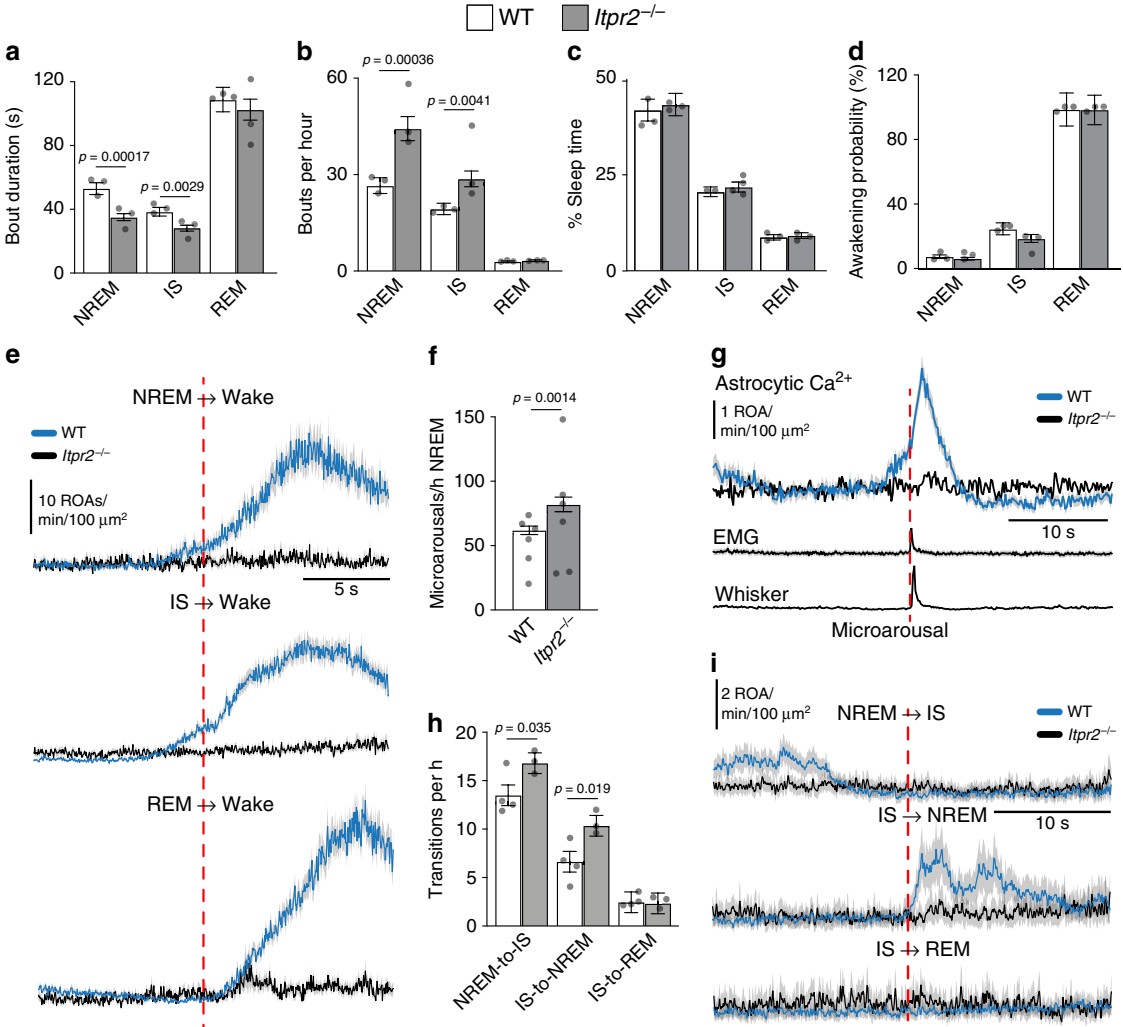

**Fig. 8 Astrocytic IP$_3$-mediated Ca$^{2+}$ signaling pathway regulates SWS. a–c** Bout duration (**a**), bouts per hour (**b**), and percentage recording time (**c**) of NREM, IS, and REM sleep in WT and *Itpr2$^{-/-}$* mice. For WT mice: $n = 3$ mice, 806 NREM episodes, 617 IS episodes, 92 REM episodes; for *Itpr2$^{-/-}$* mice: $n = 3$ mice, 1140 NREM episodes, 787 IS episodes, 86 REM episodes. **d** Awakening probability calculated as awakenings from NREM, IS, or REM sleep divided by total number of NREM, IS, or REM episodes. For WT mice: $n = 4$ mice, 806 NREM episodes, 617 IS episodes, 92 REM episodes; for *Itpr2$^{-/-}$* mice: $n = 3$ mice, 1140 NREM episodes, 787 IS episodes, 86 REM episodes. **e** Mean ± SEM of astrocytic Ca$^{2+}$ signals during transitions from NREM, IS, or REM sleep to wakefulness. For WT mice: $n = 6$ mice, 59 NREM awakenings, 141 IS awakenings, 87 REM awakenings; for *Itpr2$^{-/-}$* mice: $n = 6$ mice, 23 NREM awakenings, 56 IS awakenings, 27 REM awakenings. **f** Frequency of microarousals during NREM sleep. For WT mice: $n = 6$ mice, 355 microarousals; for *Itpr2$^{-/-}$* mice: $n = 6$ mice, 240 microarousals. **g** Mean ± SEM of (top to bottom) astrocytic Ca$^{2+}$ signals, $z$-scores of EMG activation and whisker motion aligned to the beginning of a microarousal. For WT mice: $n = 6$ mice, 303 microarousals; for *Itpr2$^{-/-}$* mice: $n = 5$ mice, 191 microarousals. **h** Number of NREM-to-IS, IS-to-NREM, and IS-to-REM transitions. For WT mice: $n = 4$ mice, 386 NREM-to-IS, 187 IS-to-NREM, 68 IS-to-REM transitions; for *Itpr2$^{-/-}$* mice: $n = 3$ mice, 440 NREM-to-IS, 267 IS-to-NREM, 61 IS-to-REM transitions. **i** Mean ± SEM of astrocytic Ca$^{2+}$ signals during NREM-to-IS, IS-to-NREM, and IS-to-REM transitions in WT and *Itpr2$^{-/-}$* mice. For WT mice: $n = 6$ mice, 90 NREM-to-IS, 35 IS-to-NREM, 25 IS-to-REM transitions; for *Itpr2$^{-/-}$* mice: $n = 6$ mice, 62 NREM-to-IS, 14 IS-to-NREM, 8 IS-to-REM transitions. Data represented as estimates ± SE and *p*-values (two-sided test, no adjustment for multiple comparisons) derived from linear regression models unless otherwise stated. For details on statistical analyses, see "Methods".

more frequent microarousals and more frequent transitioning between NREM and IS states. These results suggest a potentially critical role of astrocytic silencing during the transitional state between NREM and REM, or alternatively, a critical role for elevated astrocytic signaling in order to maintain uninterrupted SWS sleep.

In *Itpr2$^{-/-}$* mice, SWS was not only fragmented, but ECoG power characteristics were also affected, as there was a reduced power in the delta frequency range during NREM sleep. Altogether with the finding of shorter NREM bout duration detected in *Itpr2$^{-/-}$* mice, this sleeping pattern resembles one of a mouse line with astrocyte specific impairment of vesicular exocytosis, the dnSNARE mice[2]. Astrocytes are the source of the sleep pressure

agent and somnogen adenosine[2,37]. The release of adenosine is blocked in dnSNARE mice, resulting in impaired sleep homeostasis as measured by reduced ECoG delta power and time spent in sleep, and shorter NREM bout duration after sleep deprivation[2]. These data suggest that one of the downstream effects of IP$_3$-mediated Ca$^{2+}$ signaling during SWS could be the release of adenosine. IS sleep in *Itpr2$^{-/-}$* mice was associated with increased sigma ECoG power and considerably more sleep spindles, compared to WT mice. Similar to microarousals and SWS state transitions, sleep spindles in WT mice were associated with particular Ca$^{2+}$ signaling pattern, which was not observed in *Itpr2$^{-/-}$* mice. Although sleep spindles are known to be important for learning and memory[38], too many sleep spindles can be

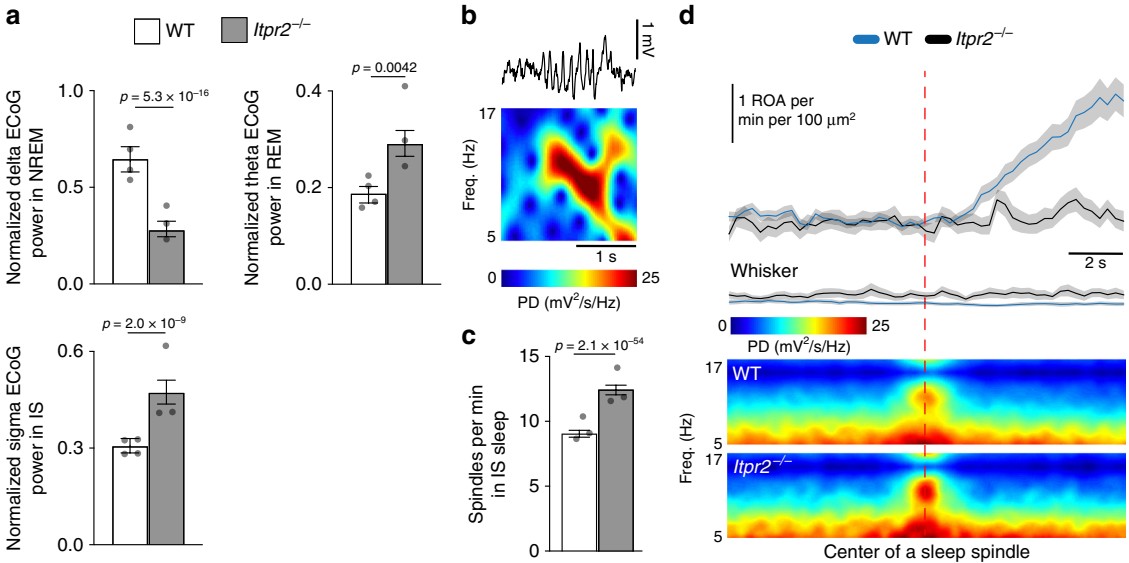

**Fig. 9 Astrocytic IP$_3$-mediated Ca$^{2+}$ signaling during sleep spindles. a** Mean ECoG power of delta (0.5–4 Hz) frequency during NREM sleep, theta (5–9 Hz) frequency during REM sleep, and sigma (10–16 Hz) frequency during IS sleep, normalized to 0.5–30 Hz total power. For WT mice: $n = 4$ mice, 785 NREM episodes, 668 IS episodes, 102 REM episodes; for *Itpr2$^{-/-}$* mice: $n = 3$ mice, 957 NREM episodes, 648 IS episodes, 74 REM episodes. **b** Representative example of a sleep spindle in ECoG trace and power spectrogram. **c** Mean number of spindles per minute during IS sleep. For WT mice: $n = 3$ mice, 516 IS episodes; for *Itpr2$^{-/-}$* mice: $n = 3$ mice, 560 IS episodes. **d** Mean ± SEM (top to bottom) astrocytic Ca$^{2+}$ signals, whisker movement $z$-score and ECoG power spectrograms aligned to the beginning of a sleep spindle. For WT mice: $n = 6$ mice, 243 sleep spindles; for *Itpr2$^{-/-}$* mice: $n = 6$ mice, 93 sleep spindles. Data represented as estimates ± SE and $p$-values (two-sided test, no adjustment for multiple comparisons) derived from linear regression models unless otherwise stated. For details on statistical analyses, see "Methods".

maladaptive. Excessive sleep spindles have been observed in humans with learning disabilities and have been shown to predict poor avoidance performance in rats[39–41]. Our data indicates that the IP$_3$-mediated astrocytic Ca$^{2+}$ signaling plays a key role in stabilizing and maintaining uninterrupted SWS and modulating SWS brain rhythms critical for learning and memory. However, since the gene knockout was global, we cannot rule out that altered astrocytic Ca$^{2+}$ signaling in non-neocortical (e.g., brainstem) regions affects neurons regulating SWS. Similarly, we did not study whether the manipulation of astrocytic Ca$^{2+}$ signaling affected specific cortical neurons thought to regulate SWS and spindles[22,23]. Future studies should delineate those specific pathways and circuits responsible for astrocytic IP$_3$-mediated modulation of sleep and investigate whether aberrant Ca$^{2+}$ signaling in astrocytes underlies sleep disorders.

The dynamics of brain state shifts are intimately linked to the brain-wide release of the neuromodulators noradrenaline and acetylcholine[14]. In recent years, astrocytes have been portrayed as one of the most important players in neuromodulator-driven state shifts in awake or anesthetized animals[42,43]. In our experiments, astrocytic Ca$^{2+}$ signaling typically preceded awakenings from SWS (both NREM and IS sleep), but not from REM sleep. This finding is consistent with the temporal profile of cortical norepinephrine (NE) release from locus coeruleus (LC) neurons upon awakening. The firing of LC noradrenergic (NA) neurons upon SWS-to-wake transitions has been shown to precede the onset of EEG activation by 0.3–1 s, whereas this predictive activity was not seen for REM sleep to wake transitions[29], suggesting that astrocytic Ca$^{2+}$ signals upon awakening are triggered by NE. NE acts on Gq-coupled α$_1$-adrenergic receptors on astrocytes, leading to Ca$^{2+}$ release from the endoplasmic reticulum through IP$_3$R2[12,18,44]. This might explain the absence of Ca$^{2+}$ surges upon awakening in the *Itpr2$^{-/-}$* mice, which lack IP$_3$R2 (Fig. 8e). It is important to mention that astrocytic Ca$^{2+}$ signals did not precede all SWS-to-wake transitions (Fig. 5a, e). The reason for this variance in Ca$^{2+}$ onset time is not entirely clear, and as we show in

Supplementary Fig. 12, there is some dependency on the depth of the preceding NREM sleep, suggesting that the prevailing neurochemistry of the brain tissue is important for how astrocytes respond to their triggers. These findings pose the intriguing question of what comprises an awakening and whether astrocytic awakening could be a new marker of the transition to wakefulness.

An unexpected finding was the lack of increase in astrocyte Ca$^{2+}$ signaling during REM sleep compared to NREM sleep. REM sleep is characterized by high cortical levels of extracellular acetylcholine compared to NREM sleep[45] and astrocytes have been shown to respond to acetylcholine with increased Ca$^{2+}$ signaling upon direct application or stimulation of cholinergic nuclei[13]. Therefore, we expected a strong increase in astrocytic Ca$^{2+}$ activity during REM sleep. This was, however, not the case, implying that acetylcholine alone is not sufficient to trigger astrocyte Ca$^{2+}$ elevations.

In recent years, astrocytes have been established not only as passive supporters of neurons, but also as active participants in the bidirectional communication between the two cell types. Firing patterns of cortical neurons vary across sleep-wake states and are of higher frequency during wakefulness than sleep[46]. We found that astrocytic Ca$^{2+}$ signaling was somewhat synchronized to neuronal activity during wakefulness, whereas very little correlation was found during sleep. It is tempting to hypothesize that during states of wakefulness, the observed astrocyte-neuron correlation reflects local neuronal activation specifically related to sensory input to the barrel cortex, whereas in sleep, astrocytic Ca$^{2+}$ signals are activated by subcortical brain state-specific circuitry, in line with our Ca$^{2+}$ activity overlap analysis showing that there were brain state-specific regions of Ca$^{2+}$ activity.

Neuronal activity data obtained from fluorescent Ca$^{2+}$ sensors as a proxy for neuronal firing should be interpreted with caution. Neuronal firing patterns are at least partially state-dependent[47], indicating that due to the sensitivity and kinetics of the sensor, Ca$^{2+}$ readouts could be affected in a non-linear manner, hence not faithfully reflecting neuronal firing across various states.

Moreover, as the neuronal $Ca^{2+}$ event detection is threshold-based, it is likely that neuronal activity is only detected above a certain spiking rate, possibly contributing to an underestimation of neuronal activity levels. Even so, our neuronal $Ca^{2+}$ data across sleep-wake states (Fig. 6b) is in line with other $Ca^{2+}$ imaging and electrophysiological studies showing reduced neuronal firing during sleep compared to wakefulness[22,46]. Future electrophysiological studies using unit recordings will have to confirm our observations.

In conclusion, our results indicate that even though reduced during sleep, astrocytic $IP_3$-mediated $Ca^{2+}$ signaling serves as a regulatory pathway to sustain uninterrupted SWS and maintain sleep spindles that are important for learning and memory.

## Methods

**Animals**. Male WT (C57BL/6J, Janvier Labs), $Itpr2^{-/-}$ ($Itpr2^{tm1.1Chen}$; MGI:3640970)[48] and Glt1eGFP mice[49] were housed on a 12-h light/dark cycle (lights on at 8 a.m.), 1–4 mice per cage. $Itpr2^{-/-}$ mice were backcrossed into a C57BL/6J background for at least 15 generations. Each animal underwent surgery at the age of 8–10 weeks, followed by accommodation to being head-restrained and two-photon imaging sessions (2–3 times per week) for up to 2 months. Adequate measures were taken to minimize pain and discomfort. Sample sizes were determined based on our previous studies using similar techniques (no power calculations were performed). No randomization or blinding was performed. All procedures were approved by the Norwegian Food Safety Authority (project number: FOTS 11983).

**Cloning and virus production**. The DNA sequences for the genetically encoded fluorescent $Ca^{2+}$ indicators GCaMP6f[50] and jRGECO1a[51] were first amplified by PCR from pGP-CMV-GCaMP6f and pGP-CMV-NES-jRGECO1a (Addgene) with 5′ BamHI and 3′ HindIII, and sub-cloned into the recombined adeno-associated virus (rAAV) vector pAAV-6P-SEWB[52] for generating pAAV-SYN-GCaMP6f and pAAV-SYN-jRGECO1a, respectively. The human glial fibrillary acidic protein (GFAP) promoter[53] was inserted with MluI and BamHI into pAAV-SYN-GCaMP6f construct for obtaining pAAV-GFAP-GCaMP6f. Serotype 2/1 rAAVs from pAAV-GFAP-GCaMP6f and pAAV-SYN-jRGECO1a were produced[54], and purified by AVB Sepharose affinity chromatography[55], following titration with real-time PCR (rAAV titers about $1.0–6.0 \times 10^{12}$ viral genomes/mL, TaqMan Assay, Applied Biosystems). For cortical rAAV-transduction of both astrocytes and neurons, rAAV-GFAP-GCaMP6f and rAAV-SYN-jRGECO1a were mixed 1:1.

**Surgical procedures and intrinsic imaging**. Mice were anesthetized with isoflurane. Two silver wires (200 μm, non-insulated, GoodFellow) were inserted epidurally into two burr holes overlying the right parietal hemisphere for ECoG recordings, and two stainless steel wires (50 μm, insulated except 1 mm tip, GoodFellow) were implanted in the nuchal muscles for EMG recordings. The skull over the left hemisphere was thinned for intrinsic signal imaging, a custom-made titanium head-bar was glued to the skull and the implant sealed with a dental cement cap. After 2 days, representations of individual whiskers in the barrel cortex were mapped by intrinsic optical imaging. The brain region activated by single-whisker deflection (10 Hz, 6 s) was identified by increased red light absorption. After 2 days, chronic window implantation and virus injection was performed[56]. A round craniotomy of 2.5 mm diameter was made over the barrel cortex using the intrinsic optical imaging map as a reference. A dental drill was used to carefully cut the skull with intermittent soaking and removal of debris with hemostatic sponges (Spongostan Dental, Ethicon) until only ~0.1 mm of the bone thickness was left. The bone flap was subsequently removed. The virus mixture (70 nL at 35 nL/min) was injected at two different locations in the barrel cortex at a depth of 400 μm. A window made of two circular coverslips of 2.5 and 3.5 mm was glued together by ultraviolet curing glue[57], was then centered in the craniotomy and fastened by dental cement. Mice with implant complications were excluded from the study.

**Behavioral training**. Mice were housed in an enriched environment with a freely spinning wheel in their home cages. One week before imaging, mice were habituated to be head-fixed on a freely spinning wheel under the two-photon microscope. Each mouse was trained head-fixed daily before the imaging for increasing durations ranging from 10 min on the first day to 70 min on the last. Mice that showed signs of stress and did not accommodate to head-restraint were not included in the study.

**Two-photon imaging**. Four weeks after the surgery, mice were imaged under an Ultima IV two-photon microscope from Bruker/Prairie Technologies[56], with a Nikon 16 × 0.8 NA water-immersion objective (model CFI75 LWD 16XW). The fluorescence of GCaMP6f and jRGECO1a was excited at 999 nm with a Spectra-Physics InSight DeepSee laser. All optical filters mentioned in the following description of the two-photon microscope are by Chroma Technology

Corporation: after the collected light is reflected towards the detection unit by the main dichroic filter (ZT473-488/594/NIRtpc), the signal light enters the detector house (four channels), passing a ZET473-488/594/NIRm filter, that is shielding the photomultiplier tubes from reflective light. Inside the detector house, the light is then split into two fractions separated at a wavelength of 560 nm by a dichroic filter (T560lpxr). The green light (GCaMP6f) is further guided by a secondary dichroic beam splitter at 495 nm (T495lpxr) and filtered by a ET525/50m-2p bandpass filter, whereas the red light (jRGECO1a) is similarly guided by a secondary beam splitter at 640 nm (T640lpxr) and subsequently filtered by a ET595/50m-2p bandpass filter. The emitted photons were detected with Peltier cooled photomultiplier tubes (model 7422PA-40 by Hamamatsu Photonics K.K.). Images (512 × 512 pixels) were acquired at 30 Hz in layer 2/3 of barrel cortex.

**Head-fixed sleep protocol**. To assist sleep in a head-fixed position, we adjusted the running disc position to mimic the body's natural position observed during unrestrained sleep[58]. We observed that locking the movement of the wheel once the mouse showed first signs of sleep, such as delta waves in ECoG and eyes closing, facilitated falling asleep. The imaging sessions of sleeping mice started at 9–10 a.m. (ZT 1–2), the beginning of light phase, and lasted until 3–6 p.m. (ZT 7–10). The mice did not have access to food or water while sleeping under the microscope, however, in natural conditions mice feed almost exclusively during the dark phase, ZT 12–24[59–61]. First signs of drowsiness, such as high delta power in the ECoG signal and eyes closing, were observed 15–45 min after head-fixation, and typically mice spent 90–120 min head-fixed under the microscope before falling asleep. Mice that did not show any signs of sleep within the first 2 h of head-fixation were removed from the microscope. The mice had an exact replica of the microscope running disc in their home cage, and in our hands this made a large difference in aiding the mice to fall asleep (i.e., initially we tried various types of stages like a spherical treadmill and a tube for immobilization with little success). Mice were not sleep deprived or manipulated in any other way before imaging to induce sleep.

**Behavior and electrophysiology recording**. ECoG and EMG signals were recorded using a Multiclamp 700B amplifier with headstage CV-7B, and digitized by Digidata 1440 (both Molecular Devices). Mouse behavior was recorded by an IR-sensitive surveillance camera. Data acquisition was synchronized by a custom-written LabVIEW software (National Instruments).

**Sleep-wake state scoring**. Sleep states were identified from filtered ECoG (0.5–30 Hz) and EMG signals (100–1000 Hz) based on standard criteria for rodent sleep[22,23] (Fig. 1c, d): NREM sleep was defined as high-amplitude delta (0.5–4 Hz) ECoG activity and low EMG activity; IS was defined as an increase in theta (5–9 Hz) and sigma (9–16 Hz) ECoG activity, and a concomitant decrease in delta ECoG activity; REM sleep was defined as low-amplitude theta ECoG activity with theta/delta ratio >0.5 and low EMG activity. Wakefulness states were identified using the IR-sensitive surveillance camera video by drawing ROIs over the running wheel and mouse snout (Fig. 1b). The signal in the wheel and snout ROIs was quantified by calculating the mean absolute pixel difference between consecutive frames in the respective ROIs. Voluntary locomotion was identified as signals above a threshold in the wheel ROI. Spontaneous whisking was defined in the snout ROI. Quiet wakefulness was defined as wakefulness with no signal above-threshold in both ROIs. Short and long quiet wakefulness episodes could represent different behavioral states, such as restful quiescence or freezing, and be influenced by the degree of habituation, which could affect astrocytic $Ca^{2+}$ signaling patterns. In our dataset >90% of quiet wakefulness bouts were shorter than 10 s and there was no difference in estimated $Ca^{2+}$ signaling levels when including long-lasting bouts compared to omitting them (Supplementary Fig. 15). Sleep episodes of >30 s duration were analyzed.

**Sleep-wake state transition and microarousal scoring**. For the transition from NREM and IS to wakefulness, onset of wakefulness was determined by the first sign of ECoG desynchronization[62] (activation) (Fig. 5b). During the transitions from REM to wakefulness, end of REM was identified by the interruption of sustained theta waves and the onset of desynchronized ECoG (Fig. 5b). Microarousals were defined as periods of above-threshold signal in the whisker ROI in the surveillance camera video (see also the above paragraph Sleep-wake state scoring) within NREM episodes with a duration of at least 0.3 s but <3 s.

**Comparison of sleep between freely behaving and head-fixed mice**. Male C57BL/6J mice were implanted with ECoG, EMG electrodes and a head-bar as described in the above section "Surgical procedures and intrinsic imaging". After recovering from the surgery (3 days), mice were habituated to both custom-built head-fixed and custom-built unrestrained sleep setups. The head-fixed setup consisted of head-bar clamps, a running wheel and IR-sensitive surveillance camera (as illustrated in Fig. 1a). The unrestrained sleep setup consisted of a transparent cage, a custom-built pulley system for the ECoG and EMG electrodes, and an IR-sensitive surveillance camera. After habituation, mice were placed in either the head-fixed or the unrestrained sleep setups and experimental sessions of sleeping mice started at 9–10 a.m. (ZT 1–2), the beginning of light phase, and lasted until

3–6 p.m. (ZT 7–10). ECoG and EMG signals were de-noised using a HumBug noise eliminator (Digitimer) and recorded using a Multiclamp 700B amplifier with headstage CV-7B, and digitized by Digidata 1440 (both Molecular Devices). Mouse behavior was recorded by an infrared-sensitive surveillance camera. Mice that did not show any signs of sleep within the first 2 h were removed from the setup. Sleep-wake states were scored as described in Sleep-wake state scoring.

**Immunohistochemistry.** Mice were deeply anesthetized with isoflurane and intracardially perfused with phosphate-buffered saline (PBS; 137 mM NaCl, 2.7 mM KCl, 4.3 mM $Na_2HPO_4·2H_2O$, 1.4 mM $KH_2PO_4$, pH 7.4, all from Sigma-Aldrich) and 4% paraformaldehyde (PFA, Merck) prior to decapitation. Brains were removed and fixed in ice-cold 4% PFA/PBS for overnight. Serial 70 μm sections were obtained on a vibratome (Leica). Free-floating sections were washed in PBS, incubated for 1 h in blocking solution (4% normal goat serum (NGS) and 0.3% Triton X-100 in PBS), washed with PBS, incubated overnight at room temperature with primary antibodies in PBS-Triton with 1% NGS (Primary antibodies: polyclonal chicken anti-GFP (1:3000, Abcam Cat#ab13970:), rabbit anti-GFP (1:4000, Abcam Cat#ab6556), mouse anti-NeuN (1:1000, Merck Cat#MAB377), mouse anti-GFAP (1:1000, Merck Cat#MAB360), rabbit anti-Iba1 (1:1000, Wako Cat#019-19741), washed in PBS, transferred into secondary antibody solution (all diluted 1:200) for 45 min (Secondary antibodies: Cy5-coupled anti-rabbit (Cat#111-175-144), Cy5-coupled anti-mouse (Cat#115-175-146), FITC-coupled anti-rabbit (Cat#111-095-144) and FITC-coupled anti-chicken (Cat#703-095-155) (Jackson ImmunoResearch Labs), washed with PBS and mounted on slides with Quick-hardening Mounting Medium (Sigma-Aldrich). Confocal images were acquired on a Zeiss LSM710/Elyra S1 confocal laser scanning microscope equipped with a 4x air objective or 20x water objective. Image analysis was done with ImageJ (v10.2, NIH).

**Image time-series alignment.** Imaging data were corrected for motion artifacts using the NoRMCorre movement correction software[63]. Images were analyzed by a custom-made image-processing toolbox in MATLAB (Mathworks).

**Region-of-activity (ROA) algorithm.** ROA algorithm consists of the following steps (see Supplementary Fig. 3): (A) Preprocessing: Imaging data was corrected for movement artifacts and smoothed in the spatial domain (gaussian smoothing, $\sigma = 2$ pixels) producing the time series $F$. (B) Calculating $\Delta F/F_0$ time series: A baseline image ($F_0$) was calculated by smoothing the pre-processed time series ($F$) in time (moving average filter, width 1.0 s), resulting in a lowpass filtered time series ($F_{LP}$), and calculating the mode of each of the pixels over time. The pre-processed time series ($F$) was then subtracted and divided by the baseline image ($\Delta F/F_0 = (F - F_0)/F_0$), resulting in a $\Delta F/F_0$ time series ($S$). (C) Calculating a noise-based activity threshold and thresholding the data: A moving average filter (width, $w = 1.0$ s) was then applied to the $\Delta F/F_0$ time series to create a smoothed, lowpass filtered time series ($S_{LP}$). A highpass filtered time series ($S_{HP}$) was then created by subtracting the smoothed time series ($S_{LP}$) from the original $\Delta F/F_0$ time series ($S$). The highpass filtering of S was done to estimate noise in our time series. To estimate the variance of the noise in S we assumed it could be approximated by variance of $S_{HP}$. As $S_{LP}$ is a moving average filtered version of S we would then expect that the standard deviation of the noise in $S_{LP}$ is a factor of $1/\sqrt{L_{filt}}$ lower than S, where $L_{filt}$ is the length (number of frames) of the moving average filter. A standard deviation image ($\sigma$) (the noise approximation of $S_{LP}$) was created by calculating the standard deviation of $S_{HP}$ and diving by $\sqrt{L_{filt}}$. Artifacts from fluorescence drifts[20] were removed by subtracting a 10 s moving average of $S_{LP}$ from $S_{LP}$, producing a final, bandpass filtered time series $S_{BP}$. Voxels in $S_{BP}$ were then thresholded by the corresponding pixels in the $\sigma$ image and multiplied by a common factor $k$ ($k = 5$ was used in the analysis), resulting in a three-dimensional (3D) matrix of active or inactive voxels. Connected components were then detected and the descriptive properties of the events were extracted. Adjacent active voxels in space and time were assigned to single ROAs. As vessel walls move within the FOV with vascular dilation and constriction, the regions over and immediately surrounding blood vessels were prone to artifacts and manually masked out before connecting voxels. For each ROA the starting time, maximal spatial extent (μm²), volume (μm²∗s), and duration (s) was recorded.

ROA frequency was calculated by counting all ROAs with starting times within a given frame and subsequently dividing by the sampled area (total FOV minus vessel masks) and the time per frame. The percentage of active voxels in a particular state episode was calculated by dividing the number of active voxels by the total number of voxels in that episode, while excluding the ignored areas. 3D renderings of ROA activity (Fig. 2) were made by outlining the ROAs and plotting with the MATLAB function patch(). The time resolution was decimated for improved performance and visual representation.

ROA heatmaps were created by calculating the percentage of active voxels along the temporal dimension. Values lower than 0.3% were left transparent while showing the average fluorescence of the astrocyte channel underneath (Fig. 3). To assess the persistence of hotspots in the heatmaps the 5% most active pixels were defined, and the overlap between different behavioral states were calculated. One heatmap was calculated per behavioral state for each unique FOV. For each heatmap a binary image was created with the 5% most active pixels marked as true.

The overlap between the binary images of different sleep-wake states was then calculated as the area of the intersection divided by the area of the union.

**ROA overlap analysis.** The overlap of active areas between episodes was evaluated by creating ROA heatmaps of the active voxels in each episode and estimating the overlap of these by calculating the Jaccard similarity coefficient. The Jaccard similarity coefficient is a number between 0 and 1, with 0 indicating no overlap and 1 indicating identical activation and was defined as $J(x, y) = \frac{\sum_i \min(x_i, y_i)}{\sum_i \max(x_i, y_i)}$, where $x$ and $y$ are vectors of pixels in each heatmap being compared.

Conversely, one minus the Jaccard similarity coefficient is a measure of dissimilarity between episodes and is also known as the Jaccard distance. We then calculated the distance matrix between all episodes from all FOVs, and compared these using the permutational multivariate analysis of variance—PERMANOVA—framework[28,64]. This method has a similar logic to ordinary ANOVA, but works on distance matrices. Say we have a FOV with 20 episodes belonging to three different states (i.e., locomotion, quiet, whisking) and we are interested in investigating whether episodes of the same are more similar than episodes of different states. If the FOV we are studying has strictly state-specific activation locations the distances within states will all be small, and the distances across states will all be large. Then the PERMANOVA will return a high $R^2$ value, and typically also a small $p$-value. The $p$-values in the PERMANOVA framework are obtained using permutations, and the approach does therefore not depend on any distributional assumptions. Further, the PERMANOVA framework includes both continuous covariates and factor variables (sleep-wake state is the factor variable in our case). Since the similarity between episodes is likely influenced by the overall activation of the episodes being compared, we have included average activation per episode as a covariate. Our result should therefore be interpreted as conditional on the total level of activation, i.e., given a certain level of activation do we see more similarities across states than between states? Only FOVs with eight or more episodes were analyzed. The $p$-values in Fig. 4 indicates how likely it is to observe such large distances across states, relative to the distances within states, under the null model where one assumes that there are no differences between the states (one dot represents a FOV). In some FOVs we only have comparisons for the awake states or sleep states (black or red, respectively). The green dots represent FOV where we have comparisons for both sleep and wakefulness. We consider the Jaccard similarity coefficient to be an intuitive and natural distance measure. However, it is important to note that the interpretation of results could be highly dependent on the choice of distance/similarity measure.

**AQuA[26] analysis.** Our data was acquired at a high scanning speed of 30 fps, and it seems like the AQuA method[26] is not optimized for our resultant signal/noise level. Consequently, imaging data was decimated from 30 to 3 Hz by applying a running average to improve signal-to-noise ratio and enable processing of the full dataset (running the AQuA algorithm on our 30 Hz imaging data would have taken close to 200 days in processing time). Default parameters were used except with a detection threshold ($thrARScl$) of 3 instead of 2. The frequency of events was then calculated by dividing the number of detected events by the total area of the FOV and time per frame.

**Hand-drawn ROI analysis.** Neuropil ROIs were marked as circles of 5 μm radius at least 5 μm away from the perimeter of astrocyte somata in areas without large visible cell processes. Astrocyte and neuron soma ROIs were manually drawn. The baseline ($F_0$) was defined as the mode of the traces. For astrocytes, mean fluorescence traces ($F$) were calculated by averaging the fluorescence intensity inside the respective ROIs. Relative fluorescence was defined with the formula $\Delta F/F_0 = (F - F_0)/F_0$. For neuron soma ($F_{neuron}$), neuropil signal was calculated from 5 μm wide regions around the neuron soma fitted to ($F_{doughnut}$) and subtracted from neuron soma traces. The relative fluorescence change for neuron soma was then defined as $\Delta F/F_0 = (F_{neuron} - F_{doughnut})/F_0$. $\Delta F/F_0$ from each ROI was filtered using a gaussian filter ($\sigma = 0.25$ s for GCaMP6f, $\sigma = 0.1$ s for jRGECO1a). A smoothed fluorescent trace (gaussian filter, $\sigma = 120$ s) was subtracted from the $\Delta F/F_0$ to remove drift. Noise traces were approximated by subtracting the smoothed trace from the raw $\Delta F/F_0$ trace. $Ca^{2+}$ events were detected as increases in $\Delta F/F_0$ larger than 2.5 times of the GCaMP6f noise trace and two times the standard deviation of the SYN-jRGECO1a noise trace. The duration of the signal was defined as the time points where the $\Delta F/F_0$ crossed 0 and the amplitude was the peak of the $\Delta F/F_0$ within the duration.

**$Ca^{2+}$ analyses in relation to brain state transitions.** Traces of ROA frequency were smoothed to reduce the effect of noise crossing the threshold. As we were concerned that a conventional gaussian kernel would skew the onset estimates to earlier values we instead used a kernel on the form $t·\exp(-t/\tau)$ ($\tau = 0.25$ s) that rises quickly, and tapers off with an exponential decay. The traces in a window ($-15$ to $15$ s) around every transition were collected to estimate the peak ROA frequency and the onset of $Ca^{2+}$ activity. The peak ROA frequencies were estimated from the maximum value of the ROA frequency traces. ROA frequency was normalized via z-score transformation, which were the mean and standard deviation was calculated from the first 10 s ($-15$ to $-5$ s from the transition). The

onset of $Ca^{2+}$ activity was then measured at the first point the traces crossed a threshold of 2.5 standard deviations.

**$Ca^{2+}$ signal correlation analysis**. For astrocytic $Ca^{2+}$ signals, an average ROA frequency trace was extracted from all neuropil ROIs per FOV from the *GFAP*-GCaMP channel. For the neuronal $Ca^{2+}$ signal, an average $\Delta F/F_0$ trace was extracted from all neuropil ROIs per FOV from the *SYN*-jRGECO1a channel. The baseline for neuronal signal was estimated by the mode of the trace. Both neuronal and astrocytic traces were subsequently smoothed with a gaussian filter ($\sigma = 0.25$ s). The maximum Pearson cross-correlation was calculated between astrocytic and neuronal $Ca^{2+}$ traces for each episode. The mean lag in all comparisons were less than a single frame.

**ECoG and sleep spindle analysis**. ECoG power spectrograms were calculated using the MATLAB function pspectrum() with default parameters from 0 to 256 Hz. To compare ECoG power across genotypes and different mice, raw data were normalized to the average total power in the 0.5–30 Hz frequency range during NREM sleep per mouse and average power was calculated using the MATLAB bandpower() function. To detect sleep spindles the ECoG signal was first normalized as above, followed by bandpass filtering in the frequency range 10–16 Hz (second order zero-phase Butterworth filter). The analytic signal was then found by using the Hilbert transform, as a measure of the instantaneous power of the bandpass filtered signal, and smoothed by a gaussian filter with a sigma of 0.2 s. Peaks in the smoothed data (found by the MATLAB function findpeaks()) were treated as putative sleep spindles. Peaks with a width at threshold of <0.5 s or >5 s were discarded.

**Statistical analyses**. Statistical analyses were conducted in R (version 3.6.0) and MATLAB using mixed effect regression models. As fixed effects, we included brain state (the binary overall sleep/overall wake, or the sub-states REM/IS/NREM/quiet wakefulness/whisking/locomotion), genotype (WT/$Itpr2^{-/-}$), cell type and subcellular compartment (astrocytic soma/astrocytic process/neuronal soma), and the interactions between these three categorical predictors. For the ROA frequency and the AQuA frequency, the percentage of active pixels we adjusted for differences in zoom factor by including the area covered by a pixel as a fixed predictor. For ROA size, duration and volume, zoom factor was included as a random effect.

We included individual mice as a random effect influencing the fixed effect of state. For all variables except ROI frequency, we assumed ordinary linear mixed effect models for the log-transformed responses (except Fig. 5e). The models were fitted using the nlme package[65] or MATLAB fitlme(). The adequacy of model assumptions was investigated by various residual plots[66]. In the cases were the residual plots indicated deviations from the assumption of constant residual variance, we extended the model by allowing the residual variance to vary as a function of genotype and state. Random effects that were estimated to be negligible, were removed from the models. For the ROI frequency, we fitted separate models for astrocytic somata, processes and neuronal somata. Astrocytic somata and processes were analyzed using a two-part mixed effects model, fitted by the GLMMadaptive package[67] due to a non-negligible probability of observing zero events. One part models the probability of an observation being equal to zero, the other part models the non-zero observations. We used a log-normal model for the non-zero observations and same fixed and random effects as described above. For the zero-part of the model we used the same fixed effects without interaction. The reported results from this analysis concern the non-zero-part of the model. The reported estimates are linear combinations of the fixed effect estimates from the fitted models. Approximate standard errors were found from the inverse observed Fisher information matrix. The reported *p*-values are based on the t-distribution, with degrees of freedom as provided from the nlme package. Standard error and *p*-values from analyses performed in MATLAB was extracted directly from the fitlme() function. No corrections for multiple comparisons were applied.

**Reporting summary**. Further information on research design is available in the Nature Research Reporting Summary linked to this article.

## Data availability
The raw data that support the findings of this study are available from the corresponding author upon reasonable request. The source data underlying Figs. 2c–d, 3b, 4b–c, 5e–g, 6b, e, 7b–e, 8a–d, f, h, 9a, c, and Supplementary Figs. 2a–d, 4b–c, 5e–f, 8g–i, 9, 10c–h, 11, 14a–e, 15 a–c, are provided as Source Data files.

## Code availability
The code supporting the current study along with a small test dataset is available as Supplementary Software. Source data are provided with this paper.

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

## Acknowledgements

Professor Erlend A. Nagelhus tragically died on 10th January 2020. We will always remember him as an inspiring leader, a good friend and a great scientist. We thank Gustavo Borges Moreno e Mello for his contribution to the establishment of the setup for awake animal imaging and monitoring, Arild Njå for guidance on electromyography, Xiaoyi Zhang and Mina Martine Frey for assistance with immunohistochemistry, and Johannes Helm for continuous technical support on the two-photon microscope. We thank Janne Grønli and Jelena Mrdalj for valuable help on establishing rodent sleep scoring. We acknowledge the support by UNINETT Sigma2 AS for making data storage available through NIRD, project NS9021K. This work was supported by the Research Council of Norway (grants #249988, #240476, and #262552), the South-Eastern Norway Regional Health Authority (grant #2016070), the European Union's Seventh Framework Program for research, technological development, and demonstration under grant agreement no. 601055, The National Association of Public Health, The Olav Thon Foundation and the Letten Foundation.

## Author contributions

Conceptualization: E.A.N., K.A.G.V.; Methodology: L.B., R.E., K.A.G.V., E.A.N., W.T., R.S.; Software: D.M.B., K.S.Å., R.E., K.H.P.; Formal analysis: D.M.B., R.E., C.C., G.H.H., K.H.P., L.B.; Investigation: L.B.; Resources: W.T., E.A.N., R.S.; Writing—original draft: L.B., R.E., E.A.N.; Writing—review and editing: L.B., R.E., E.A.N., A.C.; Visualization: L.B., R.E.; Supervision: R.E., E.A.N., K.A.G.V.; Funding acquisition: E.A.N.

## Competing interests

The authors declare no competing interests.
