## [Peer Review File · Nature Communications]

Peer Review Information

Manuscript title: Ca²⁺ signaling in mouse astrocytes is sleep-wake state specific and modulates sleep

Corresponding author name(s): Prof Rune Enger

Editorial notes:

none

Reviewer comments & decisions:

Reviewer comments, first version:

Reviewer #1 (Remarks to the Author):

In this work by Bojarskaite et al, authors conducted a thorough and detailed characterization of astrocyte Ca²⁺ activity in un-anesthetized mice as they naturally cycle through wake-sleep states. This is a much needed and, overall, well conducted study that would be well received in the field. I am concerned, however, that this work might not be fully ripe for publication yet. While authors convincingly demonstrate their capability to reliably measure and quantify astrocyte calcium activity, along with EEG, EMG and behavioral analysis, through naturally occurring states and state transitions in un-anesthetized mice (which is an achievement), the lack of clear questions and the lack of clear novel findings make this study resemble a "proof-of-concept" stage in its current form. This is apparent in the rather vague title or in that no major new findings are put forth, with the exception of the very last panel (Fig. 7F which authors hardly discuss). I am overall very supportive of this study, which I believe will be valuable to the field. But I believe authors need to re-arrange their manuscript and/or focus and further their analysis on some specific points and questions (see below) in order to deliver a few, key novel points rather than enumerate a list of features and changes, of which the physiological relevance is sometimes questionable (Fig. 6 C-E). All in all, while I believe this study will be very valuable to the field, I find difficult to fully support its publication in Nature Com until it provides a detailed account of a significant new finding that authors were able to elucidate through their newly developed analytical and "behavioral" method. I think this might be achievable rapidly without any major additional experimental work.

Regarding the technical and tool development:

1- The head-fixed/treadmill situation is, typically, rather uncomfortable for mice. Along with the many rounds of handling and "habituation" required, this makes such in vivo experiments relatively stressful

for the animals. This raises several points. First, that mice naturally fall asleep on the apparatus in these conditions seems quite extraordinary. As authors stated it, this has never been done before (that I'm aware of) and this is truly the main novelty their study brings. As such, many more details and necessary controls are required. I could not find in the methods or main text any details, tricks or procedures that authors used that permit mice to "naturally fall asleep" on the apparatus. Were they sleep deprived? Are recordings obtained at specific times of day, after mice have exhausted themselves running in their enriched environment? How many hours do mice spend, on average, head-fixed on the ball before they would fall asleep? These are key considerations, not only as far as reproducibility, but also because authors claim to study "natural sleep" here. How "natural" is mice sleep in a ball, in a head-fixed situation? Second, if this study is set to eventually mark the field as I think it is, then authors should convincingly demonstrate that the sleep pattern and architecture on the apparatus are indeed identical to those that would be recorded in a freely behaving (freely sleeping) animal. Freely behaving mice typically go through a stereotyped behavioral routine in preparation to sleep which, presumably, they are unable to carry out in a head-fixed situation. One would reasonably expect that this will alter their sleep and this should be quantified in this study. Authors could simply record EEG/EMG data from a few freely behaving mice in their home-cage, and then record from the same mice on the experimental head-fixed + treadmill apparatus. This longitudinal, paired paradigm would allow them to perform this key control experiment on a limited number of animals. Third, consistently with the two points above, what authors describe as "quiet wakefulness" might in fact be freezing: anxiety-driven behavior. While quiet wakefulness is common between bouts of sleep, rarely do mice, in the awake state, spend more than a few seconds completely immobile without grooming or whisking. Figure 1B shows 2 full minutes of such behavior, flanked by two episodes of active locomotion – which resembles freezing rather than quiet wakefulness. I think authors need to address this point thoroughly (i.e. experimentally).

2- Unless I missed it, I believe the novel ROA algorithm requires similar validation. Overall, this new algorithm is reminiscent of the analytical framework developed by the Poskanzer lab (AQUA, published a few weeks ago in Nat Neuroscience), which has been available in open-source to the community and has already been widely adopted by most over the past few years. I think this is already widely considered as the new standard and I would worry to see too many such new analysis tools being published independently to the detriment of consistency and reproducibility. To address this point I think authors should try and validate their main findings using AQUA, and show that while both algorithms might give different quantitative results as far as calcium signal detection and features, the effects of sleep and state changes on the said Ca^{2+} activity remains the same with either algorithm.

Other concerns:

3- Their study provides a wealth of rich and useful data and quantification, but going through the figures one by one, I felt that authors actually failed at putting forward any major new finding as mentioned above. Figure 1 is a description of their experimental set-up, which might as well be a supplemental figure. Figure 2 shows that astrocyte Ca^{2+} activity is state dependent, which is already known, albeit not with that degree of details (and with anesthesia instead of true sleep). Figure 3 shows that Ca^{2+} signaling is more frequent in the processes than in the soma, which is now a well-known fact in the field. Figure 4 shows that astrocytes activity precedes switches in certain neuronal states, which was partially known from Kira Poskanzer's work. Figure 5 shows that astrocytic and neuronal activity are somewhat disconnected during sleep while somewhat tuned to neuronal activity during wake states, which I believe is rather novel but the significance of this is not discussed and the experimental procedure here raises some concerns (see below). Figure 6 shows minor sleep architecture defects in IP3R2 KO mice, which are somewhat inconsistent with a publication from the Haydon lab (see below) and are rather marginal (with the exception of the increased spindles which

authors hardly discuss or interpret). To me, the most stunning finding, which only comes in Figure 7F, is that deleting IP3R2 in astrocytes prevents phase transition surges in Ca²⁺ activity in astrocytes all together. This is very intriguing but, here again, authors provide little clues regarding the meaningfulness of such observation. In addition, the idea that astrocytes “modulate” sleep has emerged in the field nearly a decade ago now and, in fact, the use of the term “modulate” by authors is indicative of a lack additional insights provided by their study on that matter.

4- It is my opinion that authors’ decision and justification to exclude micro-arousal episodes interrupting SWS (page 4) is unfounded and rather detrimental. On page 4 they justify their decision by stating “we aimed to characterize true sleep astrocyte Ca⁺ signals”. That’s rather unconvincing and on page 6, authors explain that they “investigated astrocyte Ca²⁺ signaling across the different states of wakefulness” which is in direct contradiction with the above justification. More importantly, there are very obvious Ca²⁺ signals restricted to these micro-arousal states that remind of sleep to wake transitions (Sup Fig. 4). I think this analysis of these states should be included in their study.

5- It has become very clear over the recent years that astrocytes respond to neuromodulators, and might be part of most neuromodulatory circuits; see Ma, Nature 2016, Paukert, Neuron 2014, Papouin, Neuron 2017; Mu, Cell 2019 and the work from the Nedergaard group – most of which are actually cited in the present manuscript. Therefore, I am a little surprised that authors provide little mechanistic insights into what might cause astrocyte Ca²⁺ activity to precede “neuronal awakening” in S↔W transitions, among other observations. It is easy to speculate that astrocytes might simply be responding to NE release and this would be relatively easy to investigate with tools currently available. At the very least, authors should discuss more thoroughly this point and situate their results more clearly in the context of these recent discoveries.

6- Page 8 and Figure 4. Evidently, these results bring the nearly philosophical question of what is awakening? A change in the EEG/EMG trace? A change in neuronal activity? A change in behavior? Or could astrocytes ‘awakening’ be a new marker of state transition? (I would personally welcome such discussion in the manuscript.) However, authors need to be careful with their interpretation here and clarify their overall statements. That astrocyte calcium activity precedes awakening in 60% of NREM↔wake transitions means it does not 40% of the time. If this is correct, this means that on average, astrocyte calcium activity is overall synchronous to the state transition. In this case, authors’ current statement would be a biased exaggeration. In addition, assuming that astrocytes Ca²⁺ activity in these particular transitions is the result of NE release, then one would expect to see a rostro-caudal ‘wave’ of activation in response to the release of NE by NE-releasing LC-neuronal projections through the cortex. This could be easily tested with some NE receptor (alpha1a) antagonists or ablation. This would also explain why some Ca²⁺ seem to precede awakening while others do not and authors should check if the timing of these Ca²⁺ events is spatially determined. If this is the case, then these astrocyte Ca²⁺ events are one of the first steps in the awakening process but I am unsure whether they could be described as “preceding” the wake state.

7- Page 9 and Figure 5. Employing calcium sensors as a proxy of neuronal signaling is a rather biased approach over such prolonged periods of time because it does not faithfully reflect neuronal firing across various states (which is really what matters here). Neuronal firing mode changes across the 24 hrs period and this will affect Ca²⁺ readouts in a non-linear manner. In other words, Ca²⁺ imaging might reflect more closely neuronal firing during awake states than it does during sleep states. That alone would explain the results presented in figure 5. Therefore, I believe these results should be interpreted more cautiously and might require additional supportive data that are not based on jRGECO fluorescence. In addition, authors used hand-drawn ROI to analyses neuronal activity, a method they reasonably criticize in the manuscript, and compare it to ROA-based astrocytic activity. Finally, neuronal activity (at least seen with GECO) is dramatically reduced during sleep such that the apparent lack of synchrony/relationship between neuronal activity and astrocytic activity during sleep

states could be due to under-sampling. Overall, these possible caveats make the data presented in this figure rather unconvincing. This is added to the fact that the significance and meaning of these finding is not discussed.

Other comments:

8- The first sentence of the abstract is an inaccurate overstatement. Similarly, the description of astrocytes as "housekeepers of the brain" in the third sentence of the introduction is quite shocking. Not only is this an outdated view. It is also incorrect.

9- Immunolabeling images of Sup Fig.1 come across as 'cherry-picked'. To be convincing, authors need to provide a wider view and an accurate quantification of the cell-specificity. Also, what exact brain region (cortical layer) is this?

10- Page 7. I find interesting authors' study of the stability (or lack thereof) of Ca²⁺ hotspots over state transitions. It is unclear however whether hotspots are stable in a given state over transitions? In other words, are hotspots associated with a given state consistently active during similar states, beyond the many states transitions? i.e. are there wakefulness-specific hotspots and SWS-specific hotspots? What is the significance of this? Maybe this could be extended to calcium signals other than the top 5% active ROAs.

11- The conclusion at the end of the top paragraph of page 10 is unwarranted or at least misleading ("indicating that astrocytic Ca²⁺ is uncoupled from local neuronal activity during brain states that lack dense neuronal discharge"). There is no indication of any sort that neuronal firing per se is a determining factor in the results of figure 5. In fact, in view of the most recent discoveries in the field, the disconnect between astrocytic and neuronal activity during sleep could be the result of neuromodulation acting on astrocytes to promote SWS neuronal activity (see work from Nedergaard lab). In this case, the "lack dense neuronal discharge" is mechanistically irrelevant.

12- Page 11. Line 6. I assume authors meant to refer to reference #24, not 23. Also #24, employing the VIPP mice, is not in accordance with their results contrary to what authors state. Please revise.

13- Contrary to what authors claim several times, the IP₃-dependent pathway in astrocytes is not really considered "key" to Ca²⁺ signaling. It has appeared to mostly control Ca²⁺ signaling in the soma and primary branches but is quite irrelevant when it comes to calcium signals in microdomains/processes (see work from the Khakh group mostly). This would need to be revised and taken in consideration throughout the manuscript.

14- Fig. 1, Sup Fig. 6 and others. Please provide a vertical scale and units of EEG, EMG and frequency band powers. I am surprised to see "arbitrary units" when the axis says "Power". Power does have a unit. Please explain and amend.

15- Figure 3 panel C needs to be simplified.

Reviewer #2 (Remarks to the Author):

The manuscript by Bojarskaite and colleagues investigates the role of glial calcium signaling in arousal states using the mouse as a model organism. The study builds on previous literature showing that glia may be relevant for regulating sleep states, but attempts to go a step further in delineating a role for IP₃-dependent Ca²⁺ signaling in the process.

The study is not uninteresting. It establishes a head-fixed two photon imaging prep to concurrently image neural and glial populations in mice undergoing state transitions, allowing for the observation that glial Ca^{2+} is enhanced upon arousal from sleep but not from transitions between NREM and REM sleep (Figs 1-4). To me that is the most robust finding in this study. The rest is preliminary, unconvincing or both, making the whole story as it currently stands more appropriate for a specialized audience rather than of broad impact.

Major:

1. Unclear to me conceptually why the barrel cortex is the object of study here. Is there a particular reason for this? This is not a brain region that is tied to arousal regulation in any particular way so it's hard to say why it was picked. It also raises the important point of how general are the findings here to other cortical/subcortical areas.

2. What exactly does Figure 5 say? There are multiple traces picked showing neural somata transients but no behavioral correlate is shown to understand what these responses are for. Basically there are two major problems with how the authors are relating neural activity (or glial activity for that matter) to behavior: 1. waking/locomotion/whisking are not unitary states, and the details of what the animal is doing will matter in understanding what the circuit is doing and what type of differences are meaningful across conditions 2. The correlation between what glia and neurons do across states needs to be adjusted for the differences in the frequency of both events across states. The authors ought to do subsampling of their waking data to make sure these correlations are robust to general reduction in sample size that they report.

3. The knockout data is simply not satisfying and seems like an afterthought here. The authors have the potential to make this into a broad interest study by simply determining the mechanism underlying this change in state regulation across the genotypes. Ideally, the authors would do this in a circuit-specific manner, not with a general knockout so that they can tighten their causal inference. Basically, figure out why the KO mice have fragmented sleep (less glial release of a substance that depends on this Ca^{2+} signal?)

Summary: I think this is a good study, but findings are preliminary and some require tightening

Reviewer #3 (Remarks to the Author):

This is an interesting and well-performed study dealing with an important and timely issue, the impact of astroglial activity on sleep. The authors use gene-encoding neuron- and glia- specific Ca^{2+} indicators, and a combination of physiological monitoring techniques, to relate phases of wakefulness and sleep to astroglial activity in the mouse barrel cortex. They found a strong relationship there, which prompts them to test a straightforward hypothesis that a major astroglial Ca^{2+} signalling pathway involving IP3 receptors plays a critical role in sleep regulation. Testing this hypothesis with a knockout model confirms their conclusions. The combination of new approaches and new results provides sufficient novelty and impact to the study of sleep. I have relatively minor comments, mainly related to data analyses, that the authors might want to deal with.

1. Fig. 2 and related data. The XYT ROA diagrams (Fig. 2A-B) provide a powerful tool to record and compare astrocytic Ca^{2+} signals that vary widely in their frequency, amplitude, volume spread, and duration. In this respect, summary data (Fig. 2C-D) contain somewhat curtailed information: the

percentage of active voxels (panel C) appears to combine duration and extent of the events, and the ROA frequency data (panel D) might depend on the pixel-connecting criteria in the ROA method (see below). The authors are encouraged to consider a more thorough analysis of their ROA data, perhaps based directly on their XYT records. This comment, however, does not affect qualitative conclusions of the study.

2. Supplementary Fig. 2, ROA method. The criteria for connecting active pixels need to be clarified. A common problem here is that 2D image acquisition cannot tell whether neighbouring active pixels represent fully separated Ca²⁺ signals or whether they represent parts of one contiguous 3D signal that is out of focus. This problem cannot be solved by the ROA method as it is and needs to be acknowledged. Again, it should have little impact on the qualitative conclusions of the study because similar segmentation procedures have been applied throughout the data sets. At the same time, the authors are encouraged to show more caution towards their quantitative estimates.

3. Figs. 3-4: As noted above, ROA frequency alone is a poor representative of signal dynamics. While this is partly acknowledged in the text, the data presentation here should include an unambiguous reference to ROA size and duration.

4. Fig. 5B data suggest that, during locomotion, barrel cortex neurons generate detectable Ca²⁺ signals, which supposedly reflect action potential firing, at a frequency of $\sim 1/\text{min}$. This slow rate is unrealistic and probably reflects relatively low sensitivity of Ca²⁺ imaging. In other words, in the present settings, neuronal activity is only detected above some high spiking rate. Without a proven direct relationship between neuronal spiking and recorded Ca²⁺ signals the spectral power data in Fig. 5E are uninterpretable. This is exacerbated by the fact that Ca²⁺ indicators heavily filter out higher frequencies of the underlying Ca²⁺ activity. These data need a serious rethink.

5. Fig. 5D: Neuronal Ca²⁺ activity in the bulk of tissue may tend to represent either axonal or dendritic activity depending on their relative volume fraction and on the pre- vs postsynaptic cell excitability. In any case, the present Ca²⁺ imaging method cannot distinguish individual spikes (see above), which severely limits any quantitative assessment.

6. Fig. 5F: These data require more detailed information about what was measured (amplitude, volume, duration, binary time series, etc.) to obtain correlation coefficients, and what statistical tests were used to compare them. The graph should probably show connected data points in individual animals.

Fig 7F: Ordinate units are not shown.

Author rebuttal, first version:

In this work by Bojarskaite et al, authors conducted a thorough and detailed characterization of astrocyte Ca²⁺ activity in un-anesthetized mice as they naturally cycle through wake-sleep states. This is a much needed and, overall, well conducted study that would be well received in the field. I am concerned, however, that this work might not be fully ripe for publication yet. While authors convincingly demonstrate their capability to reliably measure and quantify astrocyte calcium activity, along with EEG, EMG and

behavioral analysis, through naturally occurring states and state transitions in un-anesthetized mice (which is an achievement), the lack of clear questions and the lack of clear novel findings make this study resemble a “proof-of-concept” stage in its current form. This is apparent in the rather vague title or in that no major new findings are put forth, with the exception of the very last panel (Fig. 7F which authors hardly discuss). I am overall very supportive of this study, which I believe will be valuable to the field. But I believe authors need to re-arrange their manuscript and/or focus and further their analysis on some specific points and questions (see below) in order to deliver a few, key novel points rather than enumerate a list of features and changes, of which the physiological relevance is sometimes questionable (Fig. 6 C-E). All in all, while I believe this study will be very valuable to the field, I find difficult to fully support its publication in Nature Com until it provides a detailed account of a significant new finding that authors were able to elucidate through their newly developed analytical and “behavioral” method. I think this might be achievable rapidly without any major additional experimental work.

We are very pleased to see that Reviewer 1 is overall ‘very supportive of the study’ and sees that it would be very valuable and well received in the field. We have now performed new experiments, new analyses focusing on specific points, as well as revised and reorganized the manuscript according to the reviewers concerns which, we believe, clarified our novel findings. Please see specific answers addressed below.

Comment 1. The head-fixed/treadmill situation is, typically, rather uncomfortable for mice. Along with the many rounds of handling and “habituation” required, this makes such in vivo experiments relatively stressful for the animals. This raises several points.

- 1.1. First, that mice naturally fall asleep on the apparatus in these conditions seems quite extraordinary. As authors stated it, this has never been done before (that I’m aware of) and this is truly the main novelty their study brings. As such, many more details and necessary controls are required. I could not find in the methods or main text any details, tricks or procedures that authors used that permit mice to “naturally fall asleep” on the apparatus. Were they sleep deprived? Are recordings obtained at specific times of day, after mice have exhausted themselves running in their enriched environment? How many hours do mice spend, on average, head-fixed on the ball before they would fall asleep? These are key considerations, not only as far as reproducibility, but also because authors claim to study “natural sleep” here. How “natural” is mice sleep in a ball, in a head-fixed situation.

Response 1.1: We agree with reviewer 1 that aspects concerning how mice fall asleep being head-fixed could be dealt with more thoroughly in the manuscript. We have now expanded the METHODS part and included a new section “*Head-fixed sleeping protocol*” in the revised manuscript (page 20) describing the head-fixed sleep protocol in more detail: “*To assist sleep in a head-fixed position, we adjusted the running disc position to mimic the body’s natural position observed during unrestrained sleep*⁴⁸. We observed that locking the movement of the wheel once the mouse showed first signs of sleep, such as delta waves in ECoG and eyes closing, facilitated falling asleep. The imaging sessions of sleeping mice started at 9–10 AM (ZT 1–2), the beginning of light phase, and lasted until 3–6 PM (ZT 7–10). The mice did not have access to food

or water while sleeping under the microscope, however, in natural conditions mice feed almost exclusively during the dark phase, ZT 12–24^{49–51}. First signs of drowsiness, such as high delta power in the ECoG signal and eyes closing, were observed 15–45 min after head-fixation, and typically mice spent 90–120 min head-fixed under the microscope before falling asleep. Mice that did not show any signs of sleep within the first 2 hours of head-fixation were removed from the microscope. The mice had an exact replica of the microscope running disc in their home cage, and in our hands this made a large difference in aiding the mice to fall asleep (i.e. initially we tried various types of stages like a spherical treadmill and a tube for immobilization with little success). Mice were not sleep deprived or manipulated in any other way before imaging to induce sleep”. Two-photon imaging of mice sleeping in a head-fixed condition has been done before, but those studies focused on neurons and not astrocytes (Cox et al., 2016; Niethard et al., 2016a; Seibt et al., 2017). Also, regarding the reviewer’s sentence “How many hours do mice spend, on average, head-fixed on the ball before they would fall asleep?”, we would like to clarify that the mice are head-fixed on a flat freely spinning running disc and not a ball. We have updated Fig. 1a in the revised manuscript to better illustrate this (also please see part A of the figure below).

1.2. Second, if this study is set to eventually mark the field as I think it is, then authors should convincingly demonstrate that the sleep pattern and architecture on the apparatus are indeed identical to those that would be recorded in a freely behaving (freely sleeping) animal. Freely

behaving mice typically go through a stereotyped behavioral routine in preparation to sleep which, presumably, they are unable to carry out in a head-fixed situation. One would reasonably expect that this will alter their sleep and this should be quantified in this study. Authors could simply record EEG/EMG data from a few freely behaving mice in their home-cage, and then record from the same mice on the experimental head-fixed + treadmill apparatus. This longitudinal, paired paradigm would allow them to perform this key control experiment on a limited number of animals.

Response 1.2: We are pleased to see that reviewer 1 thinks our study will ‘mark the field’ and have now performed the comparison of sleep in head-fixed and freely behaving mice that the reviewer suggests. When comparing head-fixed and freely sleeping mice we find overall nearly identical sleep characteristics, except a few subtle differences. The parameter that is most different is the time it takes for the mice to fall asleep during the experimental conditions. This is now addressed in page 4, line 24 and Supplementary Fig. 2 of the revised manuscript: “To verify that the head-fixed situation did not perturb sleep we compared sleep characteristics between freely moving and head-fixed mice. We found nearly identical sleep characteristics between the two conditions, except a few subtle differences (Supplementary Fig. 2)”. Several other studies have compared unrestrained and head-fixed mice sleep characteristics and reported no significant differences in ECoG and sleep architecture in the two conditions (Lecci et al., 2017; Yüzgeç et al., 2018). Our head-fixed ECoG and sleep architecture results are also in agreement with other studies on head-fixed mice sleep (Cox et al., 2016; Lecci et al., 2017; Niethard et al., 2016a; Seibt et al., 2017; Yüzgeç et al., 2018).

Supplementary Figure 2. Comparison of sleep in head-fixed and freely behaving mice. (A) Sleep onset in freely behaving or head-fixed condition. (B) Average number of NREM, IS and REM sleep bouts per hour of sleep in freely behaving and head-fixed condition. (C) Percentage time spent in NREM, IS and REM sleep out of total sleep time in freely behaving and head-fixed condition. (D) Average NREM, IS and REM sleep bout duration in freely behaving and head-fixed condition. (E) ECoG power spectra of NREM, IS and REM sleep in freely behaving (free) and head-fixed (fixed) condition, normalized to the average total power in the 2–30 Hz frequency range during NREM sleep. Data represented as mean \pm SEM, for freely behaving: $n = 3$ mice, 6 days, 845 NREM episodes, 673 IS episodes, 102 REM episodes; for head-fixed: $n = 3$ mice, 6 days, 322 NREM episodes, 279 IS episodes, 35 REM episodes.

- 1.3. Third, consistently with the two points above, what authors describe as “quiet wakefulness” might in fact be freezing: anxiety-driven behavior. While quiet wakefulness is common between bouts of sleep, rarely do mice, in the awake state, spend more than a few seconds completely immobile without grooming or whisking. Figure 1B shows 2 full minutes of such behavior, flanked by two episodes of active locomotion – which resembles freezing rather than quiet wakefulness. I think authors need to address this point thoroughly (i.e. experimentally).

Response 1.3: We agree with the reviewer that mice rarely spend more than a few seconds completely immobile and that the long quiet wakefulness episode in the original manuscript Fig. 1b is not representative. We chose that particular trace to better illustrate the video-based behavioral classification. We have now chosen another trial better representing quiet wakefulness behavior (see Fig. 1b in the revised manuscript and figure below).

Below we have added a histogram of durations of quiet wakefulness periods, which in our experiments are strongly skewed towards short episodes: >90% of quiet wakefulness bouts were <10 sec.

Moreover, to check whether the particularly long bouts of quiet wakefulness affect the overall results, we have compared astrocytic Ca^{2+} event frequency in quiet wakefulness excluding quiet wakefulness bouts that are longer than 10 s with astrocytic Ca^{2+} event frequency including quiet wakefulness episodes of all durations and found no differences between the two conditions (see figure below).

We would also like to point out that our quiet wakefulness definition based on mouse movement in the video-footage also allows for minor movement as it is threshold based. One can see slight wheel and whisker movement during quiet wake periods in Fig. 1b and Supplementary Fig. 6 of the revised manuscript. Moreover, startle behavior, which is followed by freezing, is associated with endogenous norepinephrine release from noradrenergic fibers emanating from the locus coeruleus, which evokes global astrocytic Ca^{2+} events by acting on astrocyte $\alpha 1$ adrenoceptors (Bekar et al., 2008; Paukert et al., 2014). We do not observe global synchronized astrocytic Ca^{2+} events during our quiet wakefulness episodes as

supported by low Ca^{2+} event frequency throughout entire quiet wakefulness episodes (Fig. 2f, g and Supplementary Figure 6 in the revised manuscript) and by the absence of correlation between quiet wake episode duration and astrocytic Ca^{2+} event frequency (see figure below). This data supports that what we define as quiet wakefulness is not freezing behavior.

Comment 2. Unless I missed it, I believe the novel ROA algorithm requires similar validation. Overall, this new algorithm is reminiscent of the analytical framework developed by the Poskanzer lab (AQUA, published a few weeks ago in Nat Neuroscience), which has been available in open-source to the community and has already been widely adopted by most over the past few years. I think this is already widely considered as the new standard and I would worry to see too many such new analysis tools being published independently to the detriment of consistency and reproducibility. To address this point I think authors should try and validate their main findings using AQUA, and show that while both algorithms might give different quantitative results as far as calcium signal detection and features, the effects of sleep and state changes on the said Ca^{2+} activity remains the same with either algorithm.

Response 2. We agree with the reviewer that our findings would be strengthened if validated by another algorithm and have now analyzed our data with AQUA software. The effects of sleep and state changes on astrocytic Ca^{2+} activity remain the same with either algorithm (please see Supplementary Fig. 9 in the

revised manuscript and the figure below). The actual number of events detected was somewhat lower with AQuA than with the ROA method, but this is most likely due to the strong dependence on the in total about 40 parameters that are the inputs to the AQuA algorithm (as outlined in the "Parameter Guide" in instructions for use on github). The AQuA analysis is addressed in page 6, line 19 of the revised manuscript. "Importantly, our dataset yielded same trends when analyzed by another recently published astrocytic Ca^{2+} event analysis tool²³ (AQuA), but not when analyzed by conventional ROI analysis (Supplementary Figs. 9 and 10)."

Supplementary Figure 9. Ca^{2+} signaling in astrocytes across sleep-wake states analyzed by AQuA software. (A and B) Ca^{2+} signaling in astrocytes across sleep-wake states analyzed by AQuA (A) and ROA (B) methods. Data represented as estimates \pm SEM $n = 6$ mice, 278 trials, * $p < 0.05$, ** $p < 0.005$, * $p < 0.0005$. For details on statistical analyses, see Methods.**

Comment 3. Their study provides a wealth of rich and useful data and quantification, but going through the figures one by one, I felt that authors actually failed at putting forward any major new finding as mentioned above.

Response 3: We are pleased to see that the reviewer acknowledges the richness and usefulness of our data. The major new finding of this study is a thorough characterization of astrocytic Ca^{2+} signaling during natural sleep, and that astrocytic Ca^{2+} signals seem to be important for normal slow wave sleep. So far, astrocytic Ca^{2+} activity has been reported in anesthetized and awake animals, but not during natural sleep. Even though sleep and anesthesia have similar behavioral characteristic such as loss of consciousness, distinct physiological and behavioral differences clearly distinguish the two states. Unlike sleep, anesthesia is not reversible with external stimuli, is not regulated homeostatically and via circadian rhythms (Song et al., 2018; Tung and Mendelson, 2004), the electroencephalographic (EEG) pattern is devoid of distinct stages and of the cycling between stages, and there are no microarousals and awakenings. We acknowledge that the main take home messages in our study could have been more clearly presented and have now addressed this throughout the paper and in particular in the discussion. We have also emphasized this in the INTRODUCTION of the revised manuscript on page 2 line 24: "Astrocytic Ca^{2+} signals have been characterized under anesthetic conditions that have some resemblance to sleep^{4,5}. However, anesthesia is

fundamentally different from natural sleep because of clear distinct physiological and behavioral differences, such as lack of awakenings and microarousals, and lack of distinct stages seen in the electroencephalographic (EEG) signal and of cycling between stages.”

3.1 Figure 1 is a description of their experimental set-up, which might as well be a supplemental figure.

Response 3.1: We agree with the reviewer that Fig. 1 could be a supplemental figure, but as our experimental setup is quite complex and comprises six different behavioral states, we believe that Fig. 1 helps the reader to understand and to follow the paper.

3.2 Figure 2 shows that astrocyte Ca²⁺ activity is state dependent, which is already known, albeit not with that degree of details (and with anesthesia instead of true sleep).

Response 3.2: We agree with the reviewer that it has been demonstrated that astrocytic Ca²⁺ signaling is state dependent, but notably only during wakefulness and anesthesia. However, to the best of our knowledge, we are the first ones to report that astrocytic Ca²⁺ signaling is also sleep-state dependent. Sleep, however, comprises a completely different neurochemistry than wakefulness and anesthesia. Therefore, the RESULTS section describing these results in the original and revised manuscripts is “*Astrocytic Ca²⁺ signaling is reduced during sleep and is sleep state specific*”. We have also changed the title of Fig. 2 from “*Astrocytic Ca²⁺ signaling is brain state specific and reduced during sleep*” in the original manuscript to “*Astrocytic Ca²⁺ signaling is reduced during sleep and is sleep state specific*” in the revised manuscript.

3.3 Figure 3 shows that Ca²⁺ signaling is more frequent in the processes than in the soma, which is now a well-known fact in the field.

Response 3.3: We agree with the reviewer that it is already well documented that astrocytic Ca²⁺ signaling is more frequent in processes, but notably only during wakefulness and anesthesia. However, no such characterization has been performed in natural sleep, where mechanisms of triggering astroglial Ca²⁺ signals both in processes and somata could be quite different from wakefulness and anesthesia. We are, to the best of our knowledge, the first ones to report that astrocytic Ca²⁺ signaling is more frequent in processes during natural sleep. To make this finding clearer in the manuscript we have changed the RESULTS section title as well as Fig. 3 title from “*Astrocytic Ca²⁺ signals are most frequent in processes during all sleep-wake states*” in the original manuscript (page 7 line 5) to “*Astrocytic Ca²⁺ signals during sleep are most frequent in processes*” in the revised manuscript (page 7 line 1).

3.4 Figure 4 shows that astrocytes activity precedes switches in certain neuronal states, which was partially known from Kira Poskanzer’s work.

Response 3.4: We agree with the reviewer that certain clues into astrocytic activity preceding switches in certain neuronal states were reported in previous work by Kira Poskanzer. However, those studies were done in urethane anesthetized mice and not in naturally sleeping mice – the two conditions being fundamentally different as described above in response 3. Moreover, in Kira Poskanzer’s work on urethane anesthetized mice, astrocytic Ca²⁺ signals preceded shifts to the slow-oscillation dominated state, whereas

we show that astrocytic Ca^{2+} signals precede shifts from the slow-oscillation dominated state – slow wave sleep – to the fast-oscillation dominated wakefulness (Fig. 4 in the original manuscript).

3.5 Figure 5 shows that astrocytic and neuronal activity are somewhat disconnected during sleep while somewhat tuned to neuronal activity during wake states, which I believe is rather novel but the significance of this is not discussed and the experimental procedure here raises some concerns (see below).

Response 3.5: We have now discussed the significance of neuronal-astrocytic interplay findings in more detail in the DISCUSSION section in the revised manuscript (page 17, line 5): “*In recent studies, astrocytes have been established not only as passive supporters of neurons, but also as active participants in the bidirectional communication between the two cell types. Firing patterns of cortical neurons vary across sleep-wake states and are of higher frequency during wakefulness than sleep*”³⁸. We found that astrocytic Ca^{2+} signaling was somewhat synchronized to neuronal activity during wakefulness, whereas very little correlation was found during sleep. It is interesting to hypothesize that during states of higher neuronal activity (wakefulness), astrocytic Ca^{2+} signals might reflect increased metabolic demand and homeostatic burden, whereas during sleep when neurons are less active, astrocytes take a more independent role, where Ca^{2+} signals are driven by an intrinsic sleep state dependent factor rather than the activity of the neighboring neurons”. Concerns for the experimental procedure are answered below.

3.6 Figure 6 shows minor sleep architecture defects in IP3R2 KO mice, which are somewhat inconsistent with a publication from the Haydon lab (see below) and are rather marginal (with the exception of the increased spindles which authors hardly discuss or interpret).

Response 3.6: We agree that our data are somewhat inconsistent with the data from the Haydon group. There are a number of potential explanations why the results are not identical in these two studies. Firstly, Haydon et al. used a different mouse model than what we have used. Even though both models have attenuated Ca^{2+} signaling in astrocytes, the underlying mechanisms are different. Secondly, Haydon et al. identified only two different sleep states, namely SWS and REM sleep, whereas we subdivided SWS into NREM and IS, the latter being a transitory phase from NREM to REM which occurs at the end of NREM episode (Seibt et al., 2017).

To strengthen our findings regarding sleep architecture and ECoG differences in IP3R2KO mice, we have now compared sleep between WT and IP3R2KOs also in freely behaving mice and extended sleep analyses by comparing number of state transitions, microarousals and awakenings between the genotypes and correlating these transitions with astrocytic Ca^{2+} signals (please see figure below and Fig. 8 in the revised manuscript). The findings in unrestrained mice are in line with our initial findings in head-fixed mice (Fig. 6 in the original manuscript and Fig. 8 in the revised manuscript).

We have included the updated and more thorough analyses on the differences in sleep between the genotypes in the re-structured RESULTS section paragraph on IP3R2KO data “*Astrocytic IP_3 -mediated Ca^{2+} signaling regulates slow-wave sleep architecture and brain rhythms*” in the revised manuscript, page 11, line 1: “*Our observations of astrocytes during sleep revealed a spatiotemporal specificity of Ca^{2+}*

activity, particularly during SWS, that could indicate causal roles for astrocytic signaling in sleep regulation. To identify these roles of astrocytic Ca^{2+} signals in sleep, we employed the *Itp2*^{-/-} mouse model, in which astrocytic Ca^{2+} signaling is strongly attenuated, but not abolished (as shown in experiments on awake and anesthetized mice)¹⁷.

In agreement with previous reports, we found that *Itp2*^{-/-} mice exhibited reduced astrocytic Ca^{2+} signaling in all states of wakefulness as measured by ROA frequency and active voxels (x-y-t) (Supplementary Fig. 15a, b)¹⁷. However, astrocytic Ca^{2+} activity measured by both the percentage of active voxels and ROA frequency did not significantly differ between WT and *Itp2*^{-/-} mice during sleep (Fig. 7b, c). Even so, *Itp2*^{-/-} mice exhibited Ca^{2+} signals with disrupted spatiotemporal features – namely, longer duration and smaller spatial extent (Fig. 7a, d, e). This finding was observed in all states of wakefulness, but during sleep was restricted to SWS (NREM and IS states).

Next, we investigated whether IP_3 -dependent astrocytic Ca^{2+} signaling had any effect on sleep dynamics. We compared sleep architecture and spectral ECoG properties between the two genotypes and found that *Itp2*^{-/-} mice exhibited more frequent NREM and IS bouts that were of shorter duration than in the WT mice (Fig. 8a, b). More fragmented SWS sleep could be a consequence of more frequent microarousals (short wakefulness intrusions characterized by a reduction of low-frequency ECoG power) and awakenings which interrupt the sleep states, or more frequent NREM-to-IS and IS-to-NREM transitions. The number of awakenings did not differ between the genotypes in any of the sleep states (Fig. 8d). However, we found that *Itp2*^{-/-} mice have ~20 more microarousals per hour than WT mice (Fig. 8f). Such microarousals were associated with abrupt increases in astrocytic Ca^{2+} signaling in WT mice, whereas no such response was observed in *Itp2*^{-/-} mice (Fig. 8g). Surprisingly, *Itp2*^{-/-} mice were completely devoid of the prominent astrocytic Ca^{2+} increases seen upon awakenings in WT mice (Fig. 8e).

NREM-to-IS as well as IS-to-NREM transitions were more frequent in *Itp2*^{-/-} mice compared to WT mice (Fig. 8h), indicating abnormal SWS state dynamics in the knockouts. Interestingly, in WT mice, NREM-to-IS transitions were preceded by a decrease in Ca^{2+} signaling, whereas IS-to-NREM transitions were followed by an increase in astrocytic Ca^{2+} signaling. This was not the case in *Itp2*^{-/-} mice (Fig. 8i), and it is tempting to hypothesize that IP_3 mediated astrocytic Ca^{2+} signaling is important to sustain uninterrupted SWS by regulating NREM and IS state transitioning and possibly preventing microarousals.

Finally, we assessed the spectral ECoG properties of NREM, IS and REM sleep between the genotypes. We detected a decrease in delta power during NREM sleep, an increase in theta during REM sleep and an increase in sigma power during IS sleep in *Itp2*^{-/-} mice (Fig. 8j). ECoG activity in the sigma frequency range is indicative of sleep spindles – bursts of neuronal activity linked to memory consolidation²⁷. As *Itp2*^{-/-} mice displayed higher sigma power in IS, we next evaluated the occurrence of sleep spindles in WT and *Itp2*^{-/-} mice (Fig. 8k). The frequency of sleep spindles in IS sleep was indeed almost twice as high in *Itp2*^{-/-} mice than in WT mice (Fig. 8l). Intriguingly, sleep spindles were followed by an IP_3 -dependent increase in astrocytic Ca^{2+} signals (Fig. 8m). These data indicate a role for astrocytic IP_3 -mediated Ca^{2+} signaling pathway in regulating the architecture and brain rhythms of slow wave sleep”.

We have also included a more detailed discussion of the sleep spindle findings in the DISCUSSION part of the revised manuscript (page 15, line 3): *“IS sleep in $Itr2^{-/-}$ mice was associated with increased sigma ECoG power and twice as many sleep spindles, compared to WT mice. Similar to microarousals and SWS state transitions, sleep spindles in WT mice were associated with particular Ca^{2+} signaling pattern, which was not observed in $Itr2^{-/-}$ mice. Although sleep spindles are known to be important for learning and memory³², too many sleep spindles could be maladaptive. Excessive sleep spindles have been observed in humans with learning disabilities and have been shown to predict poor avoidance performance in rats^{33–35}. Our data indicates that the IP3-mediated astrocytic Ca^{2+} signaling plays a key role in stabilizing and maintaining uninterrupted SWS and regulating SWS brain rhythms critical for learning and memory”*.

Fig. 8. Astrocytic IP_3 -mediated Ca^{2+} signaling pathway regulates slow wave sleep. (A to C) Bout duration (A), mean number of bouts per hour (B), and percentage recording time (C) of NREM, IS and REM sleep states in WT and *Itpr2*^{-/-} mice. For WT mice: n = 4 mice, 806 NREM episodes, 617 IS episodes, 92 REM

episodes; for *Itp2^{-/-}* mice: *n* = 3 mice, 1140 NREM episodes, 787 IS episodes, 86 REM episodes. (D) Awakening probability calculated as a number of awakenings from NREM, IS or REM sleep divided by total number of NREM, IS or REM episodes, respectively, in WT and *Itp2^{-/-}* mice. For WT mice: *n* = 4 mice, 806 NREM episodes, 617 IS episodes, 92 REM episodes; for *Itp2^{-/-}* mice: *n* = 3 mice, 1140 NREM episodes, 787 IS episodes, 86 REM episodes. (E) Mean time-course of astrocytic Ca²⁺ signals during transitions from NREM, IS or REM sleep to wakefulness in WT and *Itp2^{-/-}* mice. For WT mice: *n* = 6 mice, 59 NREM awakenings, 141 IS awakenings, 87 REM awakenings; for *Itp2^{-/-}* mice: *n* = 6 mice, 23 NREM awakenings, 56 IS awakenings, 27 REM awakenings (F) Frequency of microarousals during NREM sleep in WT and *Itp2^{-/-}* mice. For WT mice: *n* = 6 mice, 355 microarousals, for *Itp2^{-/-}* mice: *n* = 6 mice, 240 microarousals. (G) Mean time-course of (top to bottom) astrocytic Ca²⁺ signals and z-scores of EMG activation and whisker motion aligned to the beginning of a microarousal in WT and *Itp2^{-/-}* mice. For WT mice: *n* = 6 mice, 303 microarousals, for *Itp2^{-/-}* mice: *n* = 5 mice, 191 microarousals. (H) Number of NREM-to-IS, IS-to-NREM and IS-to-REM transitions in WT and *Itp2^{-/-}* mice. For WT mice: *n* = 4 mice, 386 NREM-to-IS, 187 IS-to-NREM, 68 IS-to-REM transitions; for *Itp2^{-/-}* mice: *n* = 3 mice, 440 NREM-to-IS, 267 IS-to-NREM, 61 IS-to-REM transitions. (I) Mean time-course of astrocytic Ca²⁺ signals during NREM-to-IS, IS-to-NREM and IS-to-REM transitions in WT and *Itp2^{-/-}* mice. For WT mice: *n* = 6 mice, 90 NREM-to-IS, 35 IS-to-NREM, 25 IS-to-REM transitions; for *Itp2^{-/-}* mice: *n* = 6 mice, 62 NREM-to-IS, 14 IS-to-NREM, 8 IS-to-REM transitions (J) Mean ECoG power of (left to right) delta (0.5–4 Hz) frequency during NREM sleep, theta (5–9 Hz) frequency during REM sleep, and sigma (10–16 Hz) frequency during IS sleep, normalized to 0.5–30 Hz total power. For WT mice: *n* = 4 mice, 785 NREM episodes, 668 IS episodes, 102 REM episodes; for *Itp2^{-/-}* mice: *n* = 3 mice, 957 NREM episodes, 648 IS episodes, 74 REM episodes. (K) Representative example of a sleep spindle in ECoG trace and power spectrogram. (L) Mean number of spindles per minute during IS sleep in WT and *Itp2^{-/-}* mice. For WT mice: *n* = 3 mice, 516 IS episodes; for *Itp2^{-/-}* mice: *n* = 3 mice, 560 IS episodes. (M) Mean time-course of (top to bottom) astrocytic Ca²⁺ signals, whisker movement z-score and ECoG power spectrograms aligned to the beginning of a sleep spindle. For WT mice: *n* = 6 mice, 243 sleep spindles; for *Itp2^{-/-}* mice: *n* = 6 mice, 93 sleep spindles. All values represent estimates ± SEM, **p* < 0.05, ***p* < 0.005, ****p* < 0.0005. For details on statistical analyses, see Methods.

3.7 To me, the most stunning finding, which only comes in Figure 7F, is that deleting IP₃R2 in astrocytes prevents phase transition surges in Ca²⁺ activity in astrocytes all together. This is very intriguing but, here again, authors provide little clues regarding the meaningfulness of such observation.

Response 3.7: We thank the reviewer for pointing out the novelty of this observation. As mentioned in reviewer's comment 5 and our response 5 below (page 14), it is highly likely that astrocytic Ca²⁺ signals upon awakening are triggered by norepinephrine released from locus coeruleus. Norepinephrine activates astrocytic Gq-coupled α₁ adrenoceptors which trigger Ca²⁺ release from endoplasmic reticulum via IP₃R2. In that case, the absence of Ca²⁺ surges in *Itp2^{-/-}* mice upon awakenings could be explained by lack of IP₃R2 in these mice. We have addressed this in the DISCUSSION of revised manuscript (page 16 line 6): "In our experiments, astrocytic Ca²⁺ signaling typically preceded awakenings from SWS (both NREM and IS sleep), but not from REM sleep. This finding is consistent with the temporal profile of cortical NE release

from locus coeruleus (LC) neurons upon awakening. The firing of LC noradrenergic (NA) neurons upon SWS-to-wake transitions has been shown to precede the onset of EEG activation by 0.3–1 s, whereas this predictive activity was not seen for REM sleep to wake transitions²⁶, suggesting that astrocytic Ca^{2+} signals upon awakening are triggered by norepinephrine. Norepinephrine acts on Gq-coupled $\alpha 1$ -adrenergic receptors on astrocytes, leading to Ca^{2+} release from the endoplasmic reticulum through IP_3R2 ^{12,17,36}. This might explain the absence of Ca^{2+} surges upon awakening in the $Itpr2^{-/-}$ mice which lack IP_3R2 (Fig. 8e)”. To try to elucidate the consequences of absent phase transition Ca^{2+} surges in $Itpr2^{-/-}$ mice (Fig. 7f in the original manuscript), we have now quantified the number of sleep-to-wake state transitions in the two genotypes in freely behaving mice (Fig. 8e in the revised manuscript, also see figure above on page 11). We did not detect any differences between the genotypes in number of awakenings from neither of the sleep states.

3.8 In addition, the idea that astrocytes “modulate” sleep has emerged in the field nearly a decade ago now and, in fact, the use of the term “modulate” by authors is indicative of a lack additional insights provided by their study on that matter.

Response 3.8: We agree with the reviewer. After additional experiments in freely behaving mice, additional analyses (please see response 3.6 on page 9 and the figure above on page 11) and careful revision of the manuscript, we have now crystalized our main findings to “astrocytic Ca^{2+} signal reduction during natural sleep and the importance of astrocytic Ca^{2+} signaling for normal slow wave sleep”. This topic is now the common thread throughout the manuscript:

- A. The title was changed from “ Ca^{2+} signaling in mouse astrocytes is sleep-wake state specific and modulates sleep” in the original manuscript to “ Ca^{2+} signaling in astrocytes is reduced during sleep and is involved in the regulation of slow wave sleep” in the revised manuscript;
- B. We re-phrased the end of the ABSTRACT from “Genetic ablation of a key astrocytic Ca^{2+} signaling pathway resulted in fragmentation of slow-wave sleep, yet increased the frequency of sleep spindles. Our findings demonstrate an essential role of astrocytic Ca^{2+} signaling in modulating sleep” in the original manuscript (page 2 lines 9–11) to “Genetic ablation of an important astrocytic Ca^{2+} signaling pathway impaired slow wave sleep and resulted in an increased number of microarousals, abnormal brain rhythms, and an increased frequency of slow wave sleep state transitions and sleep spindles.” in the revised manuscript (page 2 line 9)
- C. We re-phrased the end of INTRODUCTION from “Finally, we demonstrate that the inositol triphosphate (IP_3)-mediated astrocytic Ca^{2+} signaling pathway modulates sleep states, brain rhythms and sleep spindle dynamics. Taken together, our data indicate an essential role for astrocytic Ca^{2+} signaling in modulating sleep” in the original manuscript (page 3 lines 17–20) to “Finally, we have demonstrated that the inositol triphosphate (IP_3)-mediated astrocytic Ca^{2+} signaling regulates SWS by maintaining uninterrupted SWS, and affecting SWS state dynamics and sleep spindles. Taken together, our data indicate a role for astrocytic Ca^{2+} signaling in regulating SWS” in the revised manuscript (page 3 line 18).
- D. We re-phrased the beginning of the DISCUSSION from “Genetic ablation of a key astrocytic Ca^{2+} signaling pathway altered sleep architecture and resulted in fragmentation of SWS, yet increased the frequency of sleep spindles. Our data show that astrocytes are essential for normal sleep through

mechanisms involving intracellular Ca²⁺ signals” in the original manuscript (page 13 line 2) to “Genetic ablation of IP₃R2, an important astrocytic Ca²⁺ signaling pathway, led to abnormal sleep architecture, state dynamics and brain rhythms of slow wave sleep. Taken together our data show that astrocytes are essential for normal slow wave sleep through mechanisms involving intracellular Ca²⁺ signals” in the revised manuscript (page 13 line 2).

- E. In general, we replaced the term “*modulates sleep*” with “*regulates slow wave sleep*” throughout the revised manuscript.

Comment 4. It is my opinion that authors’ decision and justification to exclude micro-arousal episodes interrupting SWS (page 4) is unfounded and rather detrimental. On page 4 they justify their decision by stating “we aimed to characterize true sleep astrocyte Ca⁺ signals”. That’s rather unconvincing and on page 6, authors explain that they “investigated astrocyte Ca²⁺ signaling across the different states of wakefulness” which is in direct contradiction with the above justification. More importantly, there are very obvious Ca²⁺ signals restricted to these micro-arousal states that remind of sleep to wake transitions (Sup Fig. 4). I think this analysis of these states should be included in their study.

Response 4. We thank the reviewer for this observant comment. We would like to clarify that we have made a mistake and mis-labelled what are actually “brief awakenings” (3–5 s) as “microarousals”. What we have excluded from the analyses were brief awakenings (now marked in the revised manuscript Supplementary Fig. 7 with a symbol “#”), and not microarousals (now marked in the revised manuscript Supplementary Figure 7 with a single asterisk “*”). We have clarified this confusion in the revised manuscript by removing the sentence “*We excluded microarousals in SWS since we aimed to characterize true sleep astrocytic Ca²⁺ signals*”. We agree with the reviewer that Ca²⁺ signals during microarousals are interesting to analyze. Therefore, we have now analyzed astrocytic Ca²⁺ responses during microarousals in WT and *Itr2*^{-/-} mice. This is addressed in Fig. 8f, g and page 11 line 25 in the revised manuscript: “*However, we found that Itr2^{-/-} mice have ~20 more microarousals per hour than WT mice (Fig. 8f). Such microarousals were associated with abrupt increases in astrocytic Ca²⁺ signaling in WT mice, whereas no such response was observed in Itr2^{-/-} mice (Fig. 8g).*”

Comment 5. It has become very clear over the recent years that astrocytes respond to neuromodulators, and might be part of most neuromodulatory circuits; see Ma, Nature 2016, Paukert, Neuron 2014, Papouin, Neuron 2017; Mu, Cell 2019 and the work from the Nedergaard group – most of which are actually cited in the present manuscript. Therefore, I am a little surprised that authors provide little mechanistic insights into what might cause astrocyte Ca²⁺ activity to precede “neuronal awakening” in S↔W transitions, among other observations. It is easy to speculate that astrocytes might simply be responding to NE release and this would be relatively easy to investigate with tools currently available. At the very least, authors should discuss more thoroughly this point and situate their results more clearly in the context of these recent discoveries.

Response 5. We agree with the reviewer. These aspects are now addressed more thoroughly in the DISCUSSION, page 16, line 6: “*In our experiments, astrocytic Ca²⁺ signaling typically preceded awakenings from SWS (both NREM and IS sleep), but not from REM sleep. This finding is consistent with*

the temporal profile of cortical NE release from locus coeruleus (LC) neurons upon awakening. The firing of LC noradrenergic (NA) neurons upon SWS-to-wake transitions has been shown to precede the onset of EEG activation by 0.3–1 s, whereas this predictive activity was not seen for REM sleep to wake transitions (Takahashi et al., 2010), suggesting that astrocytic Ca²⁺ signals upon awakening are triggered by norepinephrine”

Comment 6. Page 8 and Figure 4. Evidently, these results bring the nearly philosophical question of what is awakening? A change in the EEG/EMG trace? A change in neuronal activity? A change in behavior? Or could astrocytes ‘awakening’ be a new marker of state transition? (I would personally welcome such discussion in the manuscript.) However, authors need to be careful with their interpretation here and clarify their overall statements. That astrocyte calcium activity precedes awakening in 60% of NREM→wake transitions means it does not 40% of the time. If this is correct, this means that on average, astrocyte calcium activity is overall synchronous to the state transition. In this case, authors’ current statement would be a biased exaggeration. In addition, assuming that astrocytes Ca²⁺ activity in these particular transitions is the result of NE release, then one would expect to see a rostro-caudal ‘wave’ of activation in response to the release of NE by NE-releasing LC-neuronal projections through the cortex. This could be easily tested with some NE receptor (alpha1a) antagonists or ablation. This would also explain why some Ca²⁺ seem to precede awakening while others do not and authors should check if the timing of these Ca²⁺ events is spatially determined. If this is the case, then these astrocyte Ca²⁺ events are one of the first steps in the awakening process but I am unsure whether they could be described as “preceding” the wake state.

Response 6. We agree that the question of what really comprises an awakening is intriguing. Observing astrocytic activity often even preceding the classical definition of the transition to wakefulness (Takahashi et al., 2006, 2009, 2010), should maybe prompt us to revisit the definition of awakenings. We have included a sentence addressing this topic in the DISCUSSION of the revised manuscript (page 16 line 20): “*These findings pose the intriguing question of what comprises an awakening and whether astrocytic ‘awakening’ could be a new marker of the transition to wakefulness.*”

We acknowledge that stating that astrocytic Ca²⁺ signals precede awakenings could be seen as an exaggeration as this is not the case in all transitions. A more nuanced wording is now adopted throughout the revised manuscript:

- 1) RESULTS section, page 9, line 15: “*However, to our surprise, astrocytic Ca²⁺ signals typically preceded the awakenings from SWS (NREM and IS sleep states)*”.
- 2) DISCUSSION section, page 16, line 6: “*In our experiments, astrocytic Ca²⁺ signaling typically preceded awakenings from SWS (both NREM and IS sleep), but not from REM sleep*”.

The reason for this variance in Ca²⁺ onset time is not entirely clear, and as we show in the supplementary data (Supplementary Fig. 13; revised manuscript), there is a dependency on the depth of the preceding NREM sleep, suggesting that the prevailing neurochemistry of the brain tissue is important for how astrocytes respond to their putative triggers. We have now addressed this more cautiously in the

DISCUSSION part on page 16 line 16 in the revised manuscript: *“It is important to mention that astrocytic Ca^{2+} signals did not precede all SWS-to-wake transitions (Fig. 5a, e). The reason for this variance in Ca^{2+} onset time is not entirely clear, and as we show in Supplementary Fig. 13, there is some dependency on the depth of the preceding NREM sleep, suggesting that the prevailing neurochemistry of the brain tissue is important for how astrocytes respond to their triggers.”*

We were not aware of – and have not been able to uncover – publications demonstrating that norepinephrine from locus coeruleus is released in a wave-like rostro-caudal manner, even though the norepinephric fibers follow such an anatomical distribution. However, if this is the case, we would still not expect this factor to explain the differences in timing we observed, as our imaging region is carefully placed over the same region in barrel cortex every time, and that the internal relation between the ECoG electrodes and imaging region is similar in all mice. Similarly, if astrocytic Ca^{2+} events would follow a wave-like release in a rostro-caudal fashion, we would expect to see the same temporal organization in REM sleep, which is not the case (Fig. 5 in the revised manuscript).

Comment 7.

7.1 Page 9 and Figure 5. Employing calcium sensors as a proxy of neuronal signaling is a rather biased approach over such prolonged periods of time because it does not faithfully reflect neuronal firing across various states (which is really what matters here). Neuronal firing mode changes across the 24 hrs period and this will affect Ca^{2+} readouts in a non-linear manner. In other words, Ca^{2+} imaging might reflect more closely neuronal firing during awake states than it does during sleep states. That alone would explain the results presented in figure 5. Therefore, I believe these results should be interpreted more cautiously and might require additional supportive data that are not based on jRGECO fluorescence.

Response 7.1: We thank the reviewer for the insightful comments. We agree that neuronal data obtained from Ca^{2+} sensors should be interpreted more cautiously. We have now addressed the points brought up by the reviewer in the DISCUSSION of the revised manuscript page 17 line 15: *“Neuronal activity data obtained from fluorescent Ca^{2+} sensors as a proxy for neuronal firing should be interpreted with caution. Neuronal firing patterns are at least partially state-dependent (Miyawaki et al., 2019) indicating that due to the sensitivity and kinetics of the sensor, Ca^{2+} readouts could be affected in a non-linear manner, hence not faithfully reflecting neuronal firing across various states. Moreover, as the neuronal Ca^{2+} event detection is threshold-based, it is likely that neuronal activity is only detected above a certain spiking rate, possibly contributing to an underestimation of neuronal activity levels. Even so, our neuronal Ca^{2+} data across sleep-wake states (Fig. 6b) is in line with other Ca^{2+} imaging and electrophysiological studies showing reduced neuronal firing during sleep compared to wakefulness (Kanda et al., 2017; Niethard et al., 2016a). Future electrophysiological studies using unit recordings will have to confirm our observations”*. It is important to mention that Ca^{2+} imaging is now routinely used to investigate neuronal activity due to their technical advantages (Ali and Kwan, 2019; Dana et al., 2016; Peron et al., 2015), and, even though not to a single action potential resolution, still reflect neuronal activity.

7.2 In addition, authors used hand-drawn ROI to analyses neuronal activity, a method they reasonably criticize in the manuscript, and compare it to ROA-based astrocytic activity.

Response 7.2: Conventional ROI analyses are insufficient for astrocytic Ca^{2+} signals because of the highly complex and spatiotemporally dynamic nature of astrocytic Ca^{2+} signaling (as discussed in the original manuscript page 5 lines 1–19 and Supplementary Figure 5). However, for neurons, Ca^{2+} increases reflect the binary nature of action potentials and appear homogenous within the somata, due to the relatively slow kinetics of jRGECO/acquisition rate. Therefore, we believe that ROI analyses are sufficient to investigate neuronal somata Ca^{2+} signals.

7.3 Finally, neuronal activity (at least seen with GECO) is dramatically reduced during sleep such that the apparent lack of synchrony/relationship between neuronal activity and astrocytic activity during sleep states could be due to under-sampling.

Response 7.3: We would like to clarify that Fig. 5f in the original manuscript describes the correlation between astrocytic Ca^{2+} signals in neuropil (detected by the ROA method) and neuronal Ca^{2+} signals in neuropil, and not the neuronal somata Ca^{2+} signals. We have now changed the title of that particular figure part from “*Neuronal Ca^{2+} vs. astrocytic Ca^{2+}* ” in the original manuscript (Fig. 5f) to “*Neuronal Ca^{2+} vs. astrocytic Ca^{2+} in neuropil*” in the revised manuscript (Fig. 6e). As neuropil Ca^{2+} signaling increases during sleep compared to wakefulness (Fig. 5e in the original manuscript), reduced correlation between neuronal activity and astrocytic activity during sleep states could not be due to undersampling. Moreover, to illustrate reduced astrocyte-neuron correlation during sleep compared to wakefulness, we have now also included examples of astrocytic ROA frequency traces aligned to neuropil Ca^{2+} trace in Fig. 6d in the revised manuscript. Please also see response to the comment 2 of reviewer 2 below.

7.4 Overall, these possible caveats make the data presented in this figure rather unconvincing. This is added to the fact that the significance and meaning of these finding is not discussed.

Response 7.4: We have addressed the significance and meaning of these findings in the DISCUSSION of the revised manuscript (page 17, line 5): “*In recent studies, astrocytes have been established not only as passive supporters of neurons, but also as active participants in the bidirectional communication between the two cell types. Firing patterns of cortical neurons vary across sleep-wake states and are of higher frequency during wakefulness than sleep (Kanda et al., 2017). We found that astrocytic Ca^{2+} signaling was somewhat synchronized to neuronal activity during wakefulness, whereas very little correlation was found during sleep. It is interesting to hypothesize that during states of higher neuronal activity (wakefulness), astrocytic Ca^{2+} signals might reflect increased metabolic demand and homeostatic burden, whereas during sleep when neurons are less active, astrocytes take a more independent role, where Ca^{2+} signals are driven by an intrinsic sleep state dependent factor rather than the activity of the neighboring neurons*”.

We have also removed two panels from the original manuscript (Fig. 5d, e in the original manuscript) that we agree were difficult to interpret having both technical and physiological caveats in mind and included additional raw data traces (Fig. 6d in the revised manuscript) to aid in interpreting the correlational data.

Comment 8. The first sentence of the abstract is an inaccurate overstatement. Similarly, the description of astrocytes as "housekeepers of the brain" in the third sentence of the introduction is quite shocking. Not only is this an outdated view. It is also incorrect.

Response 8: The first sentence of the ABSTRACT in the original manuscript was "*Astrocytic Ca²⁺ signaling has been intensively studied in health and disease but remains uncharacterized in sleep*". It is not entirely clear which part of the sentence is an inaccurate overstatement. We do believe that astrocytic Ca²⁺ signalling has been intensively studied in health and disease as shown by numerous publications. We guess the reviewer alludes to: "*but remains uncharacterized in sleep*". Even though the role of astrocytic Ca²⁺ signals in sleep has been investigated in Foley et al., no publications including Foley et al., to the best of our knowledge, have shown or quantified Ca²⁺ signals in astrocytes during sleep. Therefore, in essence, we believe the statement to be correct. For clarification purposes we have now changed the first sentence of the ABSTRACT in the revised manuscript to "*Astrocytic Ca²⁺ signaling has been intensively studied in health and disease but has not been quantified in natural sleep*" (page 2, line 2). We agree with the reviewer that astrocytes are more than housekeepers, however, their higher order functions do not eliminate their homeostatic "housekeeper" functions. For clarification purposes we have rephrased the beginning of INTRODUCTION from "*Still, the nightlife of the housekeepers of the brain – the astrocytes – is poorly characterized*" (page 2, line 15; original manuscript) to "*Recent studies show that not only neurons, but also glial cells are essential for sleep. Still, the nightlife of the main glial cell in the brain – the astrocytes – is poorly characterized*" in the revised manuscript (page 2, line 15).

Comment 9. Immunolabeling images of Sup Fig.1 come across as 'cherry-picked'. To be convincing, authors need to provide a wider view and an accurate quantification of the cell-specificity. Also, what exact brain region (cortical layer) is this?

Response 9: We thank the reviewer for the thorough evaluation of our manuscript. We have now included micrographs of larger fields-of-view in the revised manuscript, Supplementary Fig. 1 (please see figure below and Supplementary Fig. 1 in the Supplementary Information of the revised manuscript, page 2). We have evaluated the cell-specificity and found sparse astrocytic jRGECO1a labeling, but only in layer I (all our imaging was done in layer II/III) (please see figure below, part A, white arrowheads; and Supplementary Fig. 1a of the revised manuscript). We also found sparse (<1%) neuronal labeling with anti-GFP in layer II/III (please see figure below, part B, white arrowhead; and Supplementary Fig. 1b of the revised manuscript). All our imaging was done in barrel cortex, layer II/III. We have included this information into the legend of Supplementary Fig. 1 in the revised manuscript (please see figure legend below and Supplementary Fig. 1 legend in revised manuscript).

Supplementary Figure 1. Selectivity of the expression of GFAP-GCaMP6f in astrocytes and SYN-jRGECO1a in neurons in barrel cortex. (A) Immunolabeling of green fluorescent protein (anti-GFP) and glial fibrillary acidic protein (anti-GFAP), as well as jRGECO1a fluorescence demonstrate that astrocytes, and not neurons, were labeled with GCaMP6f. No astrogliosis is observed in the injected hemisphere, as compared to the contralateral non-injected hemisphere. Sparse astrocytic jRGECO1a labeling was noted (white arrowheads), but only in layer I. (B) Immunolabeling with anti-GFP and anti-NeuN, and fluorescence signal of jRGECO1a show that neurons, and not astrocytes, in the virus injection site were labeled with jRGECO1a. Sparse (<1%) neuronal anti-GFP labeling was noted in layer II/III (yellow arrowhead). (C) Immunolabeling with anti-GFP and the microglial marker Iba1 (anti-Iba1), as well as jRGECO1a fluorescence, demonstrate that injection of GCaMP6f and jRGECO1a did not induce microglial activation. All imaging was done in layer II/III. Scale bars 50 μm .

Comment 10. Page 7. I find interesting authors' study of the stability (or lack thereof) of Ca^{2+} hotspots over state transitions. It is unclear however whether hotspots are stable in a given state over transitions? In other words, are hotspots associated with a given state consistently active during similar states, beyond the many states transitions? i.e. are there wakefulness-specific hotspots and SWS-specific hotspots? What is the significance of this? Maybe this could be extended to calcium signals other than the top 5% active ROAs.

Response 10: We are pleased to see that the reviewer found the hotspot analysis interesting. We have now quantified whether areas of Ca^{2+} activity are associated with a given state and are consistently active during different episodes of the same brain state. This is addressed in a new RESULTS section "Astrocytic Ca^{2+} signals exhibit state-specific locations" on page 8 line 3, starting on page 8 line 3, and Fig. 4 (also see figure below) of the revised manuscript: "Next, we studied whether any Ca^{2+} signals in astrocytes occurred at sleep-wake state specific locations. For each sub-state of sleep and wakefulness, we created masks outlining active areas, and calculated the overlap between these areas within or across states (Fig. 4b). Within each state, the overlap of active areas was low except during locomotion, where most of the FOV was active (Fig. 4c, left.). Still, for all sleep-wake states except REM sleep, the overlap between active areas was significantly higher than chance (Fig. 4c, left; and Supplementary Fig. 12, left). We then compared whether certain locations of activity were specific to sleep or wakefulness by calculating the overlap between different states. Interestingly, we found that overlap is significantly higher than chance between the different wakefulness states (Loc-Wh, Wh-QW, Loc-QW) and between the states of slow wave sleep (IS-NREM) (Fig. 4c, right; and Supplementary Fig. 12, right). However, locations of activity did not significantly overlap between slow wave sleep states and REM sleep (IS-REM, NREM-REM) (Fig. 4c, right; and Supplementary Fig. 12, right). Taken together, these data suggest the presence of state-specific locations of astrocytic Ca^{2+} signals."

Fig. 4. Astrocytic Ca^{2+} signals exhibit state-specific locations. (A) Representative astrocytic Ca^{2+} activity maps outlining areas active during episode 1 (green) and episode 2 (blue) of each of the sleep-wake states, and the overlap between the active areas (red). (B) Average overlap between the active areas of two episodes of same sleep-wake state (left) and of different sleep-wake states (right) compared to chance, calculated as the overlap between two episodes of same sleep-wake state from different fields-of-view. Statistical significance was computed with a randomization test, see Supplementary Fig. 12 and Methods. Data represented as estimates \pm SEM, $n = 6$ mice, 195 unique FOVs, 2944 locomotion-locomotion, 8106 whisking-whisking, 3077 QW-QW, 672 NREM-NREM, 644 IS-IS, 11 REM-REM, 1315 IS-NREM, 336 IS-REM, 4243 locomotion-QW, 7532 locomotion-whisking, 340 NREM-REM, 9739 QW-whisking

comparisons. For details on statistical analyses, see Methods. Loc – locomotion, Wh – whisking, QW – quiet wakefulness.

Comment 11. The conclusion at the end of the top paragraph of page 10 is unwarranted or at least misleading (“indicating that astrocytic Ca²⁺ is uncoupled from local neuronal activity during brain states that lack dense neuronal discharge”). There is no indication of any sort that neuronal firing per se is a determining factor in the results of figure 5. In fact, in view of the most recent discoveries in the field, the disconnect between astrocytic and neuronal activity during sleep could be the result of neuromodulation acting on astrocytes to promote SWS neuronal activity (see work from Nedergaard lab). In this case, the “lack dense neuronal discharge” is mechanistically irrelevant.

Response 11: We agree with the reviewer and have now rephrased the sentence “*indicating that astrocytic Ca²⁺ is uncoupled from local neuronal activity during brain states that lack dense neuronal discharge*” in the original manuscript (page 10, lines 7–9) to “*Similar to the temporal relationship with the neuronal somata, astrocytic Ca²⁺ signals displayed a modest correlation with neuronal signals in neuropil during wakefulness, but were significantly decorrelated during the sleep states (Fig. 6e)*” in the revised manuscript (page 10, line 19). We were not aware of – and have not been able to uncover – publications from Nedergaards lab presenting data on astrocyte-neuron communication during sleep.

Comment 12. Page 11. Line 6. I assume authors meant to refer to reference #24, not 23. Also #24, employing the VIPP mice, is not in accordance with their results contrary to what authors state. Please revise.

Response 12: We have now revised this.

Comment 13. Contrary to what authors claim several times, the IP₃-dependent pathway in astrocytes is not really considered “key” to Ca²⁺ signaling. It has appeared to mostly control Ca²⁺ signaling in the soma and primary branches but is quite irrelevant when it comes to calcium signals in microdomains/processes (see work from the Khakh group mostly). This would need to be revised and taken in consideration throughout the manuscript.

Response 13: We partly agree with the reviewer and have replaced the word “key” with “important” in relevant sentences: 1) page 2 line 9 in the original manuscript “*Genetic ablation of a key astrocytic Ca²⁺ signaling pathway resulted in fragmentation of slow-wave sleep, yet increased the frequency of sleep spindles*” was changed to “*Genetic ablation of an important astrocytic Ca²⁺ signaling pathway resulted in fragmentation of slow-wave sleep, yet increased the frequency of sleep spindles*” in the revised manuscript page 2 line 9; 2) page 13 line 2 in the original manuscript “*Genetic ablation of a key astrocytic Ca²⁺ signaling pathway altered sleep architecture and resulted in fragmentation of SWS, yet increased the frequency of sleep spindles*” was changed to “*Genetic ablation of an important astrocytic Ca²⁺ signaling pathway altered sleep architecture and resulted in fragmentation of SWS, yet increased the frequency of*

sleep spindles” in the revised manuscript page 13 line 6. Still, even though there are several other sources of astrocytic Ca^{2+} signals, we believe that IP_3 mediated pathway is one of the major Ca^{2+} signaling pathways in astrocytes as shown by numerous publications (Verkhatsky and Nedergaard, 2018). Furthermore, the IP_3 pathway seems of particularly importance in sleep as this pathway is key to neuromodulatory driven astrocytic Ca^{2+} events (Ding et al., 2013; Petravicz et al., 2008).

Comment 14. Fig. 1, Sup Fig. 6 and others. Please provide a vertical scale and units of EEG, EMG and frequency band powers. I am surprised to see “arbitrary units” when the axis says “Power”. Power does have a unit. Please explain and amend.

Response 14: We thank the reviewer for an observant comment. We have now included a vertical scale bar and units for EEG and EMG to Fig. 1c, d (see revised figure below and Fig. 1 in the revised manuscript). Supplementary Fig. 6 (now Supplementary Fig. 13 in the revised manuscript) has the scale and units as the y axis. The power has “arbitrary units” because it is normalized to the average total power in the 0.5–30 Hz frequency range during NREM sleep. We have added the word “*Normalized*” to the y axis labels of all ECoG power plots in Figs. 1, 5 and 8, and Supplementary Figs. 2 and 13 in the revised manuscript. We have also clarified this in Fig. 1 by changing Fig. 1e legend from “(E) *Normalized ECoG power density spectrum of sleep-wake states, n = 6 mice, average \pm SEM*” in the original manuscript (page 33, lines 11-12) to “(E) *ECoG power density spectrum of sleep-wake states, normalized to the average total power in the 0.5–30 Hz frequency range during NREM sleep, n = 6 mice, average \pm SEM*” in the revised manuscript; and by adding the following sentence “*ECoG power in A–E was normalized to the average total power in the 0.5–30Hz frequency range during NREM sleep*” to Supplementary Fig. 13 legend in revised manuscript.

Comment 15. Figure 3 panel C needs to be simplified.

Response 15: We have now replaced images in Fig. 3 panel C in the original manuscript with larger magnification images to simplify the figure (Fig. 4a in the revised manuscript)

Reviewer #2

The manuscript by Bojarskaite and colleagues investigates the role of glial calcium signaling in arousal states using the mouse as a model organism. The study builds on previous literature showing that glia may be relevant for regulating sleep states, but attempts to go a step further in delineating a role for IP3-dependent Ca²⁺ signaling in the process. The study is not uninteresting. It establishes a head-fixed two photon imaging prep to concurrently image neural and glial populations in mice undergoing state transitions, allowing for the observation that glial Ca²⁺ is enhanced upon arousal from sleep but not from transitions between NREM and REM sleep (Figs 1-4). To me that is the most robust finding in this study.

The rest is preliminary, unconvincing or both, making the whole story as it currently stands more appropriate for a specialized audience rather than of broad impact.

We thank the reviewer for taking time to review our paper. We have now added additional experiments, performed new analyses and rewritten multiple parts of our manuscript to strengthen our results and highlight our main findings. We believe that with this our study is more convincing and better suited for a broad impact.

Major:

Comment 1. Unclear to me conceptually why the barrel cortex is the object of study here. Is there a particular reason for this? This is not a brain region that is tied to arousal regulation in any particular way so it's hard to say why it was picked. It also raises the important point of how general are the findings here to other cortical/subcortical areas.

Response 1: The reviewer has raised an interesting question. The main focus of our study was the investigation of how different brain states, in particular three different states of natural sleep (NREM, IS and REM), affect the Ca^{2+} signaling of cortical astrocytes. Different brain states are associated with global changes in neuronal activity and brain neurochemistry which presumably affect the entire cortex. Moreover, a paper by Niethard et al shows no differences in neuronal calcium activity across sleep-wake states in different cortical areas encompassing parietal, occipital lobes and sensorimotor cortices (Niethard et al., 2016b), further supporting the global nature of brain states. We chose barrel cortex for several reasons. First, barrel cortex is an intensively studied and well-characterized model system. Second, one can functionally map the barrel cortex which allows for imaging at exactly the same location in every mouse without the need to rely on stereotactic coordinates. Third, it is easy to monitor the input to barrel cortex by tracking whisker movement, whereas for other cortices it would be difficult to control for variables which could affect the data e.g. for auditory cortex – sounds from the surroundings, for visual cortex – laser light flashes. We have addressed this now in the RESULTS section in the revised manuscript page 4 line 7: “*We chose barrel cortex because it is intensively studied and well-characterized system with known circuitry, which can be reliably mapped by intrinsic imaging and the input easily monitored by whisker tracking*(Feldmeyer et al., 2013)”.

Comment 2. What exactly does Figure 5 say? There are multiple traces picked showing neural somata transients but no behavioral correlate is shown to understand what these responses are for. Basically there are two major problems with how the authors are relating neural activity (or glial activity for that matter) to behavior:

2.1 waking/locomotion/whisking are not unitary states, and the details of what the animal is doing will matter in understanding what the circuit is doing and what type of differences are meaningful across conditions

Response 2.1: Needless to say, we agree with the reviewer that animal behavior is crucial for understanding and interpreting astrocyte-neuron interaction. In this paper we have identified the different wakefulness states based on what the animal was doing at any given timepoint, therefore we believe that we have taken

this account. However, as we also state in the original manuscript (page 8 lines 3–4): “ Ca^{2+} signals are not necessarily evenly distributed within a brain state”. Therefore, we also explored signaling related to behavioral transitions (Fig. 5 in the revised manuscript). Moreover, an even more dissected behavior is shown in Supplementary Figs. 6 and 7 where both astrocytic and neuronal Ca^{2+} traces are also aligned to continuous wheel/whisker/EMG/ECOG traces.

2.2 The correlation between what glia and neurons do across states needs to be adjusted for the differences in the frequency of both events across states. The authors ought to do subsampling of their waking data to make sure these correlations are robust to general reduction in sample size that they report.

Response 2.2: The correlation coefficient between neuronal and astrocytic Ca^{2+} activity shown in Fig. 5f in the original manuscript describes the relation between astrocytic Ca^{2+} signals (ROA frequency trace) and neuronal Ca^{2+} signals in the neuropil (in the regions outlined by black circles in Fig. 5a in the original manuscript). The neuronal Ca^{2+} activity extracted from the neuropil is a continuous signal with no clearly defined events, or event frequency, but rather ‘oscillations’. Even though there are less astrocytic Ca^{2+} signals during sleep compared to locomotion and whisking, which could potentially lead to a lower correlation detected in sleep, it is important to note that quiet wakefulness has much less astrocytic Ca^{2+} signals compared to locomotion and whisking, but the correlation coefficient is very similar for all wake states, independent of the differences in the frequency. Hence, the correlation data shown in Fig. 5f in the original manuscript is unlikely to be simply a result of differences in Ca^{2+} event frequency in neurons and astrocytes across states. To illustrate lower astrocyte-neuron correlation during sleep compared to wakefulness, we have now also included examples of astrocytic ROA frequency traces aligned to the neuropil Ca^{2+} trace in Fig. 6d in the revised manuscript.

Comment 3. The knockout data is simply not satisfying and seems like an afterthought here. The authors have the potential to make this into a broad interest study by simply determining the mechanism underlying this change in state regulation across the genotypes. Ideally, the authors would do this in a circuit-specific manner, not with a general knockout so that they can tighten their causal inference. Basically, figure out why the KO mice have fragmented sleep (less glial release of a substance that depends on this Ca^{2+} signal?)

Response 3: The reviewer has raised an important point. To improve our KO mice data and to understand the differences in sleep between genotypes, we have now 1) performed new experiments – compared sleep between WT and IP3R2KOs in freely behaving animals; 2) performed additional analyses – quantified the number of state transitions, microarousals and awakenings in both genotypes and evaluated the relationship of these transitions to the astrocytic Ca^{2+} signals; 3) rewritten the part of the manuscript dealing with the knockout data.

This is addressed in Figure 8 (also attached on page 12 of this rebuttal) and in the re-written RESULTS section “*Our observations of astrocytes during sleep revealed a spatiotemporal specificity of Ca^{2+} activity, particularly during SWS, that could indicate causal roles for astrocytic signaling in sleep regulation. To*

identify these roles of astrocytic Ca^{2+} signals in sleep, we employed the *Itp2^{-/-}* mouse model, in which astrocytic Ca^{2+} signaling is strongly attenuated, but not abolished (as shown in experiments on awake and anesthetized mice)(Srinivasan et al., 2015).

In agreement with previous reports, we found that *Itp2^{-/-}* mice exhibited reduced astrocytic Ca^{2+} signaling in all states of wakefulness as measured by ROA frequency and active voxels ($x-y-t$) (Supplementary Fig. 15a, b)(Srinivasan et al., 2015). However, astrocytic Ca^{2+} activity measured by both the percentage of active voxels and ROA frequency did not significantly differ between WT and *Itp2^{-/-}* mice during sleep (Fig. 7b, c). Even so, *Itp2^{-/-}* mice exhibited Ca^{2+} signals with disrupted spatiotemporal features – namely, longer duration and smaller spatial extent (Fig. 7a, d, e). This finding was observed in all states of wakefulness, but during sleep was restricted to SWS (NREM and IS states).

Next, we investigated whether IP_3 -dependent astrocytic Ca^{2+} signaling had any effect on sleep dynamics. We compared sleep architecture and spectral ECoG properties between the two genotypes and found that *Itp2^{-/-}* mice exhibited more frequent NREM and IS bouts that were of shorter duration than in the WT mice (Fig. 8a, b). More fragmented SWS sleep could be a consequence of more frequent microarousals (short wakefulness intrusions characterized by a reduction of low-frequency ECoG power) and awakenings which interrupt the sleep states, or more frequent NREM-to-IS and IS-to-NREM transitions. The number of awakenings did not differ between the genotypes in any of the sleep states (Fig. 8d). However, we found that *Itp2^{-/-}* mice have ~20 more microarousals per hour than WT mice (Fig. 8f). Such microarousals were associated with abrupt increases in astrocytic Ca^{2+} signaling in WT mice, whereas no such response was observed in *Itp2^{-/-}* mice (Fig. 8g). Surprisingly, *Itp2^{-/-}* mice were completely devoid of the prominent astrocytic Ca^{2+} increases seen upon awakenings in WT mice (Fig. 8e).

NREM-to-IS as well as IS-to-NREM transitions were more frequent in *Itp2^{-/-}* mice compared to WT mice (Fig. 8h), indicating abnormal SWS state dynamics in the knockouts. Interestingly, in WT mice, NREM-to-IS transitions were preceded by a decrease in Ca^{2+} signaling, whereas IS-to-NREM transitions were followed by an increase in astrocytic Ca^{2+} signaling. This was not the case in *Itp2^{-/-}* mice (Fig. 8i), and it is tempting to hypothesize that IP_3 mediated astrocytic Ca^{2+} signaling is important to sustain uninterrupted SWS by regulating NREM and IS state transitioning and possibly preventing microarousals.

Finally, we assessed the spectral ECoG properties of NREM, IS and REM sleep between the genotypes. We detected a decrease in delta power during NREM sleep, an increase in theta during REM sleep and an increase in sigma power during IS sleep in *Itp2^{-/-}* mice (Fig. 8j). ECoG activity in the sigma frequency range is indicative of sleep spindles – bursts of neuronal activity linked to memory consolidation(Diekelmann and Born, 2010). As *Itp2^{-/-}* mice displayed higher sigma power in IS, we next evaluated the occurrence of sleep spindles in WT and *Itp2^{-/-}* mice (Fig. 8k). The frequency of sleep spindles in IS sleep was indeed almost twice as high in *Itp2^{-/-}* mice than in WT mice (Fig. 8l). Intriguingly, sleep spindles were followed by an IP_3 -dependent increase in astrocytic Ca^{2+} signals (Fig. 8m). These data indicate a role for astrocytic IP_3 -mediated Ca^{2+} signaling pathway in regulating the architecture and brain rhythms of slow wave sleep.”

Unfortunately determining the mechanism underlying the change in state regulation across the genotypes in a circuit specific manner is outside the scope of the present paper. The goal of our study was to provide the first description of astrocytic Ca^{2+} activity during natural sleep and to determine the importance of astrocytic Ca^{2+} signals on regulation of sleep. We state in the discussion page 15 line 24 in the original manuscript that future studies should study circuit specificity: “*Future studies should delineate those specific pathways and circuits responsible for astrocytic IP_3 mediated modulation of sleep and resolve whether aberrant Ca^{2+} signaling in astrocytes underlies sleep disorders*”.

We have also elaborated on these issues in the DISCUSSION on page 15 line 13: “*However, since the gene knockout was global, we cannot rule out that altered astrocytic Ca^{2+} signaling in non-neocortical (e.g. brainstem) regions affects neurons regulating SWS. Similarly, we did not study whether the manipulation of astrocytic Ca^{2+} signaling affected specific cortical neurons thought to regulate SWS and spindles (Niethard et al., 2016a; Seibt et al., 2017). Future studies should delineate those specific pathways and circuits responsible for astrocytic IP_3 mediated modulation of sleep and investigate whether aberrant Ca^{2+} signaling in astrocytes underlies sleep disorders*”.

Summary: I think this is a good study, but findings are preliminary and some require tightening

We are pleased to see that the reviewer thinks our study is good, and we believe that the new experimental data from freely behaving animals, additional analyses and reorganized manuscript helps to tighten our study.

Reviewer #3

This is an interesting and well-performed study dealing with an important and timely issue, the impact of astroglial activity on sleep. The authors use gene-encoding neuron- and glia- specific Ca^{2+} indicators, and a combination of physiological monitoring techniques, to relate phases of wakefulness and sleep to astroglial activity in the mouse barrel cortex. They found a strong relationship there, which prompts them to test a straightforward hypothesis that a major astroglial Ca^{2+} signalling pathway involving IP_3 receptors plays a critical role in sleep regulation. Testing this hypothesis with a knockout model confirms their conclusions. The combination of new approaches and new results provides sufficient novelty and impact to the study of sleep. I have relatively minor comments, mainly related to data analyses, that the authors might want to deal with.

We are pleased to see that the reviewer thinks our study is well-performed and dealing with a timely issue. We have now addressed all reviewer’s comments.

Comment 1. Fig. 2 and related data. The XYT ROA diagrams (Fig. 2A-B) provide a powerful tool to record and compare astrocytic Ca^{2+} signals that vary widely in their frequency, amplitude, volume spread, and duration. In this respect, summary data (Fig. 2C-D) contain somewhat curtailed information: the percentage of active voxels (panel C) appears to combine duration and extent of the events, and the ROA frequency

data (panel D) might depend on the pixel-connecting criteria in the ROA method (see below). The authors are encouraged to consider a more thorough analysis of their ROA data, perhaps based directly on their XYT records. This comment, however, does not affect qualitative conclusions of the study.

Response 1: We agree with the reviewer and have now included information about ROA size, volume and duration in Supplementary Fig. 8 of the revised manuscript, and discussed the findings in page 5 lines 24 of the revised manuscript: *“We used the ROA analysis to explore astrocytic Ca²⁺ signaling during wakefulness and natural sleep (Fig. 2). Astrocytic Ca²⁺ signals across the sleep-wake cycle displayed a broad repertoire of size, duration and volume (Supplementary Fig. 8). The spatial extent of ROAs ranged from ~0.9 μm² (lower detection limit) to the full FOV (Supplementary Fig. 8a), whereas the duration of the events ranged from 0.05 s to 100 s (Supplementary Fig. 8b). The majority of astrocytic Ca²⁺ events were small and short-lasting across all sleep-wake states (~80% events < 10 μm² and < 1s) (Supplementary Fig. 8d, e, f). On average, Ca²⁺ events were of largest area and volume during active wakefulness (locomotion and whisking) (Supplementary Fig. 8g, h), and of longest duration during sleep (Supplementary Fig. 8i).”*

Comment 2. Supplementary Fig. 2, ROA method. The criteria for connecting active pixels need to be clarified. A common problem here is that 2D image acquisition cannot tell whether neighbouring active pixels represent fully separated Ca²⁺ signals or whether they represent parts of one contiguous 3D signal that is out of focus. This problem cannot be solved by the ROA method as it is and needs to be acknowledged. Again, it should have little impact on the qualitative conclusions of the study because similar segmentation procedures have been applied throughout the data sets. At the same time, the authors are encouraged to show more caution towards their quantitative estimates.

Response 2: We agree with the reviewer that the merging of several single Ca²⁺ events into larger events will make the precise estimation of single Ca²⁺ events difficult. However, as astrocytes are connected in a syncytium, it is not necessarily correct to think of them as separate events even though they originate at distinct close-by locations. Moreover, the recently published AQuA algorithm developed by Kira Poskanzer (Wang et al., 2019) involves ‘deconvolution’ of these merged signals into separate signals. As reviewer 1 requested (please see comment 2 of reviewer 1), we have analyzed our data with AQuA algorithm and detected the same astrocytic Ca²⁺ signaling trends across sleep-wake cycle as with our ROA method (see figure below and Supplementary Fig. 9 in the revised manuscript). The criteria of connecting pixels is stated on line 8, page 5: *“Connecting adjacent pixels in space and time...”* – neighboring pixels were simply joined to form larger events. The rest of the reviewer’s concerns are now discussed in the DISCUSSION, line 9-19, page 13: *“Further, analysis of our data with the recently published AQuA tool²⁵ resulted in almost identical astrocytic dynamics as detected with our ROA tool, but with the added benefit of fewer input parameters and shorter processing times. This data processing efficiency is increasingly important when coupled with high imaging frame rates, which result in large datasets but are necessary for capturing rapid (subsecond) changes in astrocytic Ca²⁺ signaling^{17,19}. It is*

important to mention that our ROA method lacks spatial signal deconvolution to identify local Ca^{2+} signals that merge into larger signaling events. Even so, one could argue that, as astrocytes are connected in a syncytium, it is not necessarily conceptually most fruitful to think of small localized events merging into larger events as separate events. It is also important to acknowledge that single-plane imaging will lose information about the three-dimensional nature of astrocytic Ca^{2+} signals¹⁷.”

Supplementary Figure 9. Ca^{2+} signaling in astrocytes across sleep-wake states analyzed by AQuA software. (A and B) Ca^{2+} signaling in astrocytes across sleep-wake states analyzed by AQuA (A) and ROA (B) methods. Data represented as estimates \pm SEM $n = 6$ mice, 278 trials, * $p < 0.05$, ** $p < 0.005$, *** $p < 0.0005$. For details on statistical analyses, see Methods.

Comment 3. Figs. 3-4: As noted above, ROA frequency alone is a poor representative of signal dynamics. While this is partly acknowledged in the text, the data presentation here should include an unambiguous reference to ROA size and duration.

Response 3: We have now included information about ROA size and duration across sleep-wake states for Figs. 3-4 in the original manuscript (Figs. 4-5 in the revised manuscript) in Supplementary Figs. 8 and 14 in the revised manuscript.

Comment 4. Fig. 5B data suggest that, during locomotion, barrel cortex neurons generate detectable Ca^{2+} signals, which supposedly reflect action potential firing, at a frequency of $\sim 1/\text{min}$. This slow rate is unrealistic and probably reflects relatively low sensitivity of Ca^{2+} imaging. In other words, in the present settings, neuronal activity is only detected above some high spiking rate. Without a proven direct relationship between neuronal spiking and recorded Ca^{2+} signals the spectral power data in Fig. 5E are

uninterpretable. This is exacerbated by the fact that Ca²⁺ indicators heavily filter out higher frequencies of the underlying Ca²⁺ activity. These data need a serious rethink.

Response 4: We agree with the reviewer that neuronal firing rate of 1/min is unrealistically slow and that this could partly stem from lowpass filtering effects of the indicator. Another contributing factor is that we report of the average Ca²⁺ signal frequency of all neuronal somata in the FOV. The firing rate varies greatly among the neurons in barrel cortex as reported by others (O'Connor et al., 2010) and as seen in our data (Supplementary Figs. 6i and 7i). Indeed, 14% of neurons are “silent” – do not have a single Ca²⁺ signal, which skews the average Ca²⁺ signal frequency towards unrealistically slow rates. We have addressed these issues in the DISCUSSION part of the revised manuscript, page 17 line 16: *“Neuronal activity data obtained from fluorescent Ca²⁺ sensors as a proxy for neuronal firing should be interpreted with caution. Neuronal firing patterns are at least partially state-dependent (Miyawaki et al., 2019) indicating that due to the sensitivity and kinetics of the sensor, Ca²⁺ readouts could be affected in a non-linear manner, hence not faithfully reflecting neuronal firing across various states. Moreover, as the neuronal Ca²⁺ event detection is threshold-based, it is likely that neuronal activity is only detected above a certain spiking rate, possibly contributing to an underestimation of neuronal activity levels. Even so, our neuronal Ca²⁺ data across sleep-wake states (Fig. 6b) is in line with other Ca²⁺ imaging and electrophysiological studies showing reduced neuronal firing during sleep compared to wakefulness (Kanda et al., 2017; Niethard et al., 2016a). Future electrophysiological studies using unit recordings will have to confirm our observations.”*

In agreement with the reviewer, we have now removed the power analyses of neuronal Ca²⁺ signals in the neuropil (panels d and e from Figure 5 in the original manuscript) due to its technical limitations.

Comment 5. Fig. 5D: Neuronal Ca²⁺ activity in the bulk of tissue may tend to represent either axonal or dendritic activity depending on their relative volume fraction and on the pre- vs postsynaptic cell excitability. In any case, the present Ca²⁺ imaging method cannot distinguish individual spikes (see above), which severely limits any quantitative assessment.

Response 5. We thank the reviewer for an observant comment. We have now removed the power analyses of neuronal Ca²⁺ signals in the neuropil (panels d and e from Figure 5 in the original manuscript)

Comment 6. Fig. 5F: These data require more detailed information about what was measured (amplitude, volume, duration, binary time series, etc.) to obtain correlation coefficients, and what statistical tests were used to compare them. The graph should probably show connected data points in individual animals.

Response 6: We thank the reviewer for an observant comment. We have now addressed this in the METHODS part in the revised manuscript (page 27, line 9): *“For astrocytic Ca²⁺ signals, an average ROA frequency trace was extracted from all neuropil ROIs per FOV from the GFAP-GCaMP channel. For the neuronal Ca²⁺ signal, an average $\Delta F/F_0$ trace was extracted from all neuropil ROIs per FOV from the SYN-jRGECO1a channel. The baseline for neuronal signal was estimated by the mode of the trace. Both*

neuronal and astrocytic traces were subsequently smoothed with a gaussian filter ($\sigma = 0.25$ s). The maximum Pearson cross-correlation was calculated between astrocytic and neuronal Ca^{2+} traces for each episode. The mean lag in all comparisons were less than a single frame.”

Comment 7. Fig 7F: Ordinate units are not shown.

Response 7. Figure 7f in the original manuscript is now Fig. 8e in the revised manuscript. We have clarified the ordinate units in the Fig. 8e.

References

- Ali, F., and Kwan, A.C. (2019). Interpreting in vivo calcium signals from neuronal cell bodies, axons, and dendrites: a review. *Neurophotonics* 7, 1.
- Bekar, L.K., He, W., and Nedergaard, M. (2008). Locus coeruleus α -adrenergic-mediated activation of cortical astrocytes in vivo. *Cereb. Cortex* 18, 2789–2795.
- Cox, J., Pinto, L., and Dan, Y. (2016). Calcium imaging of sleep–wake related neuronal activity in the dorsal pons. *Nat. Commun.* 7, 10763.
- Dana, H., Mohar, B., Sun, Y., Narayan, S., Gordus, A., Hasseman, J.P., Tsegaye, G., Holt, G.T., Hu, A., Walpita, D., et al. (2016). Sensitive red protein calcium indicators for imaging neural activity. *Elife* 5, 1–24.
- Diekelmann, S., and Born, J. (2010). The memory function of sleep. *Nat. Rev. Neurosci.* 11, 114–126.
- Ding, F., O'Donnell, J., Thrane, A.S., Zeppenfeld, D., Kang, H., Xie, L., Wang, F., and Nedergaard, M. (2013). α 1-Adrenergic receptors mediate coordinated Ca^{2+} signaling of cortical astrocytes in awake, behaving mice. *Cell Calcium* 54, 387–394.
- Feldmeyer, D., Brecht, M., Helmchen, F., Petersen, C.C.H., Poulet, J.F.A., Staiger, J.F., Luhmann, H.J., and Schwarzh, C. (2013). Barrel cortex function. *Prog. Neurobiol.* 103, 3–27.
- Kanda, T., Ohyama, K., Muramoto, H., Kitajima, N., and Sekiya, H. (2017). Promising techniques to illuminate neuromodulatory control of the cerebral cortex in sleeping and waking states. *Neurosci. Res.* 118, 92–103.
- Lecci, S., Fernandez, L.M.J., Weber, F.D., Cardis, R., Chatton, J.-Y., Born, J., and Lüthi, A. (2017). Coordinated infraslow neural and cardiac oscillations mark fragility and offline periods in mammalian sleep. *Sci. Adv.* 3, e1602026.
- Miyawaki, H., Watson, B.O., and Diba, K. (2019). Neuronal firing rates diverge during REM and homogenize during non-REM. *Sci. Rep.* 9, 1–14.
- Niethard, N., Hasegawa, M., Itokazu, T., Oyanedel, C.N., Born, J., and Sato, T.R. (2016a). Sleep-Stage-Specific Regulation of Cortical Excitation and Inhibition. *Curr. Biol.* 26, 2739–2749.
- Niethard, N., Hasegawa, M., Itokazu, T., Oyanedel, C.N., Born, J., and Sato, T.R. (2016b). Sleep-Stage-Specific Regulation of Cortical Excitation and Inhibition. *Curr. Biol.* 26, 2739–2749.
- Paukert, M., Agarwal, A., Cha, J., Doze, V.A., Kang, J.U., and Bergles, D.E. (2014). Norepinephrine controls astroglial responsiveness to local circuit activity. *Neuron* 82, 1263–1270.

- Peron, S., Chen, T.W., and Svoboda, K. (2015). Comprehensive imaging of cortical networks. *Curr. Opin. Neurobiol.* 32, 115–123.
- Petravicz, J., Fiacco, T.A., and McCarthy, K.D. (2008). Loss of IP3 receptor-dependent Ca²⁺ increases in hippocampal astrocytes does not affect baseline CA1 pyramidal neuron synaptic activity. *J. Neurosci.* 28, 4967–4973.
- Seibt, J., Richard, C.J., Sigl-Glöckner, J., Takahashi, N., Kaplan, D.I., Doron, G., De Limoges, D., Bocklisch, C., and Larkum, M.E. (2017). Cortical dendritic activity correlates with spindle-rich oscillations during sleep in rodents. *Nat. Commun.* 8, 1–12.
- Song, J., Um, Y.H., Kim, T.W., Kim, S.M., Kwon, S.Y., and Hong, S.C. (2018). Sleep and anesthesia. *Sleep Med. Res.* 9, 11–19.
- Srinivasan, R., Huang, B.S., Venugopal, S., Johnston, A.D., Chai, H., Zeng, H., Golshani, P., and Khakh, B.S. (2015). Ca²⁺ signaling in astrocytes from *Ip3r2*^{-/-} mice in brain slices and during startle responses in vivo. *Nat Neurosci* 18, 708–717.
- Takahashi, K., Lin, J.S., and Sakai, K. (2006). Neuronal activity of histaminergic tuberomammillary neurons during wake-sleep states in the mouse. *J. Neurosci.* 26, 10292–10298.
- Takahashi, K., Lin, J.S., and Sakai, K. (2009). Characterization and mapping of sleep-waking specific neurons in the basal forebrain and preoptic hypothalamus in mice. *Neuroscience* 161, 269–292.
- Takahashi, K., Kayama, Y., Lin, J.S., and Sakai, K. (2010). Locus coeruleus neuronal activity during the sleep-waking cycle in mice. *Neuroscience* 169, 1115–1126.
- Tung, A., and Mendelson, W.B. (2004). Anesthesia and sleep. *Sleep Med. Rev.* 8, 213–225.
- Verkhatsky, A., and Nedergaard, M. (2018). Physiology of Astroglia. *Physiol. Rev.* 98, 239–389.
- Wang, Y., DelRosso, N. V., Vaidyanathan, T. V., Cahill, M.K., Reitman, M.E., Pittolo, S., Mi, X., Yu, G., and Poskanzer, K.E. (2019). Accurate quantification of astrocyte and neurotransmitter fluorescence dynamics for single-cell and population-level physiology. *Nat. Neurosci.* 22, 1936–1944.
- Yüzgeç, Ö., Prsa, M., Zimmermann, R., and Huber, D. (2018). Pupil Size Coupling to Cortical States Protects the Stability of Deep Sleep via Parasympathetic Modulation. *Curr. Biol.* 28, 392–400.e3.

Reviewer comments, second version:

Reviewer #1 (Remarks to the Author):

Overall, while I still have some reservations, authors did a compelling job at amending their manuscript and provided convincing additional data to support and improve their conclusions. I also want to acknowledge the efforts authors have made to make their revisions and their answer extremely clear. It appears authors have carried out a lot of additional work, most of which should be included in the supplemental figures. I have a few more questions or points I believe need to be clarified, but I do think that the paper and its main findings now fit well within the scope and impact of Nature Com.

Comments:

1-The overall sleep architecture of head-fixed mice seems well preserved when compared to freely behaving mice, in particular NREM and IS, which is important for authors' conclusions. However, I do not agree that the few differences found between freely behaving and head-fixed mice are "subtle" (page 5 first paragraph and Sup 2). The differences in REM bouts are quite striking in fact, including the EEG power of these events. This should be adequately acknowledged, rather than dismissed, in particular because this does not put the overall findings and conclusions in jeopardy. Though preferably, authors would briefly discuss how this could impact/explain some of their results or be a confounding factor in the MS.

2-I find the state-dependent location paragraph very unclear. My understanding from this section is that authors found that active areas are mainly the same during all states, but are particularly preserved within a given state? In other words, there might be some 'hotspots' that turn on during specific states. Is that correct? If so, I do not think authors performed the adequate statistical analysis. One would expect a statistical comparison assessing whether the degree of overlap is indeed significantly greater within states than across states. Also, it is unclear how authors determined the chance level. I find the related method section somewhat obscure, though this could be that I am unfamiliar with this type of analysis: "We compare the observed values with overlap values coming from a suitable null model. We generated 1000 datasets from this null model and let these null-datasets be of the same size as the 10 observed data, and with precisely the same number of different state-state combinations" What is the "suitable null model" authors refer to here? Please clarify these points and amend the text accordingly, as this could be (if true) a major finding or (if not) somewhat of an overstatement.

3-Regarding the IP3R2 KO studies, are the "wild-type" mice indeed wild-type or is this a minor misstatement? One would expect KOs to be compared to their control littermates, not to WT mice.

4-Figure 6, please clarify what parameter is measured and compared in each panel between neurons and astrocytes (amplitude, frequency?) and amend figures and legend.

5-If the ECoG power is internally normalized then it simply does not have a unit and axis should state "normalize" instead of "A.U." This has not been amended on most of the figures as far as I can see.

6-Lastly, but importantly, I believe most of the figures provided by authors in their answers to my comments are worth adding to their set of Sup Figures.

Reviewer #2 (Remarks to the Author):

The authors have addressed my previous comments and I have no further suggestions to make. The new Figure 8 is cool, well done.

Reviewer #3 (Remarks to the Author):

The authors have addressed the comments in a satisfactory manner.

Author rebuttal, second version:

Reviewer #1 (Remarks to the Author):

Overall, while I still have some reservations, authors did a compelling job at amending their manuscript and provided convincing additional data to support and improve their conclusions. I also want to acknowledge the efforts authors have made to make their revisions and their answer extremely clear. It appears authors have carried out a lot of additional work, most of which should be included to the supplemental figures. I have a few more questions or points I believe need to be clarified, but I do think that the paper and its main findings now fit well within the scope and impact of Nature Com.

We thank the reviewer for acknowledging our work and another thorough assessment. A point-by-point list of answers is included below.

Comments:

1-The overall sleep architecture of head-fixed mice seems well preserved when compared to freely behaving mice, in particular NREM and IS, which is important for authors' conclusions. However, I do not agree that the few differences found between freely behaving and head-fixed mice are "subtle" (page 5 first paragraph and Sup 2). The differences in REM bouts are quite striking in fact, including the EEG power of these events. This should be adequately acknowledged, rather than dismissed, in particular because this does not put the overall findings and conclusions in jeopardy. Though preferably, authors would briefly discuss how this could impact/explain some of their results or be a confounding factor in the MS.

We agree with the reviewer that the differences between head-fixed and natural sleep could be addressed more thoroughly. We have now rephrased the following sentence in the original manuscript (line 2, page 5) "*We found nearly identical sleep characteristics between the two conditions, except a few subtle differences (Supplementary Fig. 2)*" to "*We found nearly identical sleep characteristics between the two conditions, except less time spent in REM sleep and higher ECoG power in delta and theta range in the head-fixed condition (Supplementary Fig. 2). The increase in delta and theta power could at least partially be explained by delayed sleep onset and consequently higher sleep pressure in head-fixed mice (Supplementary Fig. 2a), which has been*

shown to increase both delta and theta ECoG power in NREM and REM sleep²⁴." in the revised manuscript (lines 3-7, page 5).

2-I find the state-dependent location paragraph very unclear. My understanding from this section is that authors found that active areas are mainly the same during all states, but are particularly preserved within a given state? In other words, there might be some 'hotspots' that turn on during specific states. Is that correct? If so, I do not think authors performed the adequate statistical analysis. One would expect a statistical comparison assessing whether the degree of overlap is indeed significantly greater within states than across states. Also, it is unclear how authors determined the chance level. I find the related method section somewhat obscure, though this could be that I am unfamiliar with this type of analysis: "We compare the observed values with overlap values coming from a suitable null model. We generated 1000 datasets from this null model and let these null-datasets be of the same size as the 10 observed data, and with precisely the same number of different state-state combinations" What is the "suitable null model" authors refer to here? Please clarify these points and amend the text accordingly, as this could be (if true) a major finding or (if not) somewhat of an overstatement.

We agree with the reviewer that this paragraph could have been written more clearly. In our original analysis we found a generally low level of overlap of active areas, but some indications that there may be a small subset of state specific activation. A comparison of whether overlap is higher between episodes within states compared to across states, would not take into account the varying levels of activation between states – i.e. it would not say if the observed level of overlap was significantly higher than what would be the case given a random spatial distribution of events. This is why we chose to compare the degree of overlap to a simulated chance level (determined by looking at overlap between activity from different fields-of-view). Still, we agree that this analysis do not elucidate all interesting aspects of this phenomenon and does not warrant a conclusion of state-specificity as stated in our original manuscript. After careful consideration with the two statisticians that are co-authors of the manuscript (CMC, GHH) we have come up with a new statistical analysis that more directly answers the reviewer's questions. As overlap estimated by the Jaccard index models the 'similarity' of overlap between episodes of same and different states, we now investigated the inverse – the 1 minus overlap – also known as the Jaccard distance. In addition, we added a level of refinement by using graded heatmaps of astrocytic Ca²⁺ activity for our comparisons (calculated as ROA density), which take into account the degree of activity, instead of using binary heatmaps of any activity, as in our previous analyses. We used a permutational multivariate analysis of variance to analyze the Jaccard distances. With our new analyses we found that 25% of FOVs exhibited state-specific activity

patterns in wakefulness ($p < 0.05$, Fig 4b). The degree of overlap explained by state, estimated by R^2 was relatively low (Fig. 4b). We found no state specific activation between sleep states, but it must be noted that the numbers of episodes per FOV is quite low, and we are hence probably underpowered to identify very small levels of specific activation. We then compared sleep states with states of wakefulness, and found that 50% of FOVs showed sleep-wake state specific differences (i.e. episodes of similar states were more similar than episodes of different states) and that R^2 here was generally higher, indicating that overall sleep is more different from overall wakefulness compared to difference across the states of wakefulness (locomotion to whisking to quiet wake), and also compared to difference across the states of sleep (NREM, IS, REM). It is important to highlight that our analysis probably depend heavily on the distance measure utilized and more refined modeling studies could potentially discover more subtle degrees of state specific activation. We have changed Figure 4 and figure legends, updated the materials and methods section and removed Supplementary Figure 12. We have also re-written the results section and the methods part to clarify our findings. New results section:

Spatial distribution of astrocytic Ca^{2+} signals across sleep-wake states

If astrocytic Ca^{2+} signals are specifically integrated in sleep-wake dependent circuitry, one would expect to find some stability of active regions specific to sleep-wake states. We found that generally the overlap of active areas between two episodes of the same state was relatively low (ca. 5%) except during episodes of locomotion (ca. 25%), where typically most of the FOV was active (Fig. 4a). To evaluate whether some of the astrocytic Ca^{2+} signals occurred at sleep or wakefulness specific locations, we first created individual heatmaps representing the level of Ca^{2+} activity of every episode of all of the sleep-wake states within a FOV. Then, we analyzed the difference between heatmaps, here defined as 1 minus the Jaccard similarity coefficient (see Methods), by performing a permutational multivariate analysis of variance²⁸. We first checked whether there was state specific overlap within sub-states of wakefulness (locomotion, whisking, quiet wake) and within sub-states of sleep (NREM, IS, REM) (Fig. 4b). Here, we found that 25% of FOVs (19 of 76) including only wakefulness sub-states, exhibited state specific activation – i.e. within a FOV there was a greater overlap between episodes of the same state (locomotion-locomotion, whisking-whisking, quiet wake-quiet wake) compared to episodes of different states (locomotion-whisking, locomotion-quiet wake, whisking-quiet wake) (p -values under 0.05 as represented by the dashed line, Fig. 4b, *left*). In these FOVs, the degree of overlap explained by sub-states of wakefulness was still relatively low, as indicated by a low R^2 (Fig. 4b, *right*). R^2 reflects the total difference in overlap between different states divided by the total difference in overlap both within and between states. A high R^2 value indicates that episodes within the same state are very similar, while episodes from different states are very different. No state specific activation was found between the sleep states (Fig. 4b, *left*).

Next, we assessed whether there could be activity patterns specific to either sleep or wakefulness. For FOVs with both sleep and wakefulness, 50% of FOVs (43 of 86) showed a significant level of state specific activation (Fig. 4c, *left*). R^2 of FOVs with both sleep and wakefulness states (Fig. 4c, *right*) was generally higher than R^2 of FOVs with only wakefulness or only sleep states (Fig. 4b, *right*), suggesting that active areas in sleep are somewhat different from areas that are active during wakefulness.

Taken together, these data show a low degree of overlap of astrocytic Ca^{2+} activity across sleep-wake states, but indicate a moderate degree of sleep and wakefulness specific spatial activation patterns.

New Methods section:

ROA overlap analysis

The overlap of active areas between episodes was evaluated by creating ROA heatmaps of the active voxels in each episode and estimating the overlap of these by calculating the Jaccard similarity coefficient. The Jaccard similarity coefficient is a number between 0 and 1, with 0 indicating no overlap and 1 indicating identical activation and was defined as

$$J(\mathbf{x}, \mathbf{y}) = \frac{\sum_i \min(x_i, y_i)}{\sum_i \max(x_i, y_i)}$$

where \mathbf{x} and \mathbf{y} are vectors of pixels in each heatmap being compared.

Conversely, one minus the Jaccard similarity coefficient is a measure of dissimilarity between episodes and is also known as the Jaccard distance. We then calculated the distance matrix between all episodes from all FOVs, and compared these using the permutational multivariate analysis of variance – PERMANOVA – framework^{28,61}. This method has a similar logic to ordinary ANOVA, but works on distance matrices. Say we have a FOV with 20 episodes belonging to three different states (i.e. locomotion, quiet, whisking) and we are interested in investigating whether episodes within states are more similar than episodes of different states. If the FOV we are studying has strictly state-specific activation locations the distances within states will all be small, and the distances across states will all be large. Then the PERMANOVA will return a high R^2 value for the groups highly affected by the state, and typically also a small p-value. The p-values in the PERMANOVA framework are obtained using permutations, and the approach does therefore not depend on any distributional assumptions. Further, the PERMANOVA framework includes both continuous covariates and factor variables (sleep-wake state is the factor variable in our case). Since the similarity between episodes is likely influenced by the overall activation of the episodes being compared, we have included average activation per episode as a covariate. Our result should therefore be interpreted as conditional on the total level of activation, i.e. given a certain level of activation do we see more similarities across states than between states? Only FOVs with 8 or more episodes were analyzed. The p-values in Fig. 4 represents the degree of which

episodes across states tend to be more different than episodes within states (one dot represents a FOV). In some FOVs we only have comparisons between the awake states or sleep states (black or red, respectively). The green dots represent FOV where we have comparisons for both sleep and wakefulness. We consider the Jaccard similarity coefficient to be an intuitive and natural distance measure. However, it is important to note that the interpretation of results could be highly dependent on the choice of distance/similarity measure.

New figure 4:

Fig. 4 Spatial distribution of astrocytic Ca^{2+} signals across sleep-wake states (A) Representative astrocytic Ca^{2+} activity maps outlining areas active during episode 1 (green) and episode 2 (blue) of each of the sleep-wake states, and the overlap between the active areas (red). (B) Every dot represents a single FOV. Red dots are FOVs with only wakefulness sub-states whereas black dots are FOVs with only sleep sub-states. (*Left*) The probability that active region overlap between same states (locomotion-locomotion, whisking-whisking, quiet wake-quiet wake, NREM-NREM, IS-IS, REM-REM) is higher than between different states. Only FOVs below

dashed line ($p = 0.05$) have overlap of active regions that can statistically significantly be explained by state specificity. (*Right*) The degree of overlap within episodes of the same state relative to the combined degree of overlap between episodes of same and different states (a high R^2 value indicates that episodes within the same state are very similar, while episodes from different states are very different). (C) Same as (B) but for FOVs with both sleep and wakefulness sub-states. $n = 6$ mice, 165 unique FOVs, 3793 episodes. For details on statistical analyses, see Methods.

3-Regarding the IP3R2 KO studies, are the “wild-type” mice indeed wild-type or is this a minor misstatement? One would expect KOs to be compared to their control littermates, not to WT mice.

We agree with the reviewer that littermates would have been the optimal controls. However, our IP3R2 KO strain have been backcrossed with the commercial C57BL/6J mice (the wild-type strain we use) at several occasions over altogether 15 generations (also to counteract genetic drift). We hence believe C57BL/6J is a valid control. Additional Information of the IP3R2 KO mouse line is now added in Methods: “*Itpr2^{-/-} mice were backcrossed into a C57BL/6J background for 15 generations*” (line 12–14, page 18 in the revised manuscript).

4-Figure 6, please clarify what parameter is measured and compared in each panel between neurons and astrocytes (amplitude, frequency?) and amend figures and legend.

In Figure 6 panel B neuronal Ca^{2+} event frequency in somata ROIs is measured indicated in y axis label “*Ca²⁺ events / neuronal soma / min*” and in figure legend as “*Frequency of Ca²⁺ signals in neuron somata across sleep-wake states*”.

In panel C we display the temporal location of an astrocytic Ca^{2+} event relative to a neuronal Ca^{2+} event, 1 black dot in the raster plot represents one astrocytic Ca^{2+} event. To clarify what is displayed in this panel we changed the text at the top of panel C from “*Astrocytic Ca²⁺ events in neuropil aligned to neuronal soma Ca²⁺ events*” to “*Raster plots of astrocytic Ca²⁺ event time points aligned to neuronal somata Ca²⁺ events (1 black dot = 1 astrocytic Ca²⁺ event)*”. We have also added “*Each black dot represents the temporal location of one astrocytic Ca²⁺ event relative to a Ca²⁺ event in a neuronal soma*” to the legend of panel C.

In panel D we display representative continuous traces of astrocytic Ca^{2+} event frequency in neuropil (measured as number of ROAs per $100 \mu\text{m}^2$ per minute) and neuronal Ca^{2+} signal $\Delta F/F_0$ in neuropil ROIs. We have now clarified this by changing panel D figure legend from “*Example traces of astrocytic ROA frequency and neuronal $\Delta F/F_0$ in neuropil ROIs*” to “*Example traces of continuous astrocytic Ca^{2+} event frequency in neuropil ROIs (number of ROAs per $100 \mu\text{m}^2$ per minute), and neuronal Ca^{2+} signal $\Delta F/F_0$ in neuropil ROIs during locomotion, whisking, quiet wakefulness (QW), NREM sleep, IS sleep and REM sleep.*”

In panel E we measured the correlation coefficient between the continuous traces of astrocytic Ca^{2+} event frequency in neuropil (measured as number of ROAs per $100 \mu\text{m}^2$ per minute) and neuronal Ca^{2+} signal $\Delta F/F_0$ in neuropil ROIs (the traces displayed in panel D). We have now clarified this by changing the panel E title from “*Neuronal Ca^{2+} vs. astrocytic Ca^{2+} ”* to “*Neuronal $\Delta F/F_0$ vs. astrocytic ROA frequency in neuropil*”, and by changing panel E legend from “*Correlation coefficient between astrocytic Ca^{2+} events in neuropil, measured as number of ROAs per $100 \mu\text{m}^2$ per minute, and neuropil Ca^{2+} across sleep-wake states*” to “*Correlation coefficient between continuous traces of astrocytic Ca^{2+} event frequency in neuropil, measured as number of ROAs per $100 \mu\text{m}^2$ per minute, and neuronal Ca^{2+} signal $\Delta F/F_0$ in neuropil across sleep-wake states*”.

5-If the ECoG power is internally normalized then it simply does not have a unit and axis should state “normalize” instead of “A.U.” This has not been amended on most of the figures as far as I can see.

We thank the reviewer for pointing this out. The “(a.u.)” have been removed from the y axis labels of all plots of normalized ECoG power (Fig. 8j and Supplementary Figs. 2e, 12a,b,c,e). All of the y axis labels of normalized ECoG power already state “Normalized” in the label.

6-Lastly, but importantly, I believe most of the figures provided by authors in their answers to my comments are worth adding to their set of Sup Figures.

We have now added the 3 figures used in the rebuttal letter in response to reviewer’s comments to Supplementary Fig. 15 and referred to the figure in Methods sub-section “Sleep-wake state scoring”

on page 21: “*Short and long quiet wakefulness episodes could represent different behavioral states, such as restful quiescence and freezing, and be influenced by the degree of habituation,*

all of which could be thought to affect astrocytic Ca²⁺ signaling patterns. In our dataset >90% of quiet wakefulness bouts were shorter than 10 s and there was no difference in estimated Ca²⁺ signaling levels when including long-lasting bouts compared to omitting them (Supplementary Fig. 15)."

Reviewer comments, third version:

Reviewer #1 (Remarks to the Author):

Authors did a phenomenal job, here again, at answering my comments, concerns and suggestions and amending their manuscript. In particular, the section about the state-dependent Ca²⁺ activity is now a lot more solid, in part because it is a lot more true to the actual results/observations. Overall, I think this is now a robust and rich study. It will be very welcomed by the field.

Author rebuttal, third version: